



# Increasing Diurnal and Seasonal Amplitude of Atmospheric Methane Mole Fraction in Central Siberia between 2010-2021

Dieu Anh Tran[1,2], Jordi Vilà-Guerau de Arellano[2,3], Ingrid T. Luijkx[2], Christoph Gerbig[1], Michał Gałkowski[1,4], Santiago Botía[1], Kim Faassen[2], Sönke Zaehle[1].

[1]Biogeochemical Signal Department, Max Planck Institute for Biogeochemistry, Jena, Germany.
[2]Meteorology and Air Quality Group, Wageningen University and Research, Wageningen, the Netherlands.
[3]Atmospheric Chemistry Department, Max Planck Institute for Chemistry, Mainz, Germany.
[4]Department of Applied Nuclear Physics, AGH University, Kraków, Poland.

*Correspondence to*: Dieu Anh Tran (atran@bgc-jena.mpg.de)

**Abstract.** Siberia's vast wetlands, permafrost, and boreal forests are significant, but their sources of methane ($CH_4$) are poorly quantified. Using vertical $CH_4$ profiles and meteorological data from the ZOtino Tall Tower Observatory (ZOTTO; 60°48′ N, 89°21′ E) in Central Siberia, we analyse long-term trends in $CH_4$ growth rates, seasonal patterns, and diurnal cycles from 2010 to 2021. Our results show a persistent long-term trend in $CH_4$ mole fractions and an insignificant increasing seasonal cycle amplitude, (2.12 ppb year$^{-1}$, p = 0.12) along with a pronounced late-summer $CH_4$ peak. Diurnal analysis reveals a growing summer $CH_4$ amplitude over the analysed decade (5.55 ppb year$^{-1}$, p = 0.002), driven by rising nighttime fluxes strongly correlated with soil temperature ($R^2 = 0.7$, p < 0.001) and moisture ($R^2 = 0.60$, p = 0.031). Notably high nighttime $CH_4$ fluxes occurred in 2012 and 2019 due to wildfires and in 2016, likely due to wetland activity induced by higher temperature. These findings suggest that increasing late-summer $CH_4$ emissions, primarily from wetlands to the west and southwest of ZOTTO, contribute to the overall $CH_4$ rise. Our study underscores the importance of continuous, high-frequency greenhouse gas observations for accurately quantifying regional $CH_4$ trends.

## 1 Introduction

Methane ($CH_4$) is an important greenhouse gas, accounting for approximately 16% of global greenhouse gases radiative forcing, and representing the second largest contributor to current anthropogenic warming (IPCC, 2023). Since 1970, the global mean atmospheric $CH_4$ mole fraction has risen from 1630 ppb to 1774 ppb by 1999. This was followed by a period of stalled growth between 1999 and 2006, after which $CH_4$ levels increased to 1834 ppb by 2015 and further to 1879 ppb by 2020. The renewed increase has been primarily attributed to biogenic sources, particularly wetlands, rather than fossil fuel emissions or changes in atmospheric $CH_4$ sinks (Basu et al., 2022; Lan et al., 2021; Nisbet et al., 2016, 2019). The cause of the 1999-2006 plateau, however, remains unclear, largely due to limited observational data during that period (Nisbet et al., 2019; Nisbet et al., 2023).



Arctic-Boreal regions are characterised by extensive wetlands, permafrost, and organic-rich soils that act as both sources and sinks of $CH_4$ (Saunois et al., 2020). Warming in boreal zones has recently been observed to occur three to four times faster than the global average (Rantanen et al., 2022; IPCC 2023), fuelling significant concerns given the positive feedback between $CH_4$ emissions and climate warming (Yvon-Durocher et al., 2014; Chang et al., 2021; Zhang et al., 2017; Canadell et al., 2021). Undisturbed and disturbed boreal and temperate peatlands emit 30 and 0.1 $TgCH_4$ year$^{-1}$, respectively (Olsson et al.,

2019), and the northern permafrost region emits 15-38 $TgCH_4$ for the 2000-2020 period (Hugelius et al., 2024). While these estimates highlight the significance of boreal $CH_4$ emissions at a global scale, considerable uncertainty remains regarding their regional responses to long-term environmental changes, resulting in high uncertainty in their estimated contribution to the overall $CH_4$ budget (Saunois et al, 2020; Saunois et al., 2016; Kuhn et al., 2021). This knowledge gap is primarily due to the boreal forest being sparsely monitored by long-term atmospheric and ecosystem observatories, highlighting the need to

leverage existing datasets in this region to refine $CH_4$ budgets with better large-scale spatial coverage and fine-scale temporal precision, such as the ZOtino Tall Tower Observatory (ZOTTO).

The ZOTTO facility, situated in central Siberia, was established in 2006, and from April 2009 to February 2022, it had been continuously measuring $CH_4$ at multiple heights up to 301 meters along with other atmospheric gases and their isotopic compositions as well as meteorological data (Winderlich et al., 2010, Tran et al., 2024). This long-term monitoring effort

makes ZOTTO a valuable atmospheric research station, providing high-time-resolution (half-minute frequency) $CH_4$ measurements in a key high-latitude (above 50°N) region. The ability to collect continuous data over extended periods enables the investigation of $CH_4$ dynamics across a range of temporal scales from daily fluctuations to seasonal and interannual trends. This broad temporal scope provides complementary information to understand $CH_4$ behaviour, encompassing both surface and atmospheric processes. Surface processes, such as emissions from wetlands, agriculture, and fossil fuels, dictate the amount

of $CH_4$ released into the atmosphere (Metya et al., 2021), while atmospheric processes, including vertical mixing, horizontal transport, and chemical removal, control its distribution, dilution, and removal over time (Houghton, 2007). At the diurnal scale, boundary layer dynamics significantly influence local $CH_4$ mole fractions, while larger atmospheric patterns shape seasonal and annual variations (Metya et al., 2021, Dowd et al., 2023). Neglecting to account for both surface and atmospheric processes (Faassen et al., 2024) could lead to incomplete or inaccurate interpretations of $CH_4$ trends. As boreal regions

experience rapid warming, the ability to analyse $CH_4$ variations at different time scales using ZOTTO data offers valuable insights into how $CH_4$ emissions are responding to environmental changes, thereby enhancing the precision of $CH_4$ budget estimates in this critical region.

This study investigates long-term $CH_4$ variability at ZOTTO from 2010 to 2021 across interannual, seasonal, and diurnal scales. Specifically, we analyse changes in $CH_4$ diurnal amplitude, defined as the difference between daily maximum and

minimum mole fractions, to evaluate the long-term variations in the diurnal cycle. Additionally, we aim to disentangle the contributions of various drivers to better understand the factors shaping long-term changes in $CH_4$ diurnal amplitude.

The remainder of this study is structured as follows: Sect. 2 provides an overview of the ZOTTO site, including its $CH_4$ and meteorological measurements, and details of the dataset and fundamental concepts employed to examine long-term trends, as





well as seasonal and diurnal variations in CH$_4$. Sections 3 and 4 present the results and discuss the findings, while Sect. 5

summarises the conclusions and acknowledges the limitations of this work.

## 2 Methods

### 2.1 ZOTTO Site Description

The 304 m tall ZOTTO tower is located at 60°48′ N, 89°21′ E (114 m a.s.l.), approximately 20 km west of the village of Zotino at the Yenisei River. The surrounding area is characterized by gentle hills of 60-130 m a.s.l. covered with light taiga forests

(*Pinus sylvestris* dominated) on lichen-covered sandy soils (Schulze et al., 2002), interspersed by numerous waterlogged old river meanders and bogs. The approximate tree height around the tall tower is 20 m. The nearest airport is 90 km north in Bor (2600 inhabitants), the closest cities are Yeniseysk and Lesosibirsk (20 000 and 61 000 inhabitants) to the south-southeast, more than 300 km away, and Krasnoyarsk (1 million inhabitants) about 600 km south of ZOTTO (Heimann et al., 2014; Kozlova et al., 2008).

The climatic conditions at ZOTTO are characterized by a mean annual temperature of 3.8°C measuring at 52 m and total annual rainfall of 536 mm measured at the station. There had been a consistent upward trend in temperatures during summer over the 2010-2021 period (Appendix A – Fig. A3). Rainfall was lowest in winter, and peaks in July and August (Appendix A – Fig. A1.b). During the 2010-2021 period, climatic anomalies include summer 2012 and 2016, characterised by warmer-than-average temperatures and with reduced precipitation (Appendix A – Fig. A1). Wind patterns at ZOTTO are predominantly

south-westerly and westerly reflecting the prevailing regional dynamics (Appendix A – Figure A2). Winter at ZOTTO in Siberia is also characterised by the presence of the Siberia High (Winderlich, 2012), a persistent high-pressure system that leads to strong temperature inversions, low wind speeds, and limited vertical mixing during the winter in the artic regions (Serreze et al., 1992).

### 2.2 Data Description

#### 2.2.1 Methane Mole Fraction Observations

Continuous monitoring of atmospheric CH$_4$ has been conducted at the tall ZOTTO tower since April 2009 (Winderlich et al., 2010). Air is sampled from six inlets located at heights of 4, 52, 92, 156, 227, and 301 m above ground level. CH$_4$ mole fractions at these heights are measured using an EnviroSense 3000i gas analyser (Picarro Inc., USA) employing the Cavity Ring-Down Spectroscopy (CRDS) technique.

Data were recorded every 30 seconds from each sampling line. Each of the six tower levels was sampled for 3 minutes, discarding the first 1.5 minutes for stability. Measurements were taken sequentially in an 18-minute cycle from the top level. Since only a single gas analyser was available, each sampling line was connected to an 8-liter buffer sphere for continuous, synchronized sampling of close-to-the-same air mass at all heights (Winderlich et al., 2010). While one line was analysed, the



others were continuously flushed at 150 sccm under 700 mbar. The buffer system integrated data over 37 minutes to smooth
short-term fluctuations, ensuring stable, well-mixed air samples for reliable measurements.

Calibration is achieved using four horizontally stored aluminium tanks. The $CH_4$ mole fraction in the gas tanks that are currently used at ZOTTO (See also Tran, et al. (2024) – Supplement, Table A1) were determined in the GasLab of the MPI-BGC Jena and are traceable to scales of the World Meteorological Organization (WMO) maintained in NOAA/ESRL (WMO X2004A for $CH_4$; Dlugokencky et al., 2005). To monitor measurement accuracy, a target tank is sampled every 200 hours for
8 minutes, interspersed randomly between two calibration cycles. These target measurements are processed in the same manner as ambient air data. Following the calibration procedure described in Winderlich et al. (2010), the $CH_4$ mole fraction in the target tank was stable at $1946.5 \pm 0.2$ ppb for the entire period (Appendix B – Fig. B1). A comparison with measurements of the same tank from the Jena GasLab ($1946.4 \pm 1.4$ ppb) revealed a statistically insignificant bias and no discernible long-term stability issue in the measurements. This consistency confirms the reliability of the mole fraction measurements, ensuring their
suitability for further analysis.

We also compared our analysis at ZOTTO site from 2010 to 2022 to the Marine Boundary Layer (MBL) product at 60° N (NOAA GML, 2025). The MBL reference dataset represents well-mixed atmospheric conditions at the same latitude as ZOTTO and is representative for locations situated far from significant anthropogenic and natural sources and sinks.

### 2.2.2 Meteorological Data

The meteorological measurement system at ZOTTO has been operational since 2007, with its design and functionality described in detail by Winderlich et al. (2010). This section highlights the features relevant to this study.

Wind measurements (in m s$^{-1}$) are conducted using six 3D sonic anemometers, with 10 Hz frequency, mounted at the same 6 heights as the mole fraction measurements of $CH_4$ along the tower, and recorded every 30 min, supplemented by air temperature (in °C) and humidity (in %) sensors at all (4, 52, 92, 156, 227, and 301 m a.g.l) levels and pressure (in hPa)
transducers at three heights (4, 92, and 301 m). Pressure values for levels without direct measurements are linearly interpolated. Sensible heat flux (in W m$^{-2}$) is measured with eddy covariance system at all heights. Additional calculated variables include potential temperature (in K), specific humidity (in g kg$^{-1}$), and vapor pressure deficit (VPD, in kPa), providing a detailed vertical profile of atmospheric conditions.

Vertical profiles of soil temperature (°C) (measured at depths of 2, 4, 6, 8, 16, 32, 64, and 128 cm), soil moisture (%) (at 8, 16,
32, 64, and 128 cm), and precipitation are also recorded. These measurements are taken approximately 100 m southeast of the tower at a site within a densely wooded area characterized by sandy soil covered with lichens, representative of typical forest conditions in the region. For this study, we focus on measurements at 32 cm below the ground, which is the depth capturing soil conditions that are relevant to nutrient cycling and microbial processes (Schimel, 1995), both of which influence $CH_4$ exchange.

To characterise the large-scale atmospheric circulation patterns as well as the atmospheric conditions above the tower height at ZOTTO, we use ERA5 reanalysis data (Hersbach et al., 2023) for variables that are not directly measured at this site, namely



boundary layer height (m), and horizontal divergence of the wind velocity (s⁻¹), from the ERA5 grid point closest to the ZOTTO coordinates, located at 60°75′ N, 89°25′ E.

## 2.3 Methane Long-term Trends and Seasonal Signal Processing at ZOTTO

For long-term analyses, we used daytime-averaged $CH_4$ mole fraction measurements (13:00-17:00 LT) collected at a height of 301 m from the ZOTTO tall tower. This time window was selected to ensure sampling of well-mixed boundary layer air, making the data suitable for investigating long-term trends and seasonal variability.

$CH_4$ mole fractions at continental sites, particularly at locations like ZOTTO, where multiple local sources and sinks influence $CH_4$ levels, often exhibit an asymmetric annual distribution, characterised by large positive outliers associated with episodic

local and regional emission events. To derive a representative background long-term trend, we first aggregated the daytime-averaged $CH_4$ data into monthly bins and selected the lowest 10-30% of values within each bin. This filtering approach minimises the impact of extreme events, such as the intense wildfire season of 2012, that could otherwise bias estimates of the annual growth rate. The filtered ZOTTO background daytime $CH_4$ dataset ($ZOT_{bg}$) was then processed using the CCGCRV curve-fitting method (Thoning et al., 1989). We employed the Python implementation of CCGCRV, available as a standalone

tool from the NOAA CMDL FTP server (https://gml.noaa.gov/aftp/user/thoning/ccgcrv/; last accessed 10 Jan 2025). The curve-fitting configuration included three polynomial terms to capture the long-term trend and four harmonics to represent the seasonal cycle. Cut-off frequencies were set to the default values of 667 days for the long-term component and 80 days for short-term variability. Any data points lying outside 3 times the normalised root mean square deviation from the CCGCRV-derived smoothed curve were iteratively removed (Kozlova et al., 2008). This process eliminated 0.6% of the $ZOT_{bg}$ data,

ensuring a more accurate representation of the ZOTTO $CH_4$ long-term trend.

For seasonal cycle analysis, we first detrended the $ZOT_{bg}$ dataset by subtracting it from the derived long-term trend. Seasonal amplitude was then calculated as the difference between the maximum and minimum of the monthly medians of the detrended $ZOT_{bg}$. This seasonal amplitude is used in this study as a diagnostic metric to investigate changes in $CH_4$ seasonality at ZOTTO.

## 2.4 Methane Diurnal Signal Conceptual Framework

A common metric used to analyse shifts in the diurnal cycle of a tracer is its diurnal amplitude (Yi et al., 2004; Kretschmer et al., 2014; Bonan et al., 2024). The diurnal amplitude of $CH_4$ is typically defined as the difference between its maximum nighttime mole fraction ($CH_{4,max}$) and minimum daytime well-mixed mole fraction ($CH_{4,min}$). The diurnal cycle of atmospheric $CH_4$ is driven by surface $CH_4$ sources and sinks, and is modulated by atmospheric boundary layer dynamics, which include enhanced daytime mixing and nighttime stability (Fig. 1). Therefore, the diurnal variability of $CH_4$ mole fraction results from

a combined effect of surface fluxes and atmospheric processes.



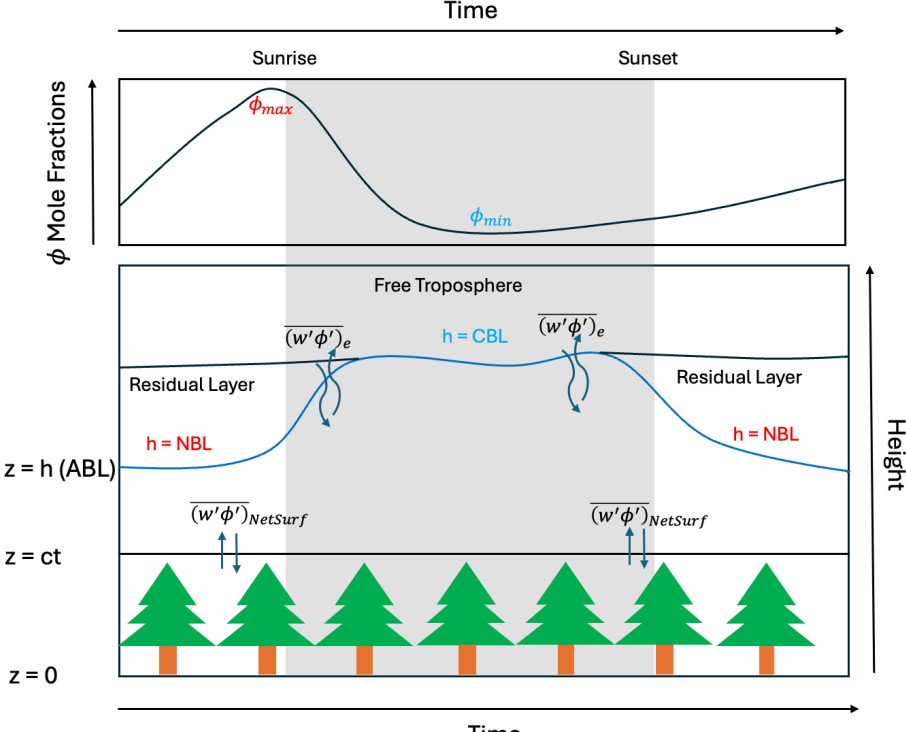

**Figure 1. Schematic overview of diurnal cycle of the mole fractions of atmospheric CH₄ from the top of a forest canopy (z = ct) to the top of the Atmospheric Boundary Layer (ABL) (z = h) illustrating Eq. (1), adapted from Faassen et al. (2024). The figure illustrates the ABL is characterised by a Convective Boundary Layer (CBL) during daytime and a Nocturnal Boundary Layer (NBL)**
**160  formation during nighttime. The "Net Surface flux of CH₄" term ($\overline{(w'\phi')}_{NetSurf}$) refers to the fluxes from the vegetation layer, up to the top of the canopy (z = ct). The fluxes up to this level depend on terrestrial processes, which contribute to the CH₄ mole fractions observed above the top of the canopy. Entrainment flux at the top of the ABL ($\overline{(w'\phi')}_e$) represents the mixing of CH₄ air from above the ABL, to inside the ABL. The horizontal advection of CH₄ (adv($\phi$)), and the chemical reaction term ($S_\phi$) in Eq. (1) are not included in this figure since they are not currently accounted for in this study.**

To integrate all the surface and boundary layer dynamics governing the diurnal variations of CH₄ mole fractions, we make use

of a time-dependent equation (Eq. 1) inspired on the mixed-layer equations as in Vilà-Guerau de Arellano et al. (2015). Figure

1 and 2 summarises the surface and atmospheric drivers of CH₄ diurnal mole fractions and hence, its diurnal amplitude.

$$\frac{1}{h-ct}\int_{z=ct}^{z=h}\frac{\partial\phi}{\partial t}(z)dz = \underbrace{\frac{1}{h-ct}}_{I}\times\left(\underbrace{\overline{(w'\phi')}_{NetSurf}}_{II} - \underbrace{\overline{(w'\phi')}_{e}}_{III}\right) - \underbrace{adv(\phi)}_{IV} + \underbrace{S_\phi}_{V} \tag{1}$$

In this context, the ($\phi$) refer to CH₄ mole fraction, the overbars ($\overline{\phi}$) refer to 30min time averaged, with the prime ($'$) representing

170   the deviation of the mole fraction from the average over time. Similarly, $w'$ represents the deviations from the mean of the

vertical wind speed, $\overline{w}$. All symbols and their corresponding units in this study are provided in Appendix F – Table F1.

Equation (1) shows that rate of change in the column-integrated CH₄ mole fraction (left hand side) (i.e. from the top of the

canopy (ct) to the top of the atmospheric boundary layer (ABL) (h)) (Fig. 1) depends on:





I. The thickness of the ABL ($h - ct$) (Fig. 1): The height of the ABL ($z = h$) exhibits a pronounced diurnal cycle. During the day, the ABL is referred to as the Convective Boundary Layer (CBL), while at night it is termed as the Nocturnal Boundary Layer (NBL) in this study. The thickness of this layer affects the dilution of $CH_4$ during the day and the accumulation of $CH_4$ at night (See Sect. C1.1 in Appendix C).

II. The Net Surface flux of $CH_4$ ($\overline{(w'\phi')}_{NetSurf}$) represents the balance between sources and sinks at the top of the canopy ($z = ct$) (Fig. 1): This flux captures small-scale processes within the canopy, such as $CH_4$ exchange from vegetation and soil, contributing to the $CH_4$ mole fractions in the ABL.

III. Entrainment flux at the top of the ABL ($z = h$) $\overline{(w'\phi')}_e$ (Fig. 1): This flux represents the mixing of $CH_4$ air from above the ABL, either the residual layer during the night or free troposphere during the day, to inside the ABL (See Sect. C1.2 in Appendix C).

IV. The horizontal advection of $CH_4$ ($adv(\phi)$), representing the meso- and long-range horizontal transport of $CH_4$ which is currently not accounted for this study as mentioned in Winderlich et al., 2014.

V. The combination of production and loss of $CH_4$ from chemical reactions with OH ($S_\phi$), which assumed to be negligible within the diurnal scale due to the slow reaction rate of $CH_4$ with OH compared to the atmospheric residence time of OH (Patra et al., 2009).

Terms I and III in Eq. (1) describe key atmospheric boundary layer dynamics that influence $CH_4$ variations from the canopy top to the ABL, primarily driven by ABL height and entrainment flux at the ABL top. A more detailed derivation of Eq. (1), along with explanations of each atmospheric driver terms in Fig. 2 (yellow-coloured boxes), is provided in Appendix C1. The key concepts driving the diurnal amplitude of $CH_4$ in this study (Eq. 1, excluding Terms IV and V) are assumed to apply under high-pressure atmospheric circulation systems. High-pressure systems are generally associated with subsidence and stable, calm weather, which limits horizontal advection. This assumption is important, as horizontal advection (Term IV in Eq. (1)) is not explicitly included in our analysis.

## 2.5 Methane Diurnal Signal Processing at ZOTTO

To calculate the diurnal amplitude of $CH_4$ at ZOTTO, we used hourly $CH_4$ mole fraction measurements at 52 m, which is the closed available measurement height above the forest canopy. Measurements at this level are well-suited to capturing short-term diurnal variations, where surface-atmosphere exchange processes are most active and pronounced.

To investigate the drivers behind shifts in $CH_4$ diurnal amplitude over the study period at ZOTTO, our analysis focused exclusively on days influenced by high-pressure systems. High-pressure conditions were identified using ERA5 geopotential height data at 550 hPa (Hersbach et al., 2023), selecting periods where geopotential height exceeded the 90th percentile of the distribution of each season following the approach of Marín et al. (2022).



### 2.5.1 Atmospheric Processes

2.5.1.1 Atmospheric Boundary Layer Height

The ABL is distinguished between the daytime CBL and the nighttime NBL (Fig. 1 and 2).

To identify the CBL height, we used ERA5 reanalysis data (Hersbach et al., 2023). Daily values were averaged between 12:00 and 15:00 LT to capture the peak convective period while avoiding the sunrise and sunset transitional periods. By examining interannual variations in the summer CBL height, we assess whether daytime dilution effects on $CH_4$ mole fractions have strengthened or weakened over time.

To determine the NBL height, we utilised the in-situ vertical potential temperature ($\theta$) observations from our 300-meter measurement tower, as ERA5 reanalysis data have been observed to overestimated nighttime conditions (Sinclair et al., 2022). We applied a least-squares regression fit to the nighttime vertical $\theta$ gradients using the following equation (Oncley et al., 1996; Frenzen and Vogel, 2001; Johansson et al., 2001):

$$X(S) = A + B \times \ln(S) + C \times \ln(S)^2 \tag{2}$$

Where $X(S)$ is the fitted function of the variable S, which in this case represents $\theta$. For nighttime data at ZOTTO, this fitting was applied to the vertical profile from 4 heights above the boreal canopy (52, 92, 156 and 227 m) (red line in the Appendix D – Fig. D1). Data from 4 m (within the canopy) and 301 m (residing in the residual layer during nighttime (See explanation in Appendix A – Fig. A4)) were excluded. Fits with an $R^2$ value greater than 0.7 were retained. The first derivative of the fitted curve $X(S)$, representing the temperature lapse rate ($\frac{\partial\theta}{\partial z}$), was computed and normalised ($X_{norm}$) as in Eq. (3):

$$X_{norm} = \frac{X'(S)}{X_{max}} \tag{3}$$

Where $X'(S)$ is the first derivative of $X(S)$ and $X_{max}$ is the maximum value of $X'(S)$. The NBL height was identified as the altitude where the $X_{norm}$ curve (blue line Appendix D – Fig. D1) decayed to its e-folding value ($\sim 1/e$ of the maximum) (Stull, 1988). We analysed temporal variations in these derived NBL heights, restricting the dataset to timestamps between 00:00-04:00 LT, period when the vertical $\theta$ profile at ZOTTO is the most pronounced.

Since both the vertical profiles of $\theta$ and $CH_4$ provide insight into the structure and evolution of the nighttime atmospheric column, with the $CH_4$ profile typically mirroring the temperature profile in the opposite direction (Appendix A – Fig. A4 and Fig. 3), we applied the same regression methodology to the $CH_4$ vertical profiles (Eq. (2) and (3)) to compare the trends in NBL height derived from both variables over time. The NBL heights derived from $CH_4$ and $\theta$ vertical profiles exhibit similarities (See Appendix D), further validating the reliability of the least-squares regression fit approach.

By examining interannual variations in the summer NBL height, we can assess whether the nighttime stability leading to accumulation of near-surface $CH_4$ mole fractions have strengthened or weakened over time.

2.5.1.2 Entrainment Effect

As entrainment is negligible during the nighttime (See Sect. C1.2 in Appendix C), we focus on quantifying the daytime entrainment rate over the 2010-2021 period. We examined trends in CBL growth ($\frac{\partial h}{\partial t}$), subsidence velocity ($w_{sub(h)}$), and $CH_4$



mole fraction differences between the CBL and its overlaying layer – the free troposphere (FT) ($\Delta_{\phi(ft-cbl)}$). As shown in Fig. 2, these factors collectively contribute to entrainment strength. Given that $\Delta_{\phi(ft-cbl)}$ is influenced by the stability of the previous night (Fig. 2 and See Sect. C1.2.3 in Appendix C), which is already assessed as NBL height as in Sect. 2.5.1.1, we primarily focused on CBL growth and subsidence velocity to assess the daytime entrainment effects.

We used positive sensible heat flux, a proxy for buoyancy turbulence, to assess whether mixing strength and CBL growth rate have changed over time. Higher values indicate increased turbulence, enhancing CBL growth and entrainment. Sensible heat flux data from 52 m at ZOTTO (where flux measurements peak just above the canopy) were compared with ERA5 surface sensible heat flux values to determine temporal trends in daytime CBL growth.

In this study, we analysed long-term variations in subsidence by examining positive divergence data from ERA5 at the 750 hPa pressure level over the course of 2010-2021 period. The 750 hPa level resides in the mid-troposphere, where large-scale vertical motions are the most prominent (Stepanyuk et al., 2017). This level, therefore, effectively captures the dynamics of vertical air movement and its influence on atmospheric stability.

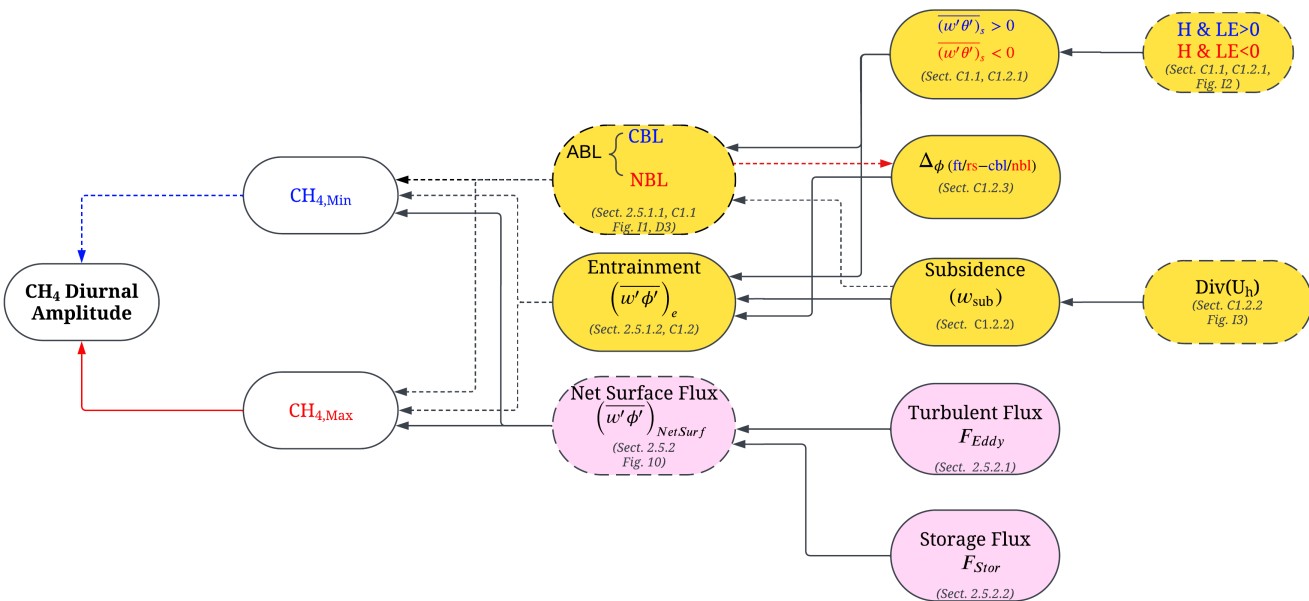

**Figure 2. Chart illustrating the cause-effect relationships governing the diurnal amplitude of CH$_4$.** Solid arrows indicate positive influences, while dashed arrows represent negative influences. Red and blue highlight key nighttime and daytime processes, respectively, while black denotes processes affecting both daytime and nighttime dynamics. Yellow and pink boxes distinguish atmospheric and surface canopy processes, respectively. Grey text within some boxes references sections and figure where further method details and their corresponding results can be found. Boxes with dashed outline are the variables investigated in this study. The figure illustrates the ABL is characterised by a Convective Boundary Layer (CBL) during daytime and a Nocturnal Boundary Layer (NBL) formation during nighttime. The "Net Surface flux of CH$_4$" term ($(\overline{w'\phi'})_{NetSurf}$) refers to the fluxes from the vegetation layer, up to the top of the canopy; integrates both turbulent flux (F$_{Eddy}$) and storage flux (F$_{Stor}$). Entrainment flux at the top of the ABL $(\overline{w'\phi'})_e$ represents the mixing of CH$_4$ air from above the ABL, to inside the ABL. $\Delta_{\phi(ft/rs-cbl/nbl)}$: Change in CH$_4$ mole fraction across layers (free troposphere/residual layer vs. CBL/NBL); $(\overline{w'\theta'})_s > 0/< 0$: Positive/negative surface buoyancy flux, indicating convective/stable boundary layers; H & LE $> 0/< 0$ : Sensible heat (H) and latent heat (LE) fluxes, with signs indicating





net heating or cooling; $w_{sub(h)}$ represent the vertical subsidence velocity at ABL height and $Div(\vec{U_h})$ is the horizontal wind divergence at the ABL height.

### 2.5.2 Surface Processes

This section gives an overview of how to assess net $CH_4$ surface fluxes $(\overline{(w'\phi')}_{NetSurf})$ at the top of the canopy (z = ct) (Fig. 1 and pink-coloured boxes in Fig. 2) using the vertical $CH_4$ profile at ZOTTO. This can be calculated using the following terms in Eq. (4) adapting from Finnigan, 1999; Yi et al., 2000; and Feigenwinter et al., 2004:

$$\overline{(w'\phi')}_{NetSurf} = \underbrace{\overline{(w'\phi')}_{ct}}_{F_{Eddy}} + \underbrace{\int_{z=0}^{z=ct} \frac{\partial \overline{\phi}}{\partial t} dz}_{F_{Stor}} \tag{4}$$

The sign convention used here gives positive $\overline{(w'\phi')}_{NetSurf}$ for ecosystem emissions, where a positive flux term (i.e., source) corresponds to transport out of the control volume (Feigenwinter et al., 2004). In Eq. (4), the first term represents the turbulent vertical flux ($F_{Eddy}$) measured at the top of the canopy (ct), while the second term represents the storage of $CH_4$ ($F_{Stor}$), which is the temporal dynamics of $CH_4$ in the air column below the $F_{Eddy}$ measurement height, not influenced by turbulence, as calculated by the integral. Each of these two terms is discussed in detail in the subsequent subsections.

#### 2.5.2.1 Inferred Turbulent Vertical Flux

Direct eddy flux measurements of $CH_4$ are not feasible at ZOTTO tall tower, due to the low measurement frequency of the $CH_4$ analyser (0.2 Hz), long tubing lengths (up to 320 m), and the use of buffer volumes with extended mixing times (~40 minutes, corresponding to ~0.0004 Hz). Instead, we apply the modified Bowen ratio approach (Businger, 1986, Winderlich et al., 2014), in which the turbulent vertical flux at the top of the canopy ($F_{Eddy}$) in Eq. (4) can be written as:

$$F_{Eddy} = \overline{(w'\phi')}_{ct} \cong \frac{H_{(ct)}}{\rho_{(ct)} \cdot C_p} \frac{\partial \phi / \partial z}{\partial \theta / \partial z} \tag{5}$$

Where $H_{(ct)}$ and $\rho_{(ct)}$ are the sensible heat flux and density at the top of the canopy. Since direct measurements at the exact canopy height at ZOTTO (~28 m) are unavailable, we use the sensible heat flux measured using eddy-covariance system and the density at 52 m, which is the closest available measurement height above the canopy. $C_p$ is specific heat capacity constant ($C_p = 1.00467$ J $g^{-1}$ $K^{-1}$). The mole fraction and temperature gradients between two adjacent heights (52 m and 92 m) are used to compute the turbulent fluxes at the intermediate level 72 m. This derived $F_{Eddy}$ at 72 m represents the turbulent flux at the top of the canopy.

#### 2.5.2.2 Storage Flux

The storage term $F_{Stor}$ in Eq. (4) represents the temporal changes in column-integrated $CH_4$ mole fraction below 72 m, the height at which $F_{Eddy}$ is calculated in Sect. 2.5.2.1. For illustration, the diurnal development of the $CH_4$ profile along the tower is given in Fig. 3. The $F_{Stor}$ can be visualised through the shaded trapezoidal areas between the half-hourly time steps $t_i$ and





$t_{i+1}$, and the two different tower heights below 72 m (i.e., $z_1 = 4$ m and $z_2 = 52$ m) (See Fig. 3, grey-shaded area). The $F_{Stor}$

term in Eq. (4) can be expanded as in Winderlich et al., 2014:

$$F_{Stor}(t_i, z_{ct}) = \int_{z=0}^{z=ct} \frac{\partial \overline{\phi}}{\partial t} dz \cong \frac{1/2((\phi_1(t_{i+1}) - \phi_1(t_i)) + (\phi_2(t_{i+1}) - \phi_2(t_i)))}{t_{i+1} - t_i} \times (z_1 - z_2) \qquad (6)$$

The $\phi_1(t_i)$ and $\phi_2(t_i)$ represent $CH_4$ mole fraction at 4 m and 52 m at the time step $t_i$, respectively. Storage fluxes are

calculated up to 52 m in this study, which is the highest available measurement below the $F_{Eddy}$ estmation at 72 m.

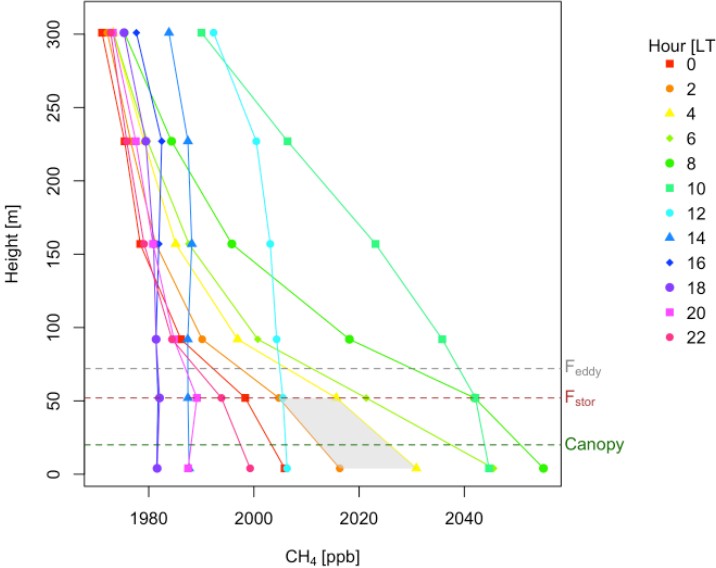

**Figure 3. The climatological summer (JJA) vertical profile of $CH_4$ mole fractions every 2 hours at ZOTTO (2010-2021 average). The shaded area represents the calculated storage flux ($F_{Stor}$) between consecutive time steps and tower heights. The dashed green line denotes the canopy height at ZOTTO, while the dashed brown and grey lines indicate the 52 m and 72 m levels, where the $F_{Stor}$ and $F_{Eddy}$ terms in Eq. (4) are calculated, respectively.**

During high-pressure systems, the downward subsidence velocity can lower the effective height at which the storage flux is

calculated up to. This downward movement of air means that the storage flux calculations need to be adjusted to reflect this

displacement. Equation 6 as in Winderlich et al., 2014 can be expanded to below:

$$F_{Stor}(t_i, z_{ct}) \cong \frac{1/2((\phi_1(t_{i+1}) - \phi_1(t_i)) + (\phi_2(t_{i+1}) - \phi_2(t_i)))}{t_{i+1} - t_i} \times (z_1 - z_2) \times \underbrace{\frac{z_2 - w_{sub(z_2)} \cdot (t_{i+1} - t_i)}{z_2}}_{\text{Correction factor}} \qquad (7)$$

Where $w_{sub(z_2)}$ is the vertical wind component at 52 m. We do not derive this vertical wind component from 3D anemometer

measurements at ZOTTO due to its sensitivity to sensor misalignment, where parts of the horizontal wind components are

inadvertently reallocated into small parts of the vertical component, causing significant errors (Winderlich, 2014). To address

this challenge, horizontal divergence from 3-hourly short-term forecast fields from the operational archive of the European



Centre for Medium-Range Weather Forecasts (ECMWF, http://www.ecmwf.int/) has been used to derive vertical mean wind speed $w_{sub(z_2)}$ at 52 m at ZOTTO.

In summary, the net surface flux at the top of the canopy (in Eq. (4)) will be represented as the sum of $F_{Stor}$ up to 52 m and 310 $F_{Eddy}$ at 72 m in this study. We focus the calculation of this net surface flux on the nighttime period (00:00-04:00 LT). This is because the Bowen ratio method used to estimate the turbulent flux ($F_{Eddy}$ in Eq. (5)), which depends on vertical gradients of $CH_4$ and $\theta$, becomes less reliable during the day. Strong daytime mixing reduces these vertical gradients, introducing significant noise into $F_{Eddy}$ and compromising the accuracy of the total net surface flux. By restricting our analysis to the nighttime period (00:00-04:00 LT), when the vertical gradients are the most pronounced, we ensure more reliable $F_{Eddy}$ and 315 hence net surface flux estimates.

During nighttime at ZOTTO, the $F_{Stor}$ becomes the dominant component of the total net surface flux, contributing approximately 60-80% (Appendix G – Fig. G1), surpassing $F_{Eddy}$. A similar pattern was previously reported for $CO_2$ net surface flux at Missouri Ozark by Yang et al. (2007) and at ZOTTO by Winderlich (2012).

### 2.6 Statistics

To analyse trends in variables over the 2010-2021 period, we applied the Theil-Sen regression method (Theil, 1992 and Sen, 1968), a robust non-parametric approach known for its resistance to outliers. To investigate potential drivers of any observed significant trends, we performed orthogonal regression, which accounts for errors in both dependent and independent variables, providing a more reliable assessment of relationships between variables.

## 3 Results

### 3.1 Long-term Trend and Growth Rate

The $CH_4$ mole fraction recorded at ZOTTO is generally higher than the MBL by around 50 ppb (Fig. 4a). ZOTTO also displays a slightly more pronounced increase in $CH_4$ levels as its trend diverges from MBL over time (Fig. 4b).





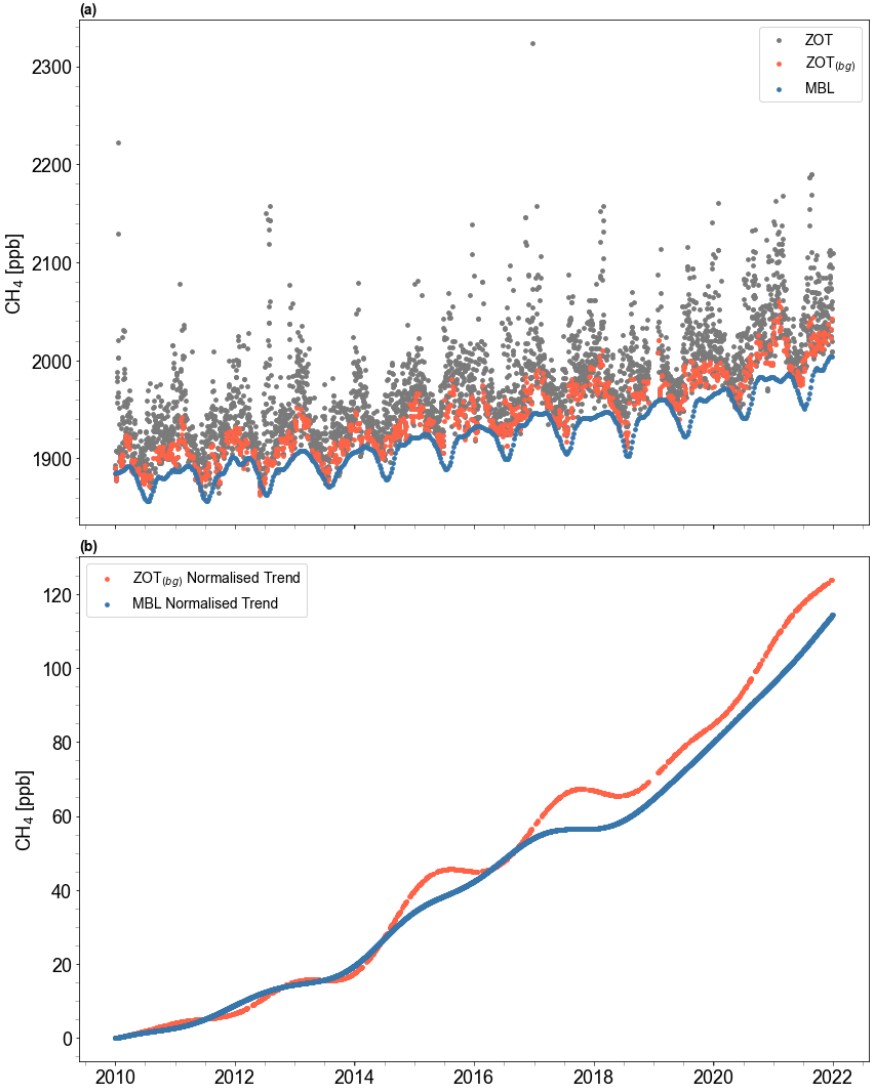

**Figure 4. (a) (a) Daytime CH$_4$ mole fractions at 301 m a.g.l from the ZOTTO tower: 13:00-17:00 LT averaged daytime data (ZOT,**
**grey) and filtered-background daytime data (ZOT$_{bg}$, red), shown alongside bi-weekly CH$_4$ measurements from the marine boundary**
**layer at 60°N (MBL, blue); (b) Long-term CH$_4$ trends of ZOT$_{bg}$ and MBL data derived from the CCGCRV curve-fitting method**
**(Thoning et al., 1989), normalized to their respective 2010 baseline values.**

The average annual CH$_4$ growth rate at ZOTTO from 2010 to 2021 is slightly higher ($9.85 \pm 7.1$ ppb year$^{-1}$) than MBL (9.05
$\pm$ 5.5 ppb year$^{-1}$). The growth rate of ZOTTO also reflects more interannual variability compared to the MBL, indicated by the
larger standard deviations (Table 1) due to local and regional sources. The annual growth rates in Table 1 show that the CH$_4$
growth rate at ZOTTO peaked in 2014 at 21.22 ppb year$^{-1}$, followed by an acceleration in 2019 (14.34 ppb year$^{-1}$) and 2020
(22.22 ppb year$^{-1}$). The MBL growth rates show similar temporal patterns, with notable increases in 2014, 2019, and 2020, but
the magnitude of these increases is not as large as those at ZOTTO.





**Table 1. Annual growth rate of ZOTTO, and MBL in ppb year$^{-1}$ derived from the first derivative of the trendlines in Fig. 4b (without normalising to their respective 2010 baseline values). Years highlighted in red indicate strong growth rate.**


| Year | MBL | ZOT |
|---|---|---|
| **2009** | 3.24 | 4.04 |
| **2010** | 2.77 | 4.10 |
| **2011** | 6.14 | 2.53 |
| **2012** | 5.54 | 8.60 |
| **2013** | 5.14 | 2.34 |
| **2014** | 14.51 | 21.22 |
| **2015** | 8.26 | 5.26 |
| **2016** | 11.74 | 12.22 |
| **2017** | 2.45 | 9.70 |
| **2018** | 8.18 | 3.53 |
| **2019** | 15.15 | 14.34 |
| **2020** | 16.09 | 22.22 |
| **2021** | 18.47 | 17.02 |
| **Mean** | 9.05 ± 5.5 | 9.78 ± 7.1 |

## 3.2 Seasonal Cycle

There is a consistent shape of the seasonal cycle across the two datasets, with higher mole fractions in the colder months (winter) and lower mole fractions in the warmer months (spring and summer) (Fig. 5). At ZOTTO, the seasonal fluctuations of $CH_4$ are more pronounced compared to the MBL data, with several clear peaks during the cold months. While the shape of

the seasonal cycle at ZOTTO shows variability from year to year, it remains relatively constant at MBL across the years. A slight time lag is observed between the seasonal $CH_4$ minima at ZOTTO and in the marine boundary layer (MBL). There is a shift in the seasonal cycle phase between the two datasets, with the MBL phase occurring one or two months later than that of ZOTTO.







**Figure 5. The yearly seasonal cycles of background-filtered daytime $CH_4$ at ZOTTO ($ZOT_{bg}$) and biweekly marine boundary layer $CH_4$ at 60°N (MBL), shown after removing long-term trends (i.e., subtracting the trend components presented in Fig. 4b from the data in Fig. 4a). The line plots with circle markers represent the monthly medians. The darker shaded boxes indicate the interquartile range (IQR). The lighter shaded boxes extend from $Q_1 - 1.5 \times IQR$ to $Q_3 + 1.5 \times IQR$, where $Q_1$ and $Q_3$ are the 25th and 75th percentiles, respectively.**

The seasonal amplitude of $CH_4$ at ZOTTO is consistently larger than that observed in the MBL dataset. The seasonal amplitude is calculated as the difference between the winter maximum and the seasonal minimum median values (in Fig. 5). Although both datasets exhibit increasing trends in seasonal amplitude over the period 2010-2021 (2.12 ppb year$^{-1}$, p = 0.12 and 0.49 ppb year$^{-1}$, p = 0.09 for ZOT and MBL respectively), neither trend is statistically significant (Fig. 6).



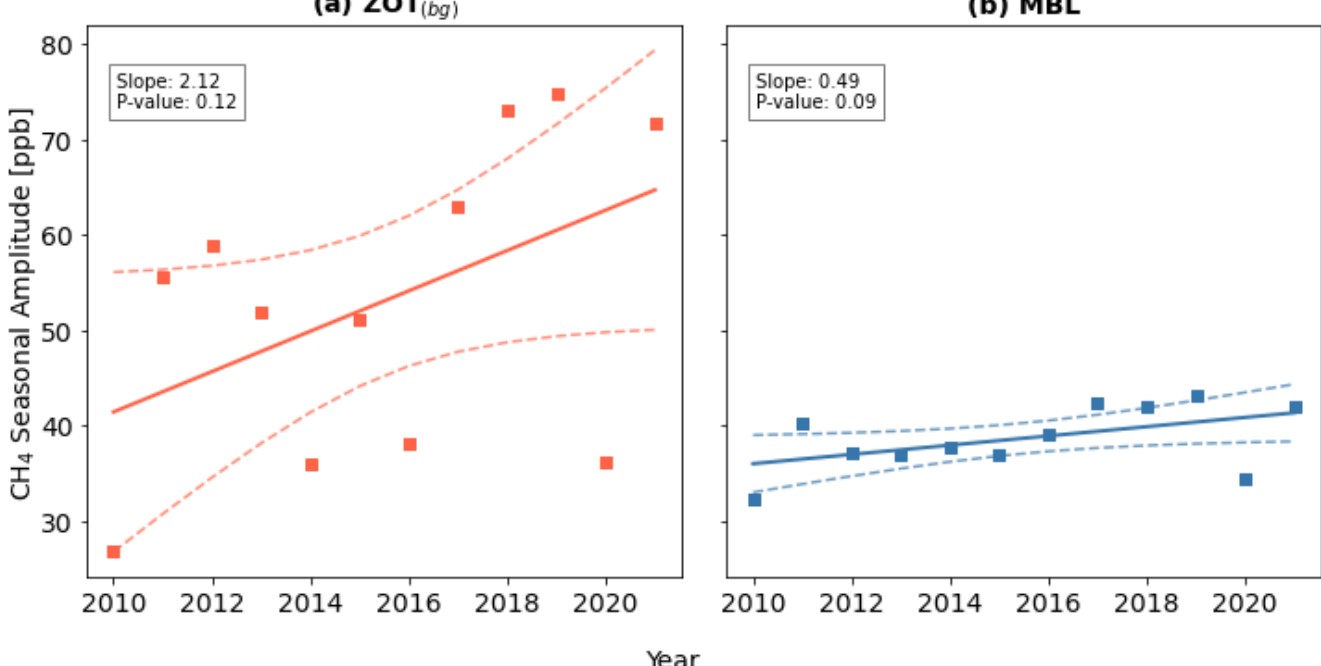

**Figure 6. Time series of the CH₄ seasonal cycle amplitude (square markers) for detrended background-filtered daytime CH₄ at ZOTTO (ZOT_bg) and biweekly marine boundary layer CH₄ at 60°N (MBL). Seasonal amplitude is calculated as the difference between the winter maximum and the seasonal minimum median values derived from Fig. 5. For ZOTTO, the amplitude is defined as the difference between the December-February maximum and the May-July trough; for MBL, it is calculated as the December-February maximum minus the July-August trough. The Theil-Sen regression trend is depicted by the solid line, with the 95%**
**confidence interval of the trend shown as dashed curves. The p-value indicates whether the slope of the regression is significantly different from zero.**

Notably, ZOTTO displays a secondary peak in late summer during the late summer (August) period, which is absent at MBL (Fig. 5). The amplitude of this late summer peak at ZOTTO, calculated as the difference between the late summer (August) maximum and the seasonal minimum (during May-July period) median values (in Fig. 5), shows a significant increasing trend

at 0.05 level (1.35 ppb year$^{-1}$, p = 0.02) along with notable interannual variability (Fig. 7). To further explore the potential factors that might contribute to this observed increase in the late summer peak unique to ZOTTO, we will focus our analysis of variations and trends in diurnal amplitude over the years during summer months (June, July and August) in the next section.



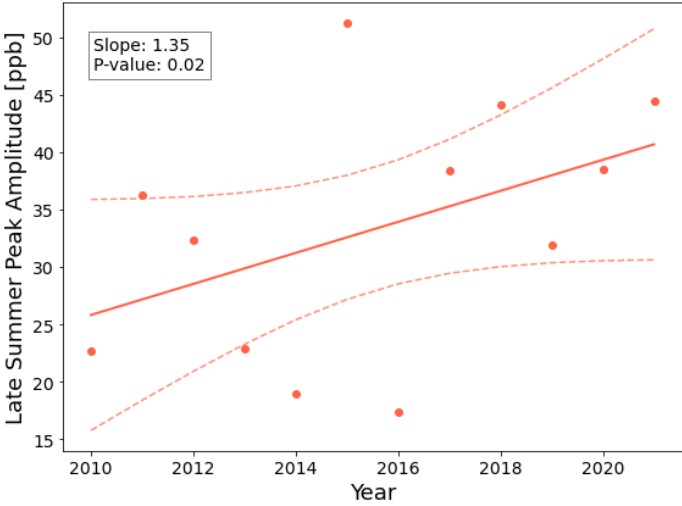

**Figure 7. Time series of the CH$_4$ seasonal late summer peak amplitude (circle markers) for detrended background-filtered daytime CH$_4$ at ZOTTO (ZOT$_{bg}$). The amplitude here is calculated as the difference between the late summer (August) maximum and the seasonal minimum (during May-July) median values in Fig. 5. The Theil-Sen regression trend is depicted by the solid line, with the 95% confidence interval of the trend shown shaded areas. The p-value indicates whether the slope of the regression is significantly different from zero.**

### 3.3 Diurnal Cycle

CH$_4$ measurements in the summer from the ZOTTO site exhibit a distinct diurnal cycle, characterized by peak mole fractions around 06:00 LT, followed by a sharp decline around 07:00 LT (Appendix H – Fig. H3). Lower values persist throughout the day until approximately 18:00 LT. Seasonally, the diurnal cycle is most pronounced during the warmer months, i.e. spring, summer, and autumn, while it remains minimal in winter (Appendix H – Fig. H2).

Between 2010 and 2021, the summer diurnal amplitude increased significantly at p = 0.01 level at a rate of 5.55 ppb year$^{-1}$ (p = 0.002) (Fig. 8a). Both daytime and nighttime CH$_4$ mole fractions significantly increased at the 0.01 level over this period (Fig. 8b, 8c), driven largely by the long-term trend observed in Fig. 4c. However, the increase was more pronounced at night (16.77 ppb year$^{-1}$, p < 0.001) compared to the daytime (11.22 ppb year$^{-1}$, p < 0.001), emphasising the dominant role of nighttime CH$_4$ mole fractions in driving the observed rise in summer diurnal amplitude, which may in turns contributed to the statistically significant rise in the late-summer CH$_4$ peak observed at ZOTTO (Fig. 7). When the long-term trend is removed, the influence of nighttime CH$_4$ becomes even more evident (Appendix H – Fig. H1). After detrending, the diurnal amplitude continues to show the same significant increase, which is driven solely by the rise in nighttime CH$_4$ mole fractions (6.03 ppb yr$^{-1}$, p = 0.02), while daytime mole fractions show no significant trend. This further suggesting that the increase of the summer diurnal amplitude is primarily due to increased nighttime CH$_4$ mole fractions.




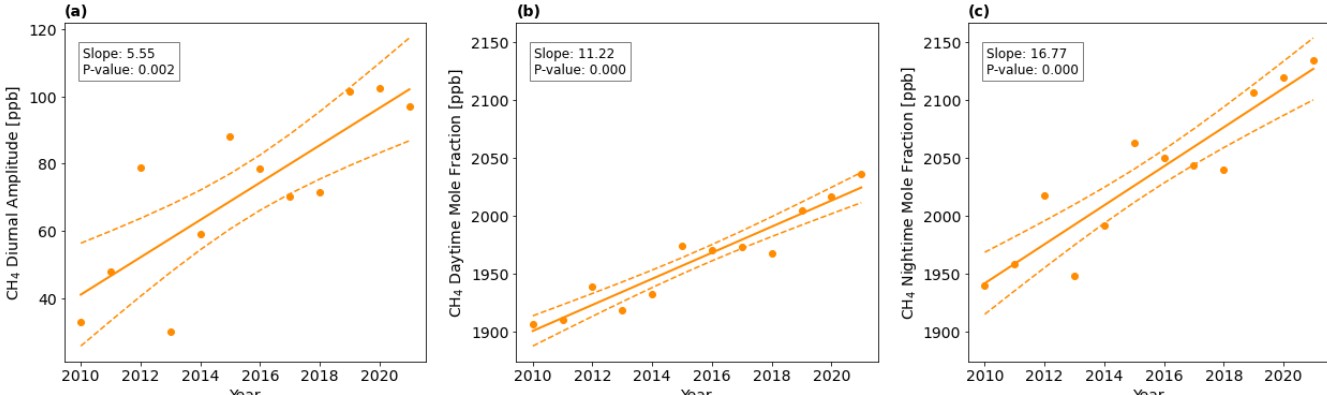

**Figure 8. Time series of yearly summer (JJA): (a) averaged CH₄ diurnal cycle amplitude; (b) its daytime (10:00-16:00 LT averaged) CH₄ mole fraction, and (c) its nighttime (00:00-04:00 LT averaged) CH₄ mole fraction (right) (circle markers) at ZOTTO using 52 m a.g.l. data. The Theil-Sen regression trend is depicted by the solid line, with the 95% confidence interval of the trend shown as a dashed line. The p-value indicates whether the slope of the regression is significantly different from zero.**

To better understand the drivers behind the observed increase trend in the summer CH₄ diurnal amplitude, we focused our analysis on summer days occurring under high-pressure conditions, when the fundamental concepts underlying the potential drivers of CH₄ diurnal amplitude (Presented in Fig. 2) are assumed to hold (See Sect. 2.4). Appendix E – Table E1 summarizes the number of high-pressure days identified for each summer month over the 11-year period (2010-2021), based on the filtering criteria described in Sect. 2.5. The following analysis is restricted to these high-pressure cases to ensure consistency in atmospheric conditions.

Our results indicate that increasing nighttime CH₄ surface fluxes during summer are the primary driver of the observed rise in nighttime CH₄ mole fractions and the associated increase in summer diurnal amplitude from 2010 to 2021, rather than changes in the NBL dynamics. Orthogonal regression analysis confirms a significant positive relationship between the diurnal amplitude and nighttime net CH₄ surface flux ($R^2 = 0.67$, $p < 0.001$), while other potential atmospheric dynamic drivers (dashed-outline boxes in Fig. 2) show no such correlation (Fig. 9). At ZOTTO, the inferred nighttime net surface CH₄ fluxes during summer range from -0.05 to 0.2 ppb m s⁻¹, with the highest values occurring in August (Fig. 10), coinciding with the observed late-summer CH₄ peak (Fig. 5). All summer months show an increasing trend in nighttime CH₄ flux, with a statistically significant rise in August at 0.05 level ($p = 0.016$), consistent with the significant increase in the amplitude of the late-summer CH₄ peak at ZOTTO (Fig. 7). No significant trends are observed in the other atmospheric drivers of the diurnal amplitude (dashed-outline yellow boxes in Fig. 2). A detailed analysis of the interannual variations in these atmospheric drivers over the 2010-2021 period is provided in in Appendix I.




**Figure 9. Relationship between monthly summer CH₄ diurnal amplitude and its potential drivers (dashed-outline boxes in Fig. 2): CBL height, NBL height, divergence at 750 hPa, cumulative sensible heat flux at 52 m a.g.l. and nighttime net CH₄ surface flux. Shaded areas represent 95% confidence intervals.**





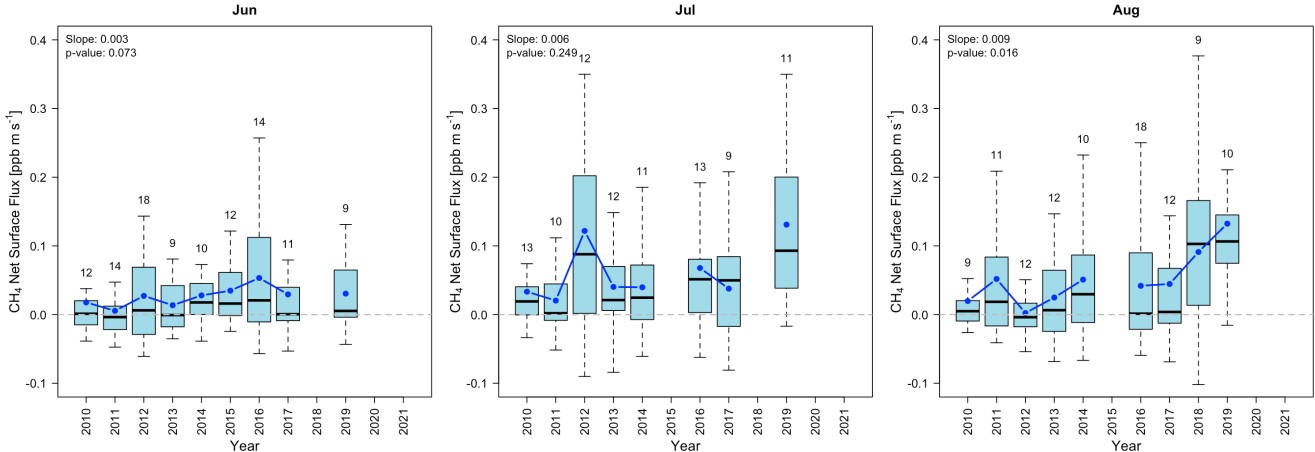

420

**Figure 10. Box and whisker plot of yearly nighttime (00:00-04:00 LT) Net CH₄ flux for each summer month. The box denotes the interquartile range (IQR), showing the median with a thick black line. The whiskers range from Q₁ - 1.5 × IQR to Q₃ + 1.5 × IQR, where Q₁ and Q₃ are the 25th and 75th percentiles, respectively. The blue line is the monthly mean. The Theil-Sen trend slope for the mean and its p-value are denoted on the top left corner of each plot. Numbers above each box indicate the sample size or the number**

425 **of available days for analysis in that month.**

Given a clear increasing trend observed in the nighttime net CH₄ surface flux, we further investigate the potential drivers of the increase in this surface flux. The relationship between nighttime net CH₄ surface flux and various environmental variables, including soil moisture, soil temperature, precipitation, air temperature, and VPD (Fig. 11) indicates a strong positive correlation between CH₄ flux and soil temperature ($R^2 = 0.7$, $p < 0.001$), soil moisture at 32 cm below ground ($R^2 = 0.60$, p =

430 0.031), and air temperature measured at 52 m a.g.l ($R^2 = 0.65$, $p = 0.042$). This result suggests that warmer air temperature as well as warmer and wetter soil conditions are associated with increased CH₄ emissions. Precipitation, however, shows a weak correlation with CH₄ flux ($R^2 = 0.43$, $p = 0.12$), implying minimal influence. VPD exhibits an insignificant negative relationship with CH₄ flux ($R^2 = 0.62$, $p = 0.051$).





**Figure 11. Relationship between monthly nighttime net CH₄ Flux and precipitation, soil temperature at the depth of 32 cm, soil moisture at the depth of 32 cm, air temperature measured at 52 m a.g.l. and Vapour Pressure Deficit (VPD) during summer. Shaded areas represent 95% confidence intervals. The colour gradient in the data points indicates temporal trends, with more recent years (darker blue) tending toward higher temperatures, lower soil moisture, and increased VPD.**

## 4 Discussion

### 4.1 Long-term Trend and Growth Rate

The observed persistent increase in CH₄ mole fractions at ZOTTO aligns with the global trend reported by NOAA and other long-term monitoring networks (Lan et al., 2021). Notably, episodic peaks in CH₄ growth rates at ZOTTO, particularly in 2014 and after 2018, are consistent with global trends of accelerating CH₄ levels, which have been picked up in 2013, steepening after 2018 and further acceleration in 2020 (Worden et al., 2017; Nisbet et al., 2019). These findings underscore the importance of continuing monitoring to better understand the changing dynamics of CH₄ emissions across diverse regions.



A comparison of CH$_4$ mole fractions between the inland tall tower at ZOTTO and MBL reference site reveals consistently higher CH$_4$ concentrations at ZOTTO. During the 2010-2021 period, ZOTTO also shows a slightly stronger long-term upward trend in CH$_4$ compared to the MBL. This suggests a notable influence from regional continental sources, including wetlands, agriculture, and fossil fuel emissions on ZOTTO data.

## 4.2 Seasonal Cycle

ZOTTO exhibits a pronounced seasonal cycle in atmospheric CH$_4$ mole fractions, characterized by maxima during December-January and minima in May-July. This seasonal pattern is consistent with earlier short-term observations at ZOTTO (2009-2012) (Winderlich, 2012). Comparable sinusoidal seasonal cycles have been documented at other monitoring sites, including urban locations such as tall tower Białystok in Poland (Popa et al., 2010), and Guro and Nowon in Seoul, South Korea (Ahmed et al., 2015), as well as remote inland stations such as Bil Fraserdale in Ontario, Canada (Worthy et al., 1998), and Ulaan Uul in Mongolia (Kim et al., 2015).

The seasonal minima in CH$_4$ mole fractions observed at ZOTTO during May-July period is likely driven by a combination of enhanced OH-driven atmospheric oxidation and hydrological constraints on CH$_4$ production. OH radicals typically peak in abundance during late spring to early summer, thereby intensifying CH$_4$ oxidation and contributing to lower atmospheric CH$_4$ levels during this period. Elevated water table elevation (WTE), defined as the depth below which the soil is saturated, may further suppress CH$_4$ emissions during the spring. A high WTE in early spring often reflects substantial snow accumulation from the preceding winter, which acts as an insulating layer that limits soil freezing (Granberg et al., 1999). Under such conditions, methanotrophic communities can remain active throughout the winter (Einola et al., 2007; Trotsenko & Khmelenina, 2005), oxidizing CH$_4$ produced during early thaw and thereby reducing net emissions (Feng et al., 2020). Moreover, because water has a higher heat capacity and latent heat of fusion compared to air, soils under high WTE conditions warm more slowly in spring. This delayed warming postpones the onset of microbial methanogenesis, further limiting CH$_4$ production during the spring-summer seasonal transition (Feng et al., 2020). Collectively, these processes contribute to the pronounced CH$_4$ trough observed at ZOTTO in the May-July timeframe.

Wintertime CH$_4$ peaks at ZOTTO could also be influenced by biomass burning and anthropogenic emissions (Saunois et al., 2020). While fossil fuel activities may contribute to winter CH$_4$ spikes at ZOTTO, their influence is likely limited due to the distance of the station from major oil and gas production sites (>300 km) (Winderlich, 2012). Former studies suggest that this might be too far to notably modify the CH$_4$ mole fraction at ZOTTO. Tohjima et al. (1996) discovered CH$_4$ mole fraction of 2900 ppb in only 150 m altitude above an oil production site, while the signal has already been vented away in 250 m altitude. Moreover, natural gas emissions sum up to 1 to 10 % of the overall wetland emissions only (Tarasova et al., 2009) and no significant increase in CO level has been detected during the winter to suggest substantial burning processes (Kozlova et al., 2008). Elevated winter CH$_4$ levels at ZOTTO are most likely driven by meteorological conditions influenced by the Siberian High, a persistent high-pressure system that leads to strong temperature inversions, low wind speeds, and limited vertical mixing during the winter in the artic regions (Serreze et al., 1992). These conditions trap CH$_4$ near the surface, contributing to



episodic of CH$_4$ enrichment during the winter (Winderlich, 2012). This winter phenomenon has been witnessed on

subcontinental scale in Western Siberia, when CH$_4$ enrichments above 2000 ppb occurred during high pressure situations in combination with temperature inversions, with temperatures below -20°C (Sasakawa et al., 2010).

We observed an increasing, though statistically insignificant, trend in the seasonal amplitude of background CH$_4$ mole fractions at ZOTTO over the 2010-2021 period. At the Waliguan station (WLG), an insignificant upward trend was also reported, where the seasonal amplitude rose from 15.1 ppb to 23.7 ppb between 1994 and 2019 (Liu et al, 2021). In contrast, Dowd et al. (2023)

and Liu et al. (2025) reported a significant decreasing trend in the seasonal CH$_4$ amplitude across high-latitude Northern Hemisphere sites since the 1980s. This discrepancy may be partly attributed to the longer observational periods used in Dowd et al. and Liu et al. (2025) studies. Additionally, their analyses were based on marine or remote background sites, which are less influenced by local emissions. In contrast, ZOTTO (and WLG) is located inland and is subject to regional influences, including variable contributions from wetlands, and fossil fuel sources. These regional emissions may have contributed to

enhanced seasonal amplitude observed at ZOTTO.

A notable and distinguishing feature of the CH$_4$ seasonal cycle at ZOTTO is the presence of a secondary peak in late summer (in August). This distinct secondary CH$_4$ peak observed at ZOTTO is primarily attributed to increased emissions from wetlands located to the west of the station. This pattern is consistent with observations from other boreal wetland systems, including sites in Western Siberia (Sasakawa et al., 2010) and Canadian boreal regions, where late summer CH$_4$ maxima have been

linked to enhanced microbial activity under persistently anaerobic conditions and elevated temperatures in late summer (Pickett-Heaps et al., 2011). Additionally, a late summer decline in OH reactivity over boreal forests may contribute to elevated CH$_4$ levels. Measurements at the SMEAR II station in Hyytiälä, Finland, indicate that OH reactivity in boreal forest peaks in spring but decreases in late summer (Nölscher et al., 2012), potentially reducing CH$_4$ oxidation and allowing for more CH$_4$ in late summer. The late summer peaks at ZOTTO show a significant increasing trend over 2010-2021 period, suggesting there

is potential enhancing wetland activity over the years.

## 4.3 Diurnal Cycle

At ZOTTO, CH$_4$ mole fractions follow a marked diurnal cycle, with higher nighttime and lower daytime mole fractions. After sunset, rapid canopy cooling creates a stable NBL that traps CH$_4$ near the surface. By day, surface warming enhances vertical mixing, dispersing CH$_4$. The CH$_4$ diurnal cycle is most pronounced in warmer months due to larger diurnal temperature ranges,

which amplify both daytime mixing and nighttime inversions, increasing discrepancy in CH$_4$ mole fraction between day and night.

We examine the 2010-2021 trend and interannual variations in the diurnal amplitude at ZOTTO focusing on the summer months to further explore the potential factors contributing to the increased amplitude of the unique late-summer seasonal peak observed at the site. We observed a significant increase in the summer diurnal amplitude of CH$_4$ at ZOTTO from 2010 to 2021,

primarily driven by the significant rise in the nighttime CH$_4$ maxima. Our analysis of high-pressure system cases revealed no observed trends in synoptic or local atmospheric processes over the 11 years, either during the day or at night, that would





suggest changes in boundary layer and synoptic dynamics as the main drivers of the increasing diurnal amplitude trend. Instead, there is a strong significant positive correlation between nighttime surface flux and the $CH_4$ diurnal amplitude. This relationship is expected, as our flux estimates are derived from $CH_4$ mole fraction measurements; increases in nighttime $CH_4$ mole fraction

naturally led to higher inferred nighttime surface fluxes. For a more independent assessment of surface fluxes, eddy covariance measurements would be preferable. Nonetheless, our findings underscore that the increase in summer $CH_4$ diurnal amplitude at ZOTTO is mainly attributable to changes in surface emission processes, rather than shifts in atmospheric boundary layer structure or synoptic conditions over the study period.

There is also a significant increase in nighttime net $CH_4$ surface flux, particularly in August over the study period, indicating an intensification of late-summer emissions. This could potentially contribute to the increasing in the late-summer seasonal peak at ZOTTO.

The strong correlations between summer nighttime net surface $CH_4$ flux and soil temperature, soil moisture, and air temperature

reinforce the well-established relationship between microbial $CH_4$ production and environmental conditions (Basu et al., 2022; Bridgham et al., 2013). In contrast, the weak correlation with precipitation suggests that short-term rainfall events have a limited influence on $CH_4$ variability, with long-term soil moisture conditions playing a more dominant role. These findings align with studies from other boreal and wetland-dominated regions, where $CH_4$ emissions peak in late summer due to sustained high soil temperature and moisture leading to high microbial activity. For instance, Bohn et al. (2015) observed increasing

late-summer $CH_4$ emissions in Siberian peatlands, driven by persistent anaerobic conditions and enhanced methanogenesis. These findings underscore the importance of assessing the effects of environmental drivers not just for isolated snapshots in time but also considering their interactions over seasonal timescales. Our results show that higher VPD is associated with lower $CH_4$ emissions. This can be explained by the fact that higher VPD is typically an indicator of drier conditions, reducing soil moisture and limiting the anaerobic environments necessary for methanogenesis in wetlands.

High net surface $CH_4$ fluxes recorded in June and July 2012, as well as July and August 2019, coincided with major wildfire events, which have been associated with increased emissions of CO, $CO_2$, and PM2.5 aerosols in Siberia (Tran et al., 2024; Mokhov and Sitnov, 2022; Bondur et al., 2020). The 2012 wildfire season, one of the most severe in the decade, saw more than 17,000 wildfires detected in July and August alone. Satellite data indicated approximately 29,000 fire sources with a total fire radiative power of ~3 TW across a region from 50°-75° N, 60°-140° E in July 2019 (Bondur et al., 2020). Such extreme

events significantly contribute to regional $CH_4$ variability, both directly through biomass burning and indirectly by altering wetland hydrology and soil organic matter decomposition. The elevated emissions observed in June 2016 may be linked to increased wetland emissions, driven by the unusually high temperatures of that period. Sitnov and Mokhov (2018) reported a rise in regional atmospheric $CH_4$ levels in 2016, particularly over the Yamal Peninsula in Western Siberia, attributing the increase to intensified $CH_4$ emissions caused by the anomalous surface temperature spike.





In this study, the analysis of the diurnal $CH_4$ net surface fluxes is limited to nighttime, as our method, based on vertical $CH_4$ and potential temperature gradients, becomes unreliable during the day due to strong mixing, which minimises vertical gradients and hinders accurate flux estimation. Future studies could apply alternative methods, such as the Monin-Obukhov Similarity Theory (MOST) approach (Physick and Garratt, 1995), or to derive more accurate estimates of daytime net $CH_4$ surface fluxes. Another limitation of this study is that the flux analysis is constrained to the summer months, as meteorological

sensors at ZOTTO are highly susceptible to icing during colder months, leading to inaccurate or incomplete meteorological measurements, which are required for the net surface flux estimation. Consequently, this seasonal limitation restricts our ability to assess the complete annual cycle of the net $CH_4$ flux, particularly to investigate whether there is a trend in winter or spring time fluxes contributing to the observed increasing in the seasonal amplitude of $CH_4$ at ZOTTO. An eddy covariance system, which provides continuous and more direct measurements of surface-atmosphere exchange, could help overcome both the

daytime and seasonal limitations by enabling more accurate, year-round flux estimates independent of vertical gradient assumptions.

Further studies are needed to accurately attribute the sources responsible for the observed increasing nighttime summer $CH_4$ fluxes. The prevailing summer wind patterns at ZOTTO primarily originate from the west and southwest (Appendix B – Fig. B1), where extensive inland marshes dominate the landscape (Zhang et al., 2023). While this suggests that enhanced wetland

activity is a major driver of increased $CH_4$ emissions, wind direction analysis alone does not provide precise source attribution. To overcome this limitation, future studies should integrate inverse modelling techniques, which estimate $CH_4$ fluxes by combining atmospheric observations with transport models. By using column-integrated $CH_4$ mole fraction and tracing them back to their origins, inverse modelling can help differentiate between various $CH_4$ sources, including wetlands, biomass burning, and thermogenic emissions (Mikaloff Fletcher et al., 2004). Additionally, these techniques could be used to

investigate the annual cycle of $CH_4$ flux and evaluate whether the observed increase in the $CH_4$ seasonal amplitude at ZOTTO is a consistent trend.

## 5 Conclusions

We investigate the temporal variability of $CH_4$ in Central Siberia across annual, seasonal and diurnal scales by utilising the 2010-2021 ZOTTO continuous dataset. This study provides new evidence that warming over the last decade has enhanced

$CH_4$ surface net fluxes in Central Siberia. We demonstrate that the observed enhancement of the summer-time diurnal amplitude (5.55 ppb year$^{-1}$, p = 0.002) is driven by an increase in nighttime surface fluxes and not by changes in atmospheric dynamics. This nighttime surface flux is positively correlated with air and soil temperature and soil moisture ($R^2 = 0.65$, p = 0.042; $R^2 = 0.7$, p < 0.001; and $R^2 = 0.60$, p = 0.031 respectively). The increase in net $CH_4$ surface flux during the summer months suggests a growing contribution from wetlands. Episodes of high $CH_4$ flux are observed in 2012 and 2019 due to

wildfire and 2016 due to increase wetland activity.



The seasonal analyses reveals an insignificant upward trend of CH₄ winter peak-late spring trough amplitude at ZOTTO over the past decade. A significant increase in the distinct late summer CH₄ peak in August at ZOTTO underscores the enhancement of regional wetland emissions. The persistent rise in CH₄ mole fractions at ZOTTO reflects global trends, underscoring the sustained impact of biogenic emissions, especially from wetlands, which are increasingly active due to rising global temperatures.

Our study highlights the importance of CH₄ surface fluxes in driving diurnal variations. In this context, advancing our understanding of soil microbial activity through direct measurements could improve estimates of ecosystem surface fluxes at both daily and seasonal scales. To extend these findings to the regional scale, high-resolution atmospheric data, combined with inverse modelling, will enhance our ability to accurately attribute CH₄ sources and sinks at ZOTTO. Continued monitoring and improved modelling efforts are critical for refining our understanding of CH₄ variability and assessing its implications for future climate feedback.

**Appendices**

**Appendix A: ZOTTO Meteorology Analysis**

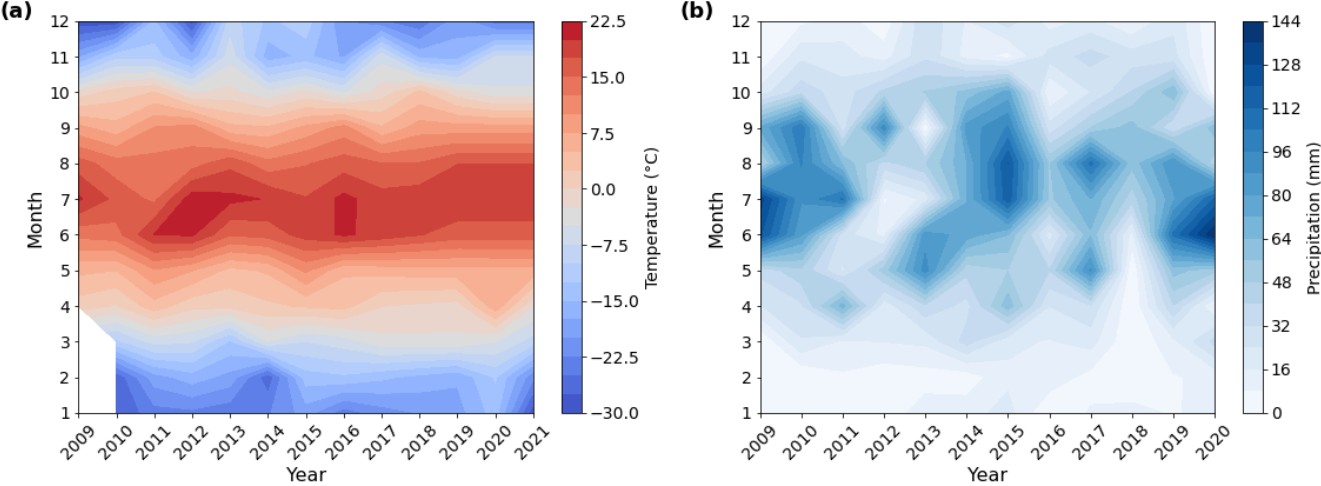

**Figure A1. (a) Temperature measured at 52 m a.g.l (in °C) and (b) precipitation measured at 2 m a.g.l (mm) for the period 2009-2021 at ZOTTO.**





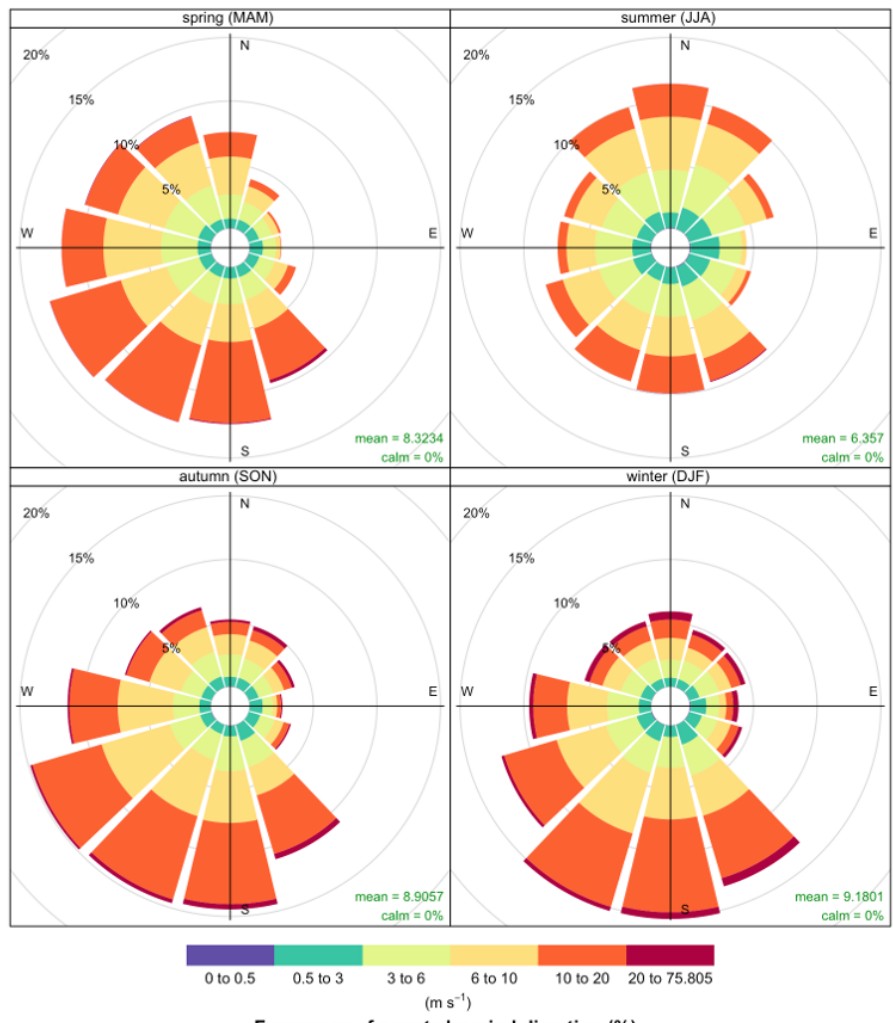

**Figure A2. Wind rose showing the average wind speed distribution at 300 m a.g.l at ZOTTO from 2010 to 2021, categorized by 12 horizontal wind directions across the four seasons.**




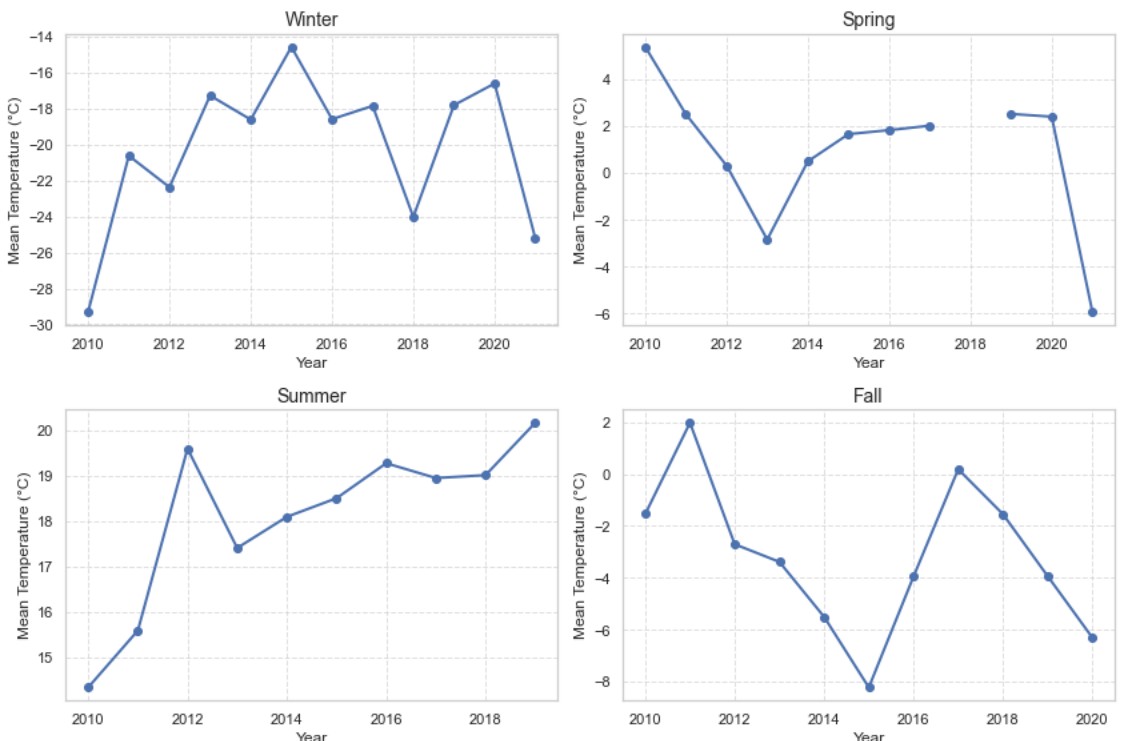

**Figure A3. Timeseries of yearly temperature measured at 52 m a.g.l at ZOTTO for each season over the 2010-2021 period.**

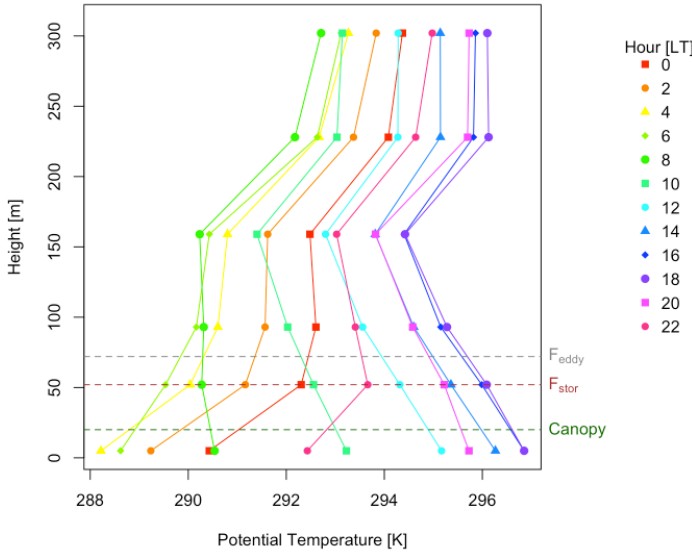

**Figure A4. Climatological (2010-2021) summer (JJA) vertical profile of potential temperature at ZOTTO. The dashed green line**
**denotes the canopy height at ZOTTO, while the dashed brown and grey lines indicate the 52 m and 72 m levels, where the $F_{Stor}$ and**





**$F_{Eddy}$ terms in Eq. (4) in the main text are calculated, respectively. At night, the vertical gradient in potential temperature between the 227 m and 301 m levels is minimal, indicating that the 301 m level is already within the residual layer during nighttime conditions.**

**Appendix B: Target Tank Time Series**

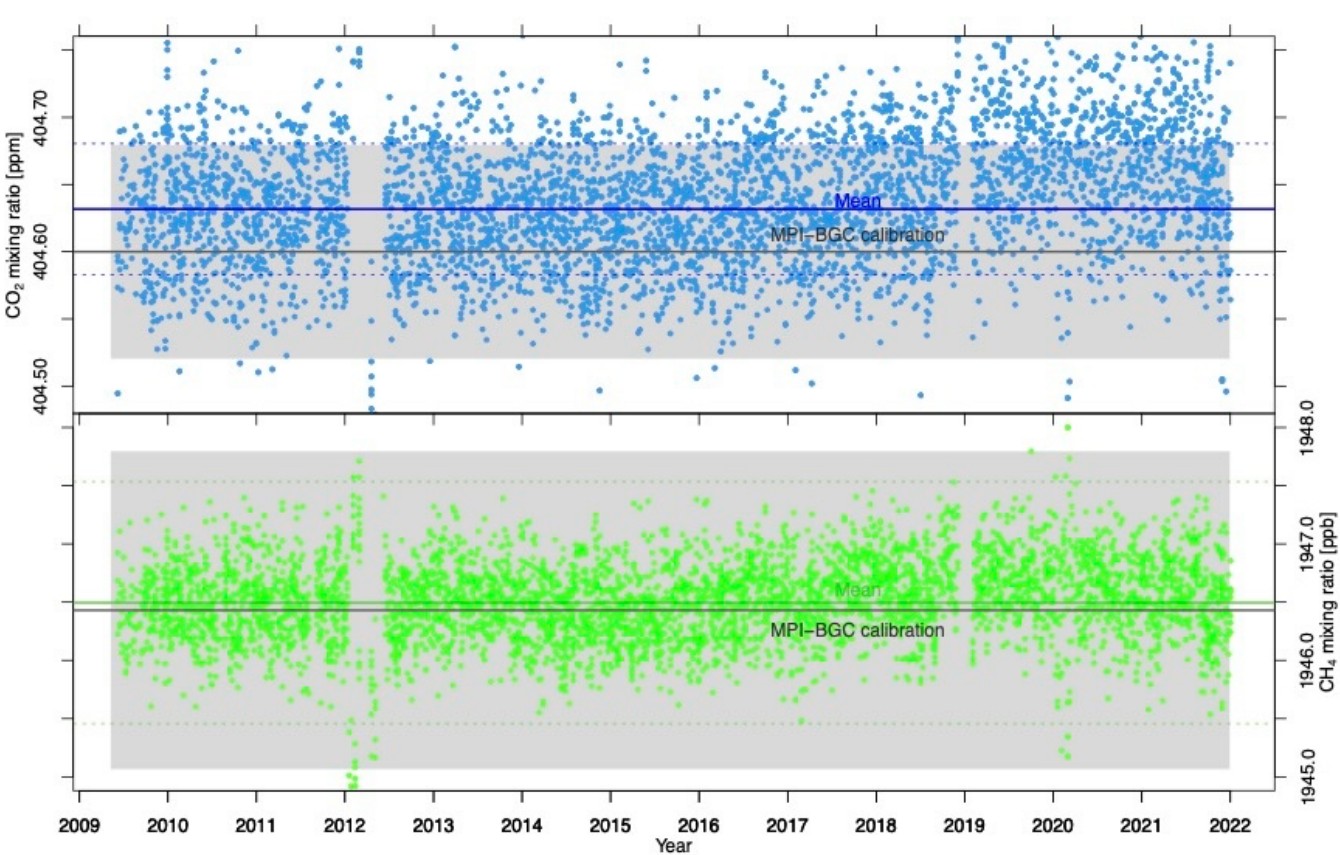

**Figure B1. Target tank time series (coloured line represents the mean ± standard deviation, grey is laboratory standard ± error).**

**Appendix C: Details explanations of the Atmospheric Drivers of CH₄ Diurnal Amplitude**

**C1. Governing Equations for the Atmospheric Drivers of CH₄ Diurnal Amplitude**

Terms I and III of Eq. (1) represent key atmospheric dynamics with distinct daytime and nighttime characteristics that influence both $CH_{4,min}$ and $CH_{4,max}$ and will be discussed below.

610             C1.1 Atmospheric Boundary Layer Height – Term I

Regarding Term I in Eq. (1), the diurnal dynamics of the atmospheric boundary layer height (ABL) (h) are driven by the surface buoyancy flux introduced into the ABL. This process is represented by the potential temperature variable (θ), solved using three additional equations as described in Appendix C2 and more detailed in Vilà-Guerau de Arellano et al. (2015). Note that we are not using Appendix C2 to solve for the ABL height in this study but to solely explain the processes. In short, the



effect of h on CH4 is the following. In the morning, solar heating destabilizes the atmospheric column, warming the surface and causing less dense air to rise. This results in an upward transfer of surface heat flux ($\overline{(w'\theta')}_s > 0$), driving turbulent convective motions. These turbulent processes increase the height of the CBL, expanding the atmospheric volume available for CH4 dilution, decreasing CH4 mole fractions.

At night, radiative cooling of the ground creates a temperature gradient where heat flows downward ($\overline{(w'\theta')}_s < 0$) from the
warmer air to the cooler surface. This cooling stabilizes the lower atmospheric layers, leading to the formation of a stratified NBL, typically ranging between 100 and 300 m above ground level (Kubiak and Zimnoch, 2022). The NBL traps surface-emitted CH4, limiting its vertical dispersion and promoting nighttime CH4 accumulation, increasing CH4 mole fractions.

### C1.2. Entrainment flux – Term III

The entrainment flux ($\overline{(w'\phi')}_e$) in Term II in Eq. (1) could be written as in Vilà-Guerau de Arellano et al. (2015) and expressed
as below:

$$\overline{(w'\phi')}_e = -\left(\left(\underbrace{\frac{\partial h}{\partial t}}_{III.1} - \underbrace{w_{sub(h)}}_{III.2}\right) \times \underbrace{\Delta_{\phi(ft/rs-cbl/nbl)}}_{III.3}\right) \tag{C1}$$

According to Eq. (C1), the entrainment flux ($\overline{(w'\phi')}_e$) depends on: the growth rate of the ABL height ($\frac{\partial h}{\partial t}$, Term III.1) which is dependent on the surface buoyancy flux; the large-scale vertical subsidence velocity ($w_{sub(h)}$, Term III.2); and the difference in CH4 mole fraction between the CBL (or NBL) and the overlying layer ($\Delta_{\phi(ft/rs-cbl/nbl)}$, Term III.3) – either the free
troposphere (FT) during the day or the residual layer (RS) at night (Fig. 1 in the main text). For the latest we assume that this jump occurs in an infinitesimal layer (zero-order approach) (Driedonks and Tennekes, 1981) Higher entrainment flux rates introduce more CH4-depleted air from the overlying layer into the integrated CH4 column, reducing the overall CH4 mole fraction. This exerts a negative impact on both CH4,max and CH4,min.

To account for the distinct atmospheric dynamics between day and night, each term in Eq. (C1) is applied differently for
nighttime and daytime, as detailed in the following sections.

### C1.2.1 Growth Rate of the ABL – Term III.1

The growth rate of the boundary layer height ($\frac{\partial h}{\partial t}$) is closely linked to the surface buoyancy flux ($\overline{(w'\theta')}_s$), as discussed earlier similarly. This term is more pronounced during the daytime, as strong turbulence and convection drive rapid changes in CBL height, while weaker nighttime turbulence results in slower changes in NBL height. A stronger $\frac{\partial h}{\partial t}$ amplifies the entrainment
flux, making entrainment more pronounced during the daytime than nighttime.

### C1.2.2 Vertical Subsidence Velocity – Term III.2

The vertical subsidence velocity represents large-scale downward motion in the atmosphere, primarily driven by synoptic-scale conditions. Using the mass conservation equation assuming incompressibility, we represent the vertical subsidence velocity at ABL height ($w_{sub(h)}$) as:





$$w_{sub(h)} = -Div(\vec{U_h}) \cdot h \tag{C2}$$

Where $Div(\vec{U_h})$ is the horizontal wind divergence. While subsidence velocities are generally small (rarely exceeding a few cm s$^{-1}$) and the same magnitude as the entrainment velocity, they can significantly influence mass conservation and the growth of the CBL/NBL (Stull, 1988), and consequently, the entrainment flux.

During the daytime, subsidence slows the growth of the CBL by introducing downward motion, which counters the upward expansion driven by surface heating and turbulence. This downward motion brings warmer, drier air from the free troposphere into the CBL, stabilizing the atmosphere and weakening convective activity. At the same time, the temperature and moisture contrast between the warm, dry overlying air and the CBL air enhances entrainment at the top of the CBL.

At night, divergence associated with subsidence laterally transport cold air masses generated by the longwave radiative cooling at NBL air causing the NBL to not grow as rapidly as would otherwise be expected (Carlson and Stull, 1986). This additional stabilization further suppresses turbulence and reduces vertical mixing within the NBL. While nighttime entrainment is minimal, subsidence could still contribute to the transport of air from the residual layer above, influencing the temperature and composition of the NBL (Carlson and Stull, 1986).

C1.2.3 The Difference in $CH_4$ Mole Fraction between the CBL (or NBL) and the Overlying Layer – Term III.3

The entrainment flux of $CH_4$ ($\overline{(w'\phi')}_e$) is also influenced by the difference in $CH_4$ mole fractions between the daytime (nighttime) CBL (NBL) and the layer above it, which is the FT (RS) ($\Delta_{\phi(ft/rs-cbl/nbl)}$). This difference evolves over time and can be expressed as:

$$\frac{\partial \Delta_{\phi(ft/rs-cbl/nbl)}}{\partial t} = \frac{\partial \phi_{(ft/rs)}}{\partial t} - \frac{1}{h-ct}\int_{z=ct}^{z=h}\frac{\partial \phi_{(cbl/nbl)}}{\partial t}dz \tag{C3}$$

Here, $\phi_{(ft/rs)}$ represents the $CH_4$ mole fraction in the FT during the daytime or in the RL during the nighttime. The last term on the right-hand side of Eq. (C3) refers to the averaged column integrated $CH_4$ mole fraction from the canopy top to the top of the CBL during the day or the NBL during the night.

During the daytime, strong convective turbulence in the CBL leads to a well-mixed $CH_4$ distribution, making its mole fraction independent of height. This is evident from minimal vertical gradients in $CH_4$ mole fraction, as shown in Fig. 3 in the main text and Fig. H3 in Appendix H (i.e. P3 period at ZOTTO). Under such conditions, the Eq. (C3) could be simplified to:

$$\frac{\partial \Delta_{\phi(ft-cbl)}}{\partial t} = \frac{\partial \phi_{(ft)}}{\partial t} - \frac{\partial \phi_{(cbl)}}{\partial t} = \gamma\frac{\partial h}{\partial t} - \frac{\partial \phi_{(cbl)}}{\partial t} \tag{C4}$$

Here, the rate of change of the free tropospheric $CH_4$ mole fraction over time ($\frac{\phi_{(ft)}}{\partial t}$) is proportional to the product of the tropospheric $CH_4$ lapse rate ($\gamma$) and the boundary layer height growth ($\frac{\partial h}{\partial t}$). Assuming a zero tropospheric $CH_4$ lapse rate ($\gamma = 0$) as in Faassen et al. (2024), integration yields:

$$\Delta_{\phi(ft-bl)}(t_i) = \Delta_{\phi}(t_0) + (\phi_{cbl}(t_0) - \phi_{cbl}(t_i)) \tag{C5}$$

Where $\phi_{cbl}(t_0)$ and $\Delta_{\phi}(t_0)$ represent the initial $CH_4$ mole fraction and the initial difference in $CH_4$ mole fraction between the ABL and the layer above it just before sunrise (e.g., P2 period in Fig. H3 in Appendix H). ($\phi_{cbl}(t_0) - \phi_{cbl}(t_i)$) represents



the change in total $CH_4$ mole fraction in the CBL over time. The initial difference ($\Delta_\phi(t_0)$) is influenced by nighttime stability, which will be discussed below.

During nighttime, the is a clear vertical gradient of $CH_4$ mole fraction (Fig. 3 in the main text and Fig. H3 in Appendix H, i.e., P1 period at ZOTTO), suggesting integrated column of $CH_4$ depends on height. Equation (C3) during nighttime could be
written as:

$$\frac{\partial \Delta_{\phi(rs-nbl)}}{\partial t} = \frac{\partial \phi_{(rs)}}{\partial t} - \frac{1}{h-ct}\int_{z=ct}^{z=h} \frac{\partial \phi_{(nbl)}}{\partial t}\,dz \qquad\qquad (C6)$$

During the night, the $CH_4$ mole fraction in the RL remains almost constant on time ($\frac{\partial \phi_{(rs)}}{\partial t} = 0$) because the RL is largely decoupled from surface processes (Stull, 1988). In absence of sources and sinks of $CH_4$, the mole fraction of $CH_4$ remains almost constant. Essentially, the RL "stores" the composition of the previous daytime mixed layer. As a result, the difference
in $CH_4$ mole fractions between the RL and NBL depends primarily on the rate of averaged $CH_4$ accumulation from the top of the canopy to NBL height ($\int_{z=ct}^{z=h} \frac{\partial \phi_{(nbl)}}{\partial t}$). Since this term is relatively constant overnight (as seen in P1 period in Fig. C1 and in Winderlich (2014)), $\frac{\partial \Delta_{\phi(rs-nbl)}}{\partial t}$ remains minimal, resulting in limited nighttime entrainment.

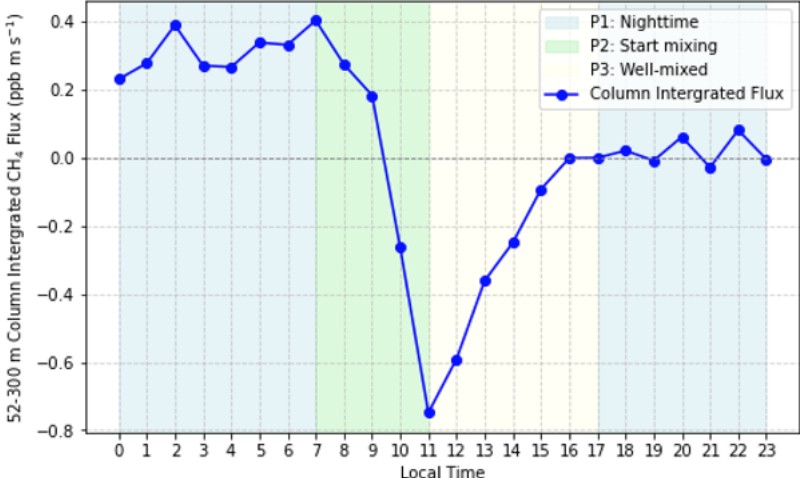

**Figure C1. Climatological (2010-2021) summer (JJA) diurnal cycle of the column-integrated of $CH_4$ mole fraction from 52 m to 300**
**m, representing the layer above the canopy to the top of the NBL during P1. The 52-300 m column-integrated flux is calculated using a method similar to Eq. (6) in the main text but applied to the 52-300 m layer. This calculation accounts for temporal changes in the 52-300 m column-integrated $CH_4$ mole fraction and the influence of vertical subsidence velocity at 300 m.**

However, this dynamic changes significantly at sunrise. As surface heating begins, convection resumes, and the turbulent eddies lead to well-mixed conditions and reconnecting it with the RL. This process leads to a sharp reduction in the averaged
$CH_4$ accumulation from the top of the canopy to NBL height (as seen in P2 period in Fig. C1), as $CH_4$ accumulated near the surface during the night is rapidly mixed into the expanding convective boundary layer (CBL). Consequently, there is a dramatic increase in the entrainment flux at sunrise.




The thermal stability of the nighttime atmospheric column plays a key role in this process. A more stable column leads to a lower NBL height, which increases the near-surface $CH_4$ mole fraction and enhances the storage flux overnight as well as

larger $CH_4$ difference between the RL and the NBL ($\Delta_\phi(t_0)$). At sunrise, the greater storage flux results in a larger reduction when convection begins, as well as a higher $\Delta_\phi(t_0)$ right before sunrise, driving stronger entrainment flux into the CBL.

**C2. Governing Equations for the evolution of the Atmospheric Boundary Layer (ABL)**

The dynamic evolution of the ABL is solely driven by the heat introduced in the ABL, represented by the virtual potential temperature ($\theta$) variable.

To describe the evolution of the ABL potential temperature ($\theta$) and the discontinuity jump of potential temperature ($\Delta\theta$) at the inversion (i.e. top of the ABL), we solve three fundamental equations as described in Vilà-Guerau de Arellan et al. (2015) and in (Driedonks and Tennekes, 1981). These equations result from the vertical integration of the conservation equations for $\theta$ within the canopy-top-to-ABL-top layer and the entrainment zone.

$$\frac{1}{h-ct}\int_{z=ct}^{z=h}\frac{\partial\theta}{\partial t}(z)dz = \frac{1}{h-ct}\times\left(\overline{(w'\theta')}_s - \overline{(w'\theta')}_e\right) - adv(\theta) \qquad (C7)$$

$$\frac{\partial\Delta\theta}{\partial t} = \gamma_\theta\left(\frac{\partial h}{\partial t}-w_s\right) - \frac{1}{h-ct}\int_{z=ct}^{z=h}\frac{\partial\theta}{\partial t}(z)dz = \gamma_\theta.w_e - \frac{1}{h-ct}\int_{z=ct}^{z=h}\frac{\partial\theta}{\partial t}(z)dz \qquad (C8)$$

Equation (C7) shows that within the top of the canopy height to the ABL height, the tendency term of $\theta$ on the left hand sides depends on the vertical turbulent flux difference between the surface heat flux ($\overline{(w'\theta')}_s$) and entrainment zone heat flux $\overline{(w'\theta')}_e$, the horizontal advection $adv(\theta)$ which is current not accounted for.

The evolution of the discontinuity or jump value $\Delta\theta$ at the entrainment zone, see Eq. (C8), is a function of the tendency value
at the residual layer or free troposphere (first term right-hand side) and the evolution of the canopy-top-to-ABL-top value ($2^{nd}$ term r.h.s). Above the jump, the profile of $\theta$ in the layer above the ABL is dependent on the vertical gradient ($\gamma_\theta$) and on the mean subsidence vertical velocity ($w_s$). This velocity is normally opposite to the boundary layer growth ($\frac{\partial h}{\partial t}$), i.e., subsidence ($w_s < 0$). We assume that the $w_e$ is a function of the entraiment flux and the jump in the virtual potential temperature in the inversion layer. This assumption is known as zero-order closure and it was first suggested by Lilly (1968). It is expressed
mathematically by:

$$w_e = \frac{\partial h}{\partial t} - w_s = -\frac{\overline{(w'\theta')}_e}{\Delta\theta} \qquad (C9)$$

Equation (C9) assumes that the inversion is characterized by a sharp discontinuity (Driedonks and Tennekes, 1981). Under conditions of weak inversion, it is convenient to include explicitly the inversion depth requiring a modification of Eq. (C9) (Kim et al., 2006). For $\theta$, Eqs. (C7-C9) contain seven variables: $h$, $\frac{1}{h-ct}\int_{z=ct}^{z=h}\theta(z)dz$, $\Delta\theta$, $\overline{(w'\theta')}_s$, $\overline{(w'\theta')}_e$, $\gamma_\theta$ and $w_s$. The
first three are solved by the system Eqs. (C7-C9) the other four need to be prescribed or calculated using additional equations or closure assumptions. The heat surface fluxes ($\overline{(w'\theta')}_s$) are either prescribed based on field measurements or calculated using a coupled land-surface scheme. The subsidence velocity ($w_s$) and the potential temperature lapse rate in the ABL





overlaying layer ($\gamma_\theta$) depend on the atmosphere at large scales. These upper boundary conditions are thus obtained either from large-scale models or by a radiosounding taken in the early morning hours of the ABL development. In consequence, to close

the set of Eqs. (C7), (C8) and (C9), we still need to relate the entrainment of heat flux to the surface flux. We assume the following relation, $\overline{(w'\theta')}_e = -\beta\,\overline{(w'\theta')}_s$, where $\beta$ represents an additional percentage of entrainment of warm air into the ABL. Here, it needs to be mentioned that the $\beta$-value can increase, depending on the contribution of shear in the ABL development (Angevine et al., 1998; Pino et al., 2003; Conzemius and Fedorovich, 2006).

**Appendix D: Nocturnal Boundary Layer Height Estimation**

This section visual the method to estimate the Nocturnal Boundary Layer (NBL) height from the regression fit (Eq. (2) and (3) in the main text) applying to the vertical gradient of $CH_4$ and potential temperature in Fig. D1. In some years, summer NBL heights based on vertical temperature profiles are unavailable (Fig. D3) due to meteorological instrument malfunctions at specific heights, resulting in incomplete data for constructing full vertical profiles.

The NBL heights derived from vertical $CH_4$ concentrations and potential temperature exhibit similar ranges, generally falling

between 100-150 m (Figs. D2 and D3), and show comparable interannual variability over the 2010-2021 period. However, there is no strong 1:1 correlation between the NBL derived by the two parameters as shown in Fig. D4. This discrepancy may result from a time lag in the vertical profile development between potential temperature and $CH_4$. The nighttime vertical $CH_4$ profile decreases with height (Fig. 3), mirroring the pattern of potential temperature (Fig. A4). The nighttime vertical $CH_4$ profile stability persists from 00:00 LT to 08:00 LT, peaking at 08:00 LT and beginning to weaken around 10:00 LT, later than

the temperature profile. This observed delay between potential temperature and measurement gases has also been observed in the 213 m tower in Cabauw (CBW: 51°97′ N, 04°93′ E, 0 m a.s.l) in the Netherlands (Casso-Torralba et al., 2008). This could be caused by the larger difference of the $CH_4$ mole fraction between the ABL and the free troposphere compared to potential temperature (Casso-Torralba et al., 2008).





**Figure D1. Vertical profile of potential temperature during the nighttime on June 21, 2010, presented as an example day for estimating the nocturnal boundary layer height. The curve fit and normalised lapse rate are calculated using Eq. (2) and (3) respectively in the main text.**




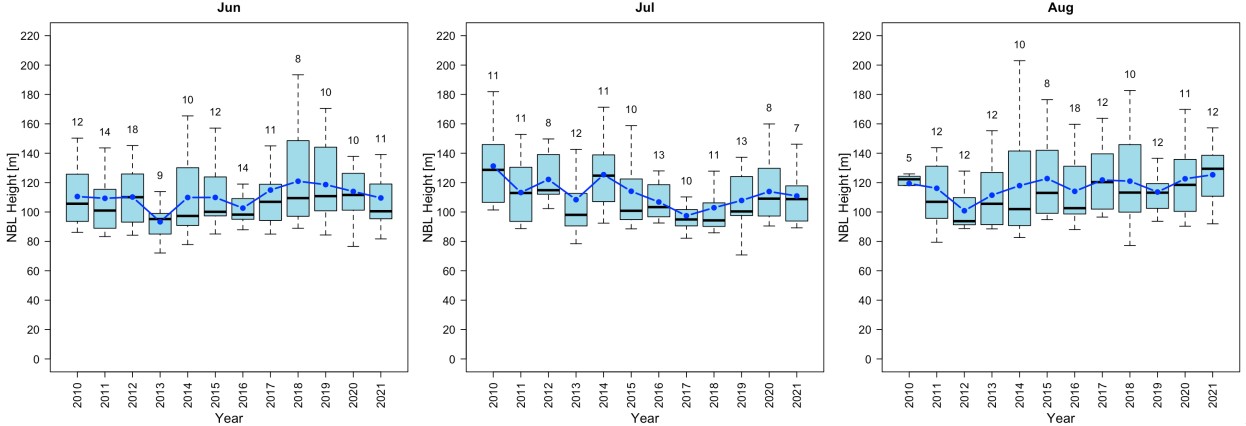

**Figure D2. Box-and-whisker plot of the yearly nocturnal boundary layer (NBL) height (averaged from 00:00 to 04:00 LT) at ZOTTO, derived from $CH_4$ vertical profiles for each summer month by applying Eq. (2) and (3) in the main text to $CH_4$. The box denotes the interquartile range (IQR), showing the median with a thick black line. The whiskers range from $Q_1 - 1.5 \times IQR$ to $Q_3 + 1.5 \times IQR$, with $Q_1$ and $Q_3$ being the 25th and 75th percentiles, respectively. The blue line is the monthly mean. Numbers above each box indicate the sample size or the number of available days for analysis in that month.**

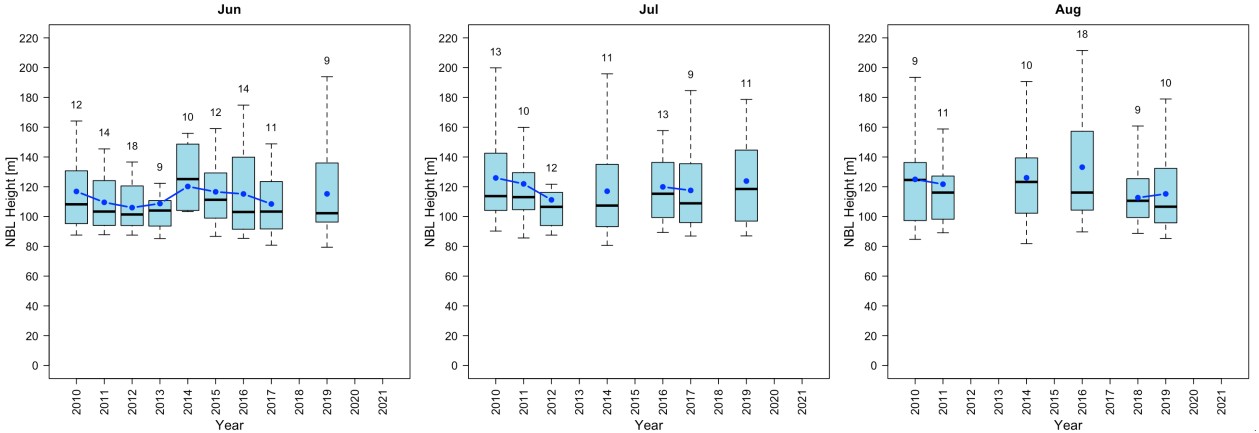

**Figure D3. Box and whisker plot of yearly nocturnal boundary layer (NBL) height (00:00 − 04:00 LT averaged) height derived from potential temperature vertical profile criteria at ZOTTO for each summer month by applying Eq. (2) and (3) in the main text to potential temperature. The box denotes the interquartile range (IQR), showing the median with a thick black line. The whiskers range from $Q_1 - 1.5 \times IQR$ to $Q_3 + 1.5 \times IQR$, with $Q_1$ and $Q_3$ being the 25th and 75th percentiles, respectively. The blue line is the monthly mean. Numbers above each box indicate the sample size or the number of available days for analysis in that month.**




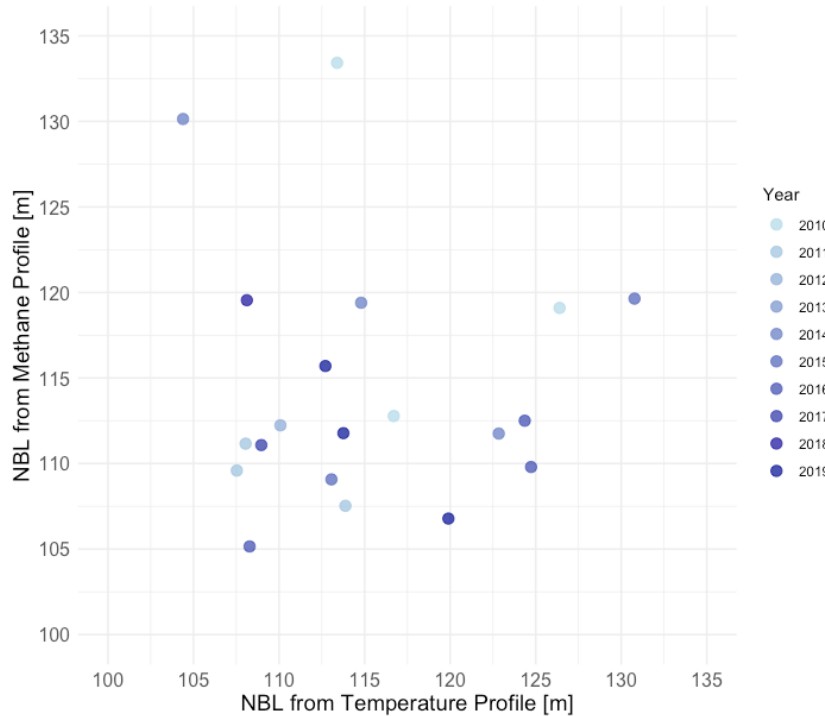


**Figure D4. Comparison of the monthly average summer NBL height derived from the potential temperature vertical profile and the CH$_4$ vertical profile using Eq. (2) and (3).**

**Appendix E: Number of High-pressure Days for Each Month from 2010-2021**

**Table E1. Number of high-pressure days identified using ERA5 geopotential height data at 550 hPa, selecting periods where geopotential height exceeded the 90th percentile of yearly summer distribution.**

| Year | Month | Number of High-Pressure Day |
|------|-------|------------------------------|
| **2010** | 6 | 12 |
| | 7 | 13 |
| | 8 | 9 |
| **2011** | 6 | 14 |
| | 7 | 13 |
| | 8 | 12 |
| **2012** | 6 | 18 |
| | 7 | 15 |
| | 8 | 12 |
| **2013** | 6 | 9 |
| | 7 | 12 |



|  | 8 | 12 |
|---|---|---|
| **2014** | 6 | 10 |
|  | 7 | 11 |
|  | 8 | 11 |
| **2015** | 6 | 12 |
|  | 7 | 10 |
|  | 8 | 8 |
| **2016** | 6 | 14 |
|  | 7 | 13 |
|  | 8 | 18 |
| **2017** | 6 | 11 |
|  | 7 | 10 |
|  | 8 | 13 |
| **2018** | 6 | 8 |
|  | 7 | 11 |
|  | 8 | 10 |
| **2019** | 6 | 10 |
|  | 7 | 15 |
|  | 8 | 12 |
| **2020** | 6 | 10 |
|  | 7 | 8 |
|  | 8 | 11 |
| **2021** | 6 | 11 |
|  | 7 | 7 |
|  | 8 | 12 |

**Appendix F: Formula Symbols and Units**

## Table F1. Formula Symbol and Units

| SYMBOLS | NAME | UNIT |
|---|---|---|
| $\phi$ | $CH_4$ mole fraction | ppb or nmol mol$^{-1}$ |
| h | Atmospheric Boundary Layer Height | m |
| ct | Top of the canopy height | m |



| $\overline{(w'\phi')}_{NS}$ | Net Surface flux | ppb m s$^{-1}$ |
|---|---|---|
| $\overline{(w'\phi')}_e$ | Entrainment flux | ppb m s$^{-1}$ |
| $adv(\phi)$ | Horizontal advection flux | ppb m s$^{-1}$ |
| $S_\phi$ | Net CH$_4$ flux from chemical reactions | ppb m s$^{-1}$ |
| $w_{sub(h)}$ | Subsidence velocity at Atmospheric Boundary Layer Height | m s$^{-1}$ |
| $\Delta_{\phi(ft/rs-cbl/nbl)}$ | Difference in CH$_4$ mole fraction between the Atmospheric Boundary Layer and its overlying layer | ppb or nmol mol$^{-1}$ |
| $-Div(U_h)$ | Horizontal wind divergence | s$^{-1}$ |
| $\gamma$ | Tropospheric CH$_4$ lapse rate | ppb m$^{-1}$ |
| $w_{sub(ct)}$ | Top of the canopy height | m s$^{-1}$ |
| $F_{Eddy}$ | Eddy turbulent flux | ppb m s$^{-1}$ |
| $F_{Stor}$ | Storage flux | ppb m s$^{-1}$ |
| $X$ | Potential temperature lapse rate | K m$^{-1}$ |
| $H$ | Sensible heat flux | W m$^{-2}$ |
| $\rho$ | Air density | g m$^{-3}$ |
| $C_p$ | Specific heat capacity constant | J g$^{-1}$ K$^{-1}$ |
| $\theta$ | Potential Temperature | K |



**Appendix G: Climatology Nighttime Column Integrated CH₄ flux up to 52 m at ZOTTO**

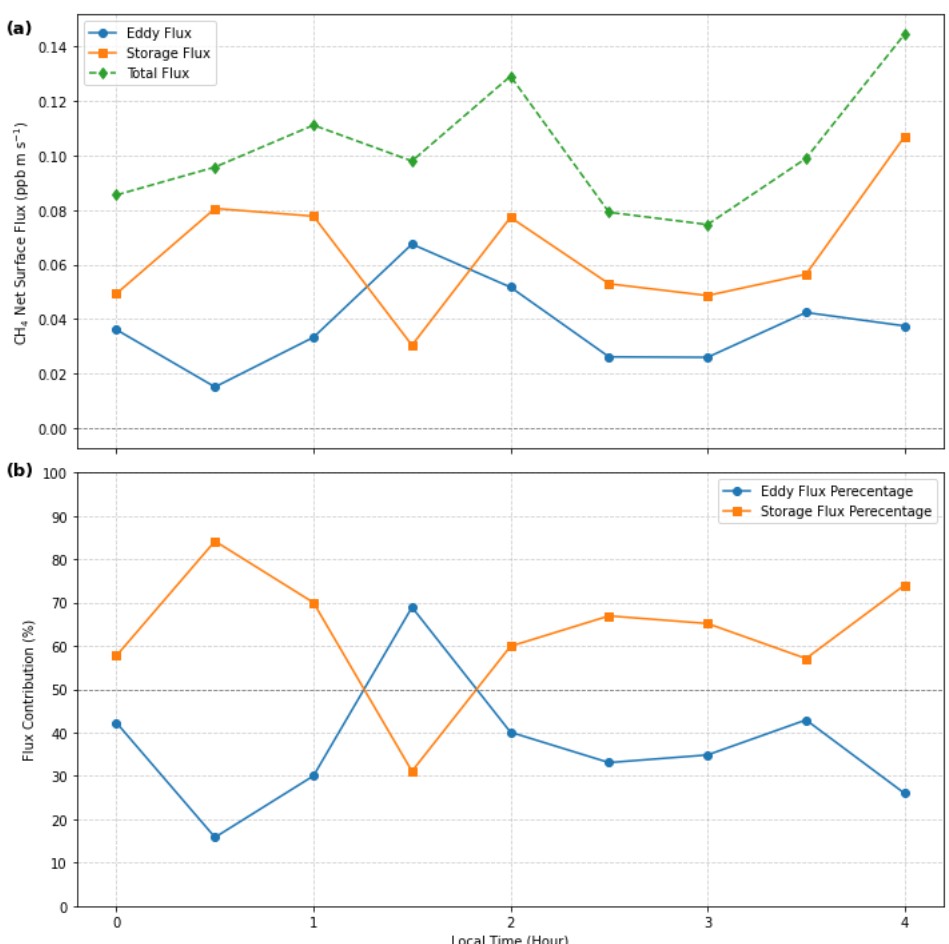

**Figure G1. Climatological (2010-2021) summer (JJA) nighttime (00:00-04:00 LT) CH₄ net surface flux up to 52 m. The individual components of Eq. (4) in the main text are plotted in (a) and their contributions to the total flux net surface flux (in percentage) are plotted in (b).**





**Appendix H: CH₄ Additional Annual Growth Rate, Seasonal and Diurnal Analysis.**

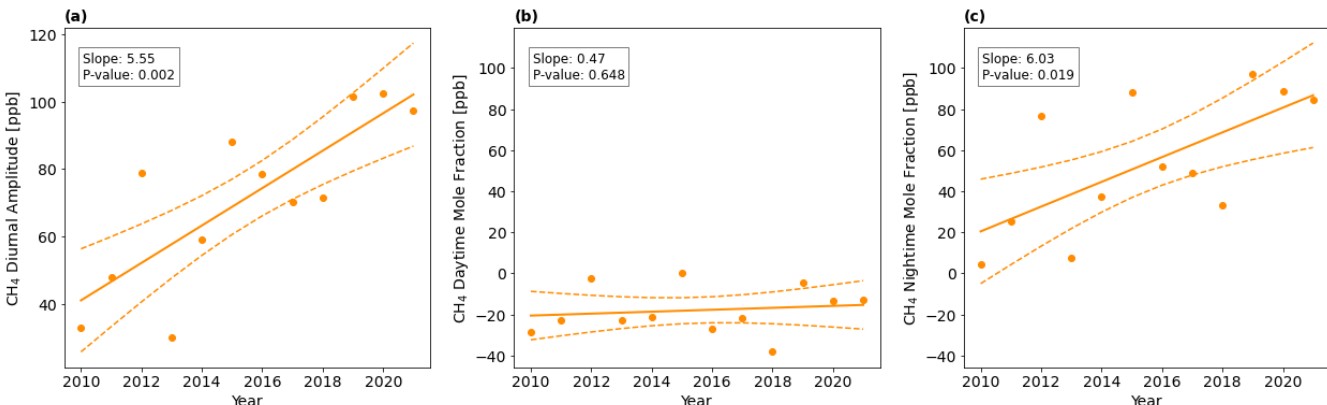

**Figure H1. Time series of yearly summer (JJA): (a) averaged CH₄ diurnal cycle amplitude; (b) its daytime (10:00-16:00 LT averaged) CH₄ mole fraction, and (c) its nighttime (00:00-04:00 LT averaged) CH₄ mole fraction (right) (circle markers) at ZOTTO using detrended 52 m a.g.l. data. The Theil-Sen regression trend is depicted by the solid line, with the 95% confidence interval of the trend shown as a dashed line. The p-value indicates whether the slope of the regression is significantly different from zero.**




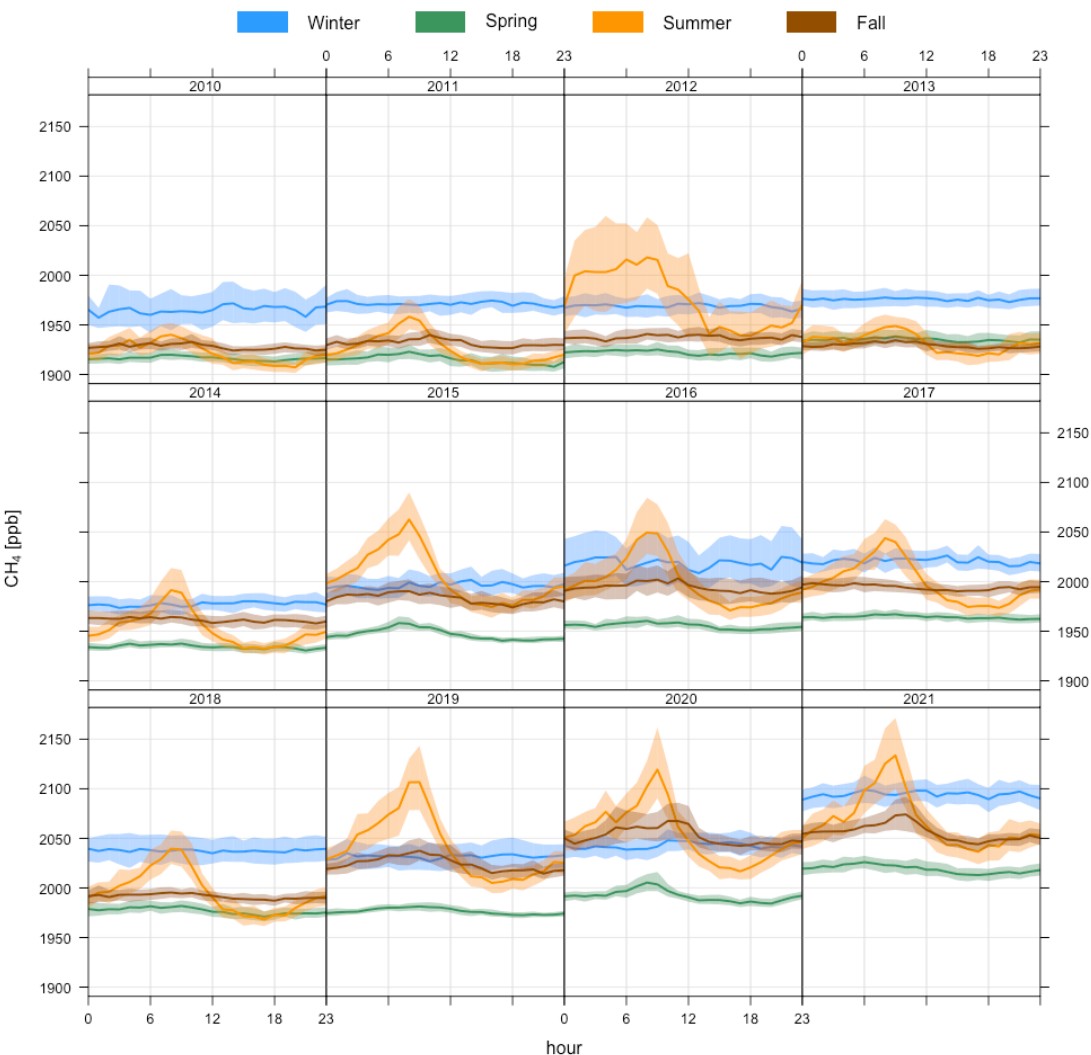

**Figure H2. Yearly diurnal cycle of CH$_4$ from 2010 to 2021 at ZOTTO 52 m a.g.l (i.e. above the canopy), in different seasons using non-detrended data. The shaded colours show 95% confidence interval of the mean.**




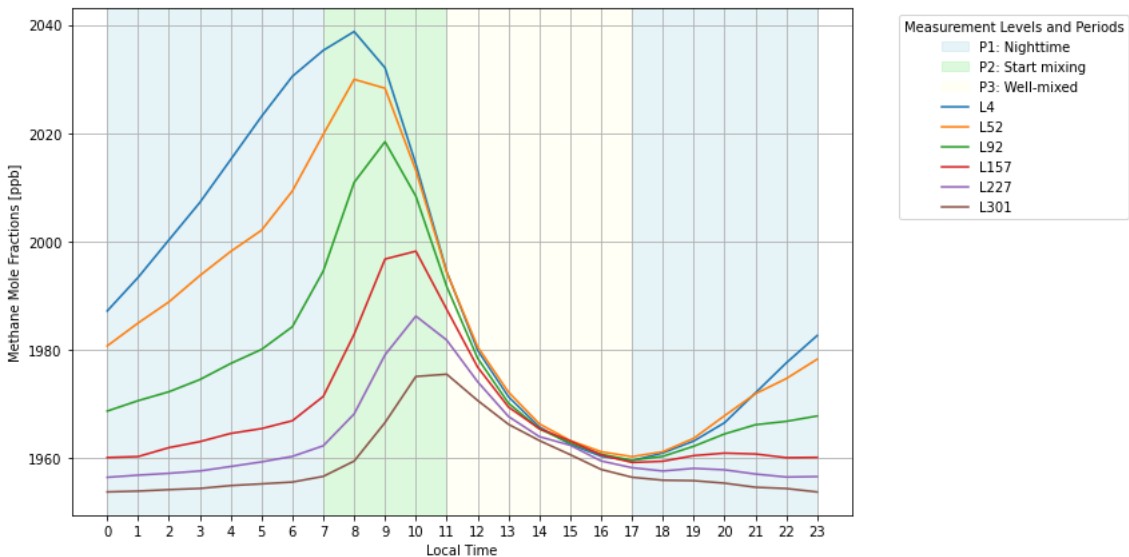

**Figure H3. Climatological (2010-2021) summer (JJA) diurnal cycle of CH₄ mole fractions measured at six different heights at**
**ZOTTO. The shadings represent different time periods: P1 corresponds to the nighttime period (17:00 to 07:00 LT (KRAT, UTC+7))**
**the following day), P2 marks the onset of mixing around sunrise (7:00 to 11:00 LT), and P3 represents the well-mixed period (11:00**
**to 17:00 LT).**

**Appendix I: Trend Analysis for Atmospheric Drivers of CH₄ Diurnal Amplitude in Summer Months**

The interannual variations in atmospheric process drivers influencing the summer CH₄ diurnal amplitude (dashed yellow boxes
in Fig. 2 in the main text) over the 2010-2021 are analysed in detail in this section.

Interannual variations are observed in the heights of both the Convective Boundary Layer (CBL) and the Nocturnal Boundary
Layer (NBL), but no significant long-term trends are detected over the study period (Fig. I1 and D3). At ZOTTO, the 12:00-
16:00 LT averaged CBL height typically reaches approximately 1500 m, while the NBL height, averaged from 00:00 LT to
04:00 LT, generally ranges between 100 and 150 m.

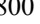

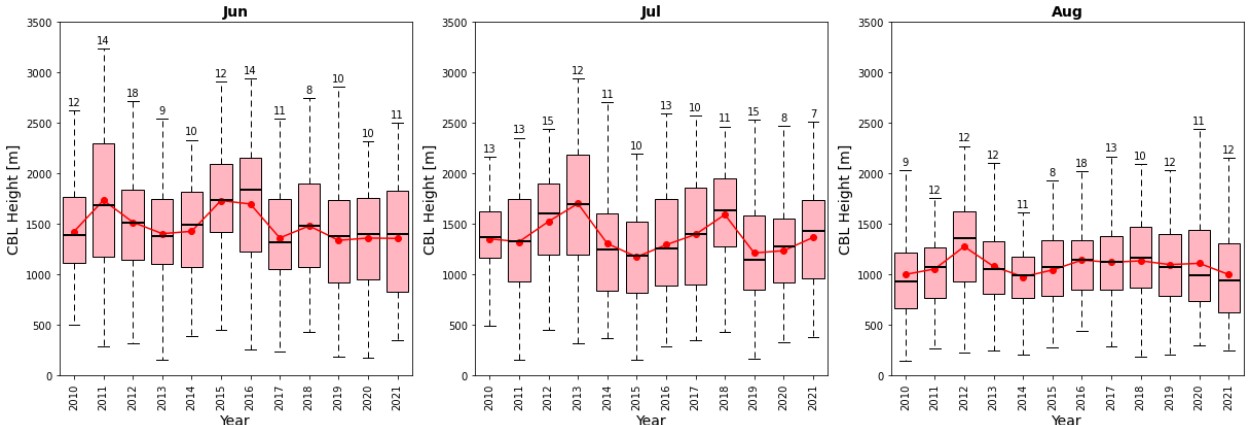





**Figure I1. Box and whisker plot of yearly daytime (12:00–16:00 LT averaged) convective boundary layer (CBL) height from ERA5 for each summer month. The box denotes the interquartile range (IQR), showing the median with a thick black line. The whiskers range from $Q_1 - 1.5 \times IQR$ to $Q_3 + 1.5 \times IQR$, with $Q_1$ and $Q_3$ being the 25th and 75th percentiles. The red line is the monthly-**
**mean. Numbers above each box indicate the sample size or the number of available days for analysis in that month.**

The cumulative daytime sensible heat flux at 52 m from ZOTTO and the ERA5 surface heat flux exhibit similar interannual variability, with no significant long-term trend detected (Fig. I2). However, notable month-to-month differences in magnitude are observed between the two datasets. In June, ERA5 consistently overestimates the observed sensible heat flux. In July, both datasets align closely, showing similar median values, interquartile ranges, and interannual variability. In August, the largest

discrepancies occur, with ERA5 underestimating the observed flux, while observations display greater variability and higher extreme values. A distinct anomaly was observed in 2012, when both datasets showed daytime sensible heat flux reached 1400-2000 W m$^{-2}$ across all three summer months.

The divergence at 750 hPa from ERA5 show no significant trends and minimal variation over the 2010-2021 period, indicating the absence of a long-term change in synoptic-scale subsidence over the ZOTTO region (Fig. I3).


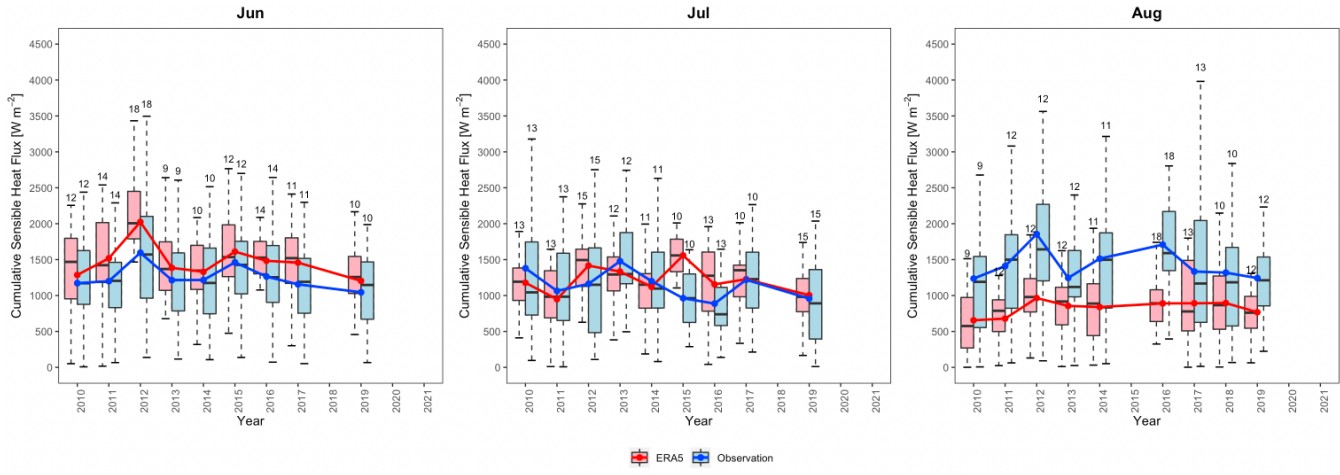

**Figure I2. Box and whisker plot of yearly cumulative positive sensible heat flux of ERA5 and ZOTTO observational measurement at 52 m a.g.l for each summer month. The box denotes the interquartile range (IQR), showing the median with a thick black line. The whiskers range from $Q_1 - 1.5 \times IQR$ to $Q_3 + 1.5 \times IQR$, with $Q_1$ and $Q_3$ being the 25th and 75th percentiles, respectively. The**
**line plots are the monthly mean. Numbers above each box indicate the sample size or the number of available days for analysis in that month.**



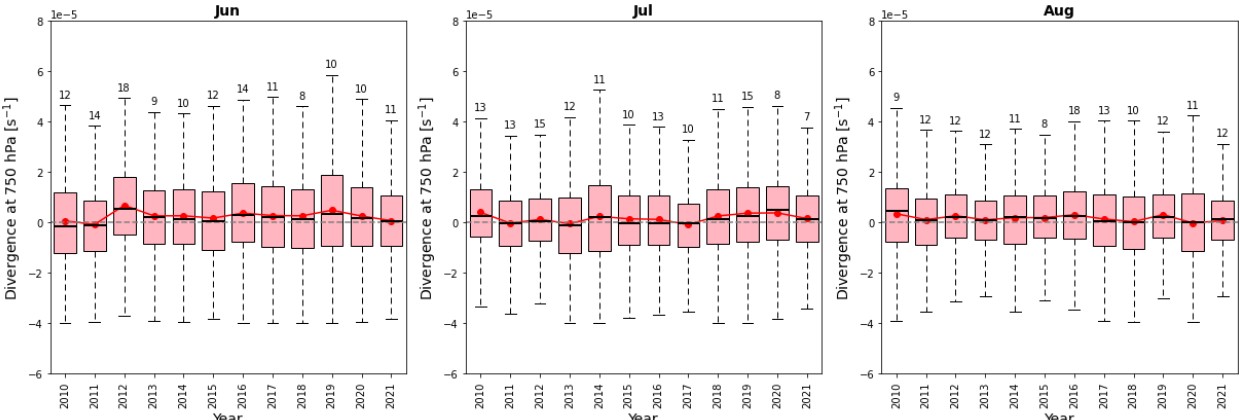

**Figure I3. Box and whisker plot of yearly divergence at 750 hPa from ERA5 for each summer month. The box denotes the interquartile range (IQR), showing the median with a thick black line. The whiskers range from $Q_1 - 1.5 \times IQR$ to $Q_3 + 1.5 \times IQR$, with $Q_1$ and $Q_3$ being the 25th and 75th percentiles, respectively. The blue line is the monthly mean. Numbers above each box indicate the sample size or the number of available days for analysis in that month.**

**Data Availability**

The $CH_4$ atmospheric mole fractions are available on request at [https://doi.org/doi:10.17617/3.JOY5D5.]. This doi will be published upon the acceptation of the manuscript. More information can be given by Dieu Anh Tran (atran@bgc-jena.mpg.de).

**Acknowledgement**

We acknowledge funding by the Max Planck Society to support the installation and maintenance of the ZOTTO until Feb. 2022. DAT acknowledges support from the International Max Planck Research School for Global Biogeochemical Cycles (IMPRS). For servicing the installed setup at the ZOTTO station we deeply appreciate the work of J. Winderlich, A. Panov, Anatoly Prokushkin, and their colleagues from the Sukachev Institute of Forest in Krasnoyarsk, Russian Federation. Technical assistance to the upkeep of the instrumentation at MPI-BGC is also acknowledged, specifically J. Lavric, T. Seifert, S. Schmidt, U. Schultz, R. Leppert, and Karl Kübler. Authors also thank Martin Heimann and Abdullah Bolek at MPI-BGC/BSI for their valuable comments and suggestions which helped us to improve this manuscript.

**Author contribution**

DAT, JV, ITL and SZ designed the study. SZ, CG and IL advised on the long-term and seasonal analysis. JV, ITL and KF advised on the diurnal analysis. JV, CG and SB advised on the calculation of nighttime net surface flux. CG and MG provided the derived vertical wind component from ECMWF. DAT carried out data curation and analysis. DAT and SZ prepared the manuscript with contributions from all co-authors.



## Competing interests

At least one of the (co-)authors is a member of the editorial board of Atmospheric Chemistry and Physics.

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

ǀ110