# Peer review of "Increasing Diurnal and Seasonal Amplitude of Atmospheric Methane Mole Fraction in Central Siberia between 2010-2021"

_EGUsphere, 2025_

## Referee Comment (RC1)

Review on Tran et al., egusphere-2025-2351

**General comment**
The study examines atmospheric $CH_4$ trends observed at the ZOtino Tall Tower Observatory (ZOTTO) during 2010-2021. The site has an excellent decadal observations of the atmospheric composition, and the authors study long-term trends on various aspects, including background level, seasonal and diurnal cycle amplitudes and summer night time mole fractions. The authors have also developed a method to calculate summer night time $CH_4$ fluxes from the vertical gradient of atmospheric $CH_4$ mole fraction observations, which I found unique. Not may observations over Siberia has such a long-term continuous observation of atmospheric $CH_4$, and ZOTTO data play really important role in understanding the effect of climate change in this region. The study is based heavily on the observed data, and it is amazing to see how much information you can obtain from the wealthy of the dataset. This requires effort in maintaining the observation site, which is not easily reachable, and continuos data quality control. I appreciate your team effort.

The study is worth of publication, but I would like to rise a few questions and comments to be addressed.

1. What derives strong GR in 2014 and 2019-2021? In the discussion, you mentioned about 2012, 2016 and 2019, but how about 2014 and 2020-2021? Why are those years have strong GR above MBL? Did you find anything particular about those years in your seasonal/diurnal cycle analysis? If not, why do you think you did not see it?

2. I am slightly surprised to see that June is included as "summer" when atmospheric $CH_4$ mole fractions are rather low (e.g. Fig. 5). To examine "the potential factors that might contribute to this observed increase in the late summer peak unique to ZOTTO (L370-371)", I feel it would be more suitable to use e.g. August-October rather than including low concentration months. For some years, the annual minima occur in July even. Have you checked trends in the diurnal cycles (and night time atmospheric $CH_4$ mole fractions) for each month separately? Do they look similar to those using all the summer months?
2.1. Related question: from which months can you calculate the nighttime $CH_4$ fluxes? If possible, I would like to see trends in fluxes also for other periods than JJA.

3. I would like see an addition of trend analysis on atmospheric $CH_4$ for all seasons/months months and separately for day and night. You may have done this, but without knowing the results, it is slightly hard to believe the increase in the seasonal cycle amplitude is associated with increase in summer atmospheric $CH_4$ only. What is the trend for low-concentration months (spring)? Do you see e.g. downward trend, in which case could also contribute to increase in the seasonal cycle amplitude. In addition, the long-term trend could be assessed using winter months, too, from which you could possibly speculate about global background or anthropogenic emission trends (although you mention that it is unlikely that ZOTTO measures anthropogenic signals).

4. Regarding the seasonal cycle analysis, not only the amplitude, but I would also like to see discussion on timing of seasonal minima and maxima. What could be the reasons for differences in timing of seasonal minima and late summer peaks between different years? In comparison to MBL, it is also noticeable that the ZOTTO data often have seasonal minima earlier than MBL, and seasonal maxima earlier than MBL. Can you speculate why?

5. You have almost discarded the effect of OH in atmospheric $CH_4$ trends. Some studies report decline in atmospheric sinks over the decades and some specific years (e.g. 2020 due to

Covid-19 pandemic). I understand that you could not include the effect in your results, but strongly recommend to add discussion on this point.

6. It is worrying to see that summer daytime atmospheric $CH_4$ show much weaker trends compared to the nighttime considering that many atmospheric inverse models discard nighttime data in they assimilations and the flux estimates are based only on the daytime data. Do you think this could be a reason why inversion results have not been reporting strong increase in $CH_4$ fluxes over northern high latitudes, i.e. they maybe underestimating the flux trend? If you could add thoughts regarding this point, it would be appreciated.

**Specific comments**
L136: "monthly bins and selected the lowest 10-30% of values within each bin."
Could you provide justification for the choice of this filtering method? I think this filtering approach works for $CH_4$ considering that there is no strong sink to the ground. Do you think the approach is still acceptable considering the environment around the tall tower? Have you considered using other approaches such as those used to defined marine boundary layer data, i.e. based on meteorological conditions? I suppose results do not alter significantly by using different methods, but would like to know how you come up with this filtering method, including the choices of the bin size (monthly) and criteria (10-30%).

L335-338 and Table 1:
The GR for 2021 is also very high. Why not to mark it as red in the Table, and modify text indicating that the strong growth of $CH_4$ continued to 2021? About comparison to MBL, I am a bit confused about the last sentence in P13. The GR in the MBL is higher than ZOTTO for 2019 and 2021. Here, I think you have two things to think: the actual values of the GR and the rate of change compared to previous year's GR. Please consider rephrasing/adding arguments about those two points clearly separated. In addition, you focus on the high GR, but not the low ones. Why?

L367: "which is absent at MBL"
I would not say it is absent. It is not as clear as the ZOTTO data, but MBL data peaks could be found in September or October.

L369: "the late summer (August) maximum and the seasonal minimum (during May-July period)"
For some years, the late summer maximum seems to happen in other months than August. Why did you fix the month to calculate maximum while the month for minimum varies? Would your results in Figure 7 change if you use yearly varying months for the late summer maximum?

L410-411: "with the highest values occurring in August (Fig. 10), coinciding with the observed late-summer $CH_4$ peak (Fig. 5)"
Did high pressure days happen mostly in August or rather equally spread during summer?

L426-432:
But why are those affecting nighttime fluxes only? Do they also affect daytime fluxes? I know you were not able to calculate the fluxes from your data, but do you assume that $CH_4$ fluxes have also strong diurnal cycle? If not, why are the daytime mole fractions showing weaker increase?

L470-481 says anthropogenic influence is limited, while L488-489 says ZOTTO data is subject to contribution of fossil fuels also. Those contradicting arguments are slightly confusing.

L541-544:

Did you also see high mole fraction values in June 2016 at ZOTTO? Were temperatures high or soil moisture low then?

Equations: Please consider removing braces brackets and associated symbols/letters within the equation as they do not have mathematical meanings. I think it is clear from the text which part of the equation you refer to. For equations 4 and 7, you could add extra equations to define $F_{Eddy}$, $F_{Stor}$ and the correction factor within each equation.

Figure 1:
- Why the net surface flux of $CH_4$ and  and entrainment flux happen twice a day? It's a bit unclear why they are there. Instead, how about illustrating them three times a day, before sunrise, after sunrise and before sunset (mid-day) and after sunset?
- Could you change the colour of h, so that it is red when h=NBL and blue when h=CBL? Maybe the top of residual layer can be illustrated in different line type so that it is not mixed with ABL?
- It would be informative to add "measurement height" so that it is clear which layer the mole fractions in the top panel is representative of. Addition information in the caption would also be sufficient that from which layer you expect to see this kind of diurnal cycle.
- Although the terms are explained in Eq. (1), please also add explanations of w' and Φ' in the caption as well – the caption should be self-understandable without reading the main text.
- Please consider revising the y-label of the top panel as "Mole Fractions ($\phi$)"

Figure 7: Could you add marks on months when you consider them to be seasonal minima and maxima?

Figure 10: Why Net surface flux are not calculated for some years?

**Technical comments**
L23-25: Please add references to the numbers given.

Section 2.2.2
What is the data frequency of soil measurements and precipitation?

L169-171: "the ($\phi$) refer to $CH_4$ mole fraction, the overbars … wind speed w."
Could you consider modifying it as "the $\phi$ refer to $CH_4$ mole fraction, w to vertical wind speed, h to xxxx, ct to xxx, t to xx, $S_\phi$ to xxx. The overlines ($\bar{\ }$) refer to 30 min. time averaged values, and the prime (′) representing the deviations from the mean."

L171: "All symbols and their corresponding units in this study are provided in Appendix F – Table F1."
- Yes, but even if you have it in Appendix, I think it is very helpful to have definitions also in the main text for the first time. It's strange that some symbols are explain, but not all.
- Although it seems obvious, please add explanation for t, which appears in the derivative of left hand side $\partial$t).

P7 bullet points I-V: There are references to Figure 1, but instead, Equation 1 should be a primary reference. Please add reference to Eq. 1 for IV and V.

End of Section 2.4: In Fig 1 caption, it is said that the atmospheric sink is also not included in this study. Please also add the statement in the main text.

Figure H3: Please add units for the heights in the caption.

---

## Referee Comment (RC2)

**Review Report for EGUsphere-2025-2351**

**General Comments**

The manuscript presents a valuable analysis of long-term methane variability in Central Siberia using ZOTTO tower data, focusing on diurnal and seasonal amplitudes. The topic is timely and relevant, given the role of boreal wetlands in global methane budgets and climate feedbacks. Overall, complex issues related to atmospheric dynamics are well described and the thinking is easy to follow. The paper is generally well written, and the figures are informative. However, even to have a greater impact, the study would benefit from clearer articulation of its objectives, improved transparency regarding methodological novelty, and a more comprehensive discussion of the implications of the findings. Additionally, some interpretations require clarification, and several technical issues need attention.

**Specific Comments**

The introduction gives a lot of background, but the main goal of the study is not clearly stated. Especially, I would like to see more discussion about how the diurnal cycle of atmospheric methane has been used in previous studies and why it is studied here.

It is not fully clear which parts of the method are new and which follow earlier work (e.g., Winderlich et al., 2014). Please make this clear in Section 2.

The paper shows that nighttime fluxes have increased. Does this also mean daytime fluxes have changed? Please discuss this.

L220: "Fits with an $R^2$ value greater than 0.7 were retained.": Did this threshold value exclude a lot of data?

L232-233: "By examining interannual variations in the summer NBL height, we can assess whether the nighttime stability leading to accumulation of near-surface $CH_4$ mole fractions have strengthened or weakened over time." Could you clarify what it would indicate if nighttime stability strengthened or weakened over time?

L335-338: Could you explain why the year 2021 is not mentioned? It has the highest annual growth rate in the MBL data and also high for the ZOTTO data.

L390-393: The detrending was done based on daytime data, right? So, in some sense, you removed the daytime trend, and it is no wonder that the trend in daytime values is close to zero after that. Or maybe the point is to show that the trend in the nighttime values is larger

than in daytime values, but that is already stated. Please, consider phrasing this paragraph differently.

L426-433: It would be nice if you explain earlier or somewhere how these environmental factors are expected to affect methane fluxes.

L451-456: There are also remote sensing products that have been studied near the ZOTTO station that could be a suitable reference or comparison. (e.g. Kivimäki et al. 2025).

L460-468: How about snow melting? How would that affect methane fluxes near ZOTTO? When does snow melt in ZOTTO? Air temperature seems to rise above zero in April-May, so I would assume that the snow melts after that.

L489-490: "These regional emissions may have contributed to enhanced seasonal amplitude observed at ZOTTO." Do you think that the enhanced seasonal amplitude is due to winter maxima being higher or summer minimum being lower?

L520-522: "These findings align with studies from other boreal and wetland-dominated regions, where $CH_4$ emissions peak in late summer due to sustained high soil temperature and moisture leading to high microbial activity." Could you expand on this point? For example, Rößger et al. (2022) also reported the highest methane emissions in August, but the long-term increase differs: Rößger et al. found the trend in July, while your results suggest the increase occurs in August.

**Technical Corrections**
It would be nice if the figures and Appendices mentioned would be in order, i.e., that the first mentioned would have A1, next one A2 or B1 etc.

L22: "global greenhouse **gases** radiative forcing" -> "global greenhouse **gas** radiative forcing"

L24: "has risen" -> "rose"

L25: Reference for the atmospheric methane levels missing.

L31: The global methane budget has been updated (Saunois et al. (2025)), you could also update this reference.

L34-35: Do the area of undisturbed and disturbed boreal and temperate peatlands overlap with the area of permafrost region? I.e., can you just sum up the two numbers mentioned in the text to get peatland emission from northern peatlands?

L90: "Data were recorded every 30 seconds from each sampling line.": What does the "30 seconds" refer to exactly? Does it mean that there is a measurement taken every 30 seconds from the currently active sampling lines?

L185: "as mentioned in" -> "similarly to" or "following…"

L208: Was the ERA5 reanalysis data you used "boundary layer height"?

L301: "Equation 6 as in Winderlich et al., 2014 can be expanded" -> "Equation 6, as in Winderlich et al., (2014), can be expanded"

L303: "Where" -> "where"

L384: "Between 2010 and 2021, the summer diurnal amplitude increased significantly at p = 0.01 level at a rate of 5.55 ppb yr$^{-1}$ (p = 0.002)": Maybe you could form it a bit differently: "Between 2010 and 2021, the summer diurnal amplitude increased significantly (p < 0.01) at a rate of 5.55 ppb yr$^{-1}$ (p = 0.002)."

L405-415: You could first present the results and then conclude that based on that increasing nighttime $CH_4$ surface fluxes during summer are the primary driver of the observed rise in nighttime $CH_4$ mole fractions.

L435, Figure 11: Why is the x-axis of VPD flipped?

L453: "signal has" -> "signal had"

L474: "natural gas emissions sum up to 1 to 10 % of the overall wetland emissions only". Which area and time period is being referred to here?

L590, Figure A1: Consider using a "neutral" (neither blue or red) colour at 0 °C in the colour bar to make it easier to see where the air temperature crosses the freezing point.

References:

Kivimäki, E., Aalto, T., Buchwitz, M., Luojus, K., Pulliainen, J., Rautiainen, K., Schneising, O., Sundström, A.-M., Tamminen, J., Tsuruta, A., and Lindqvist, H.: Environmental drivers

constraining the seasonal variability of satellite-observed methane at Northern high latitudes, EGUsphere [preprint], https://doi.org/10.5194/egusphere-2025-249, 2025.

Rößger, N., Sachs, T., Wille, C. *et al.* Seasonal increase of methane emissions linked to warming in Siberian tundra. *Nat. Clim. Chang.* **12**, 1031–1036 (2022). https://doi.org/10.1038/s41558-022-01512-4

Saunois, M., Martinez, A., Poulter, B., Zhang, Z., Raymond, P. A., Regnier, P., Canadell, J. G., Jackson, R. B., Patra, P. K., Bousquet, P., Ciais, P., Dlugokencky, E. J., Lan, X., Allen, G. H., Bastviken, D., Beerling, D. J., Belikov, D. A., Blake, D. R., Castaldi, S., Crippa, M., Deemer, B. R., Dennison, F., Etiope, G., Gedney, N., Höglund-Isaksson, L., Holgerson, M. A., Hopcroft, P. O., Hugelius, G., Ito, A., Jain, A. K., Janardanan, R., Johnson, M. S., Kleinen, T., Krummel, P. B., Lauerwald, R., Li, T., Liu, X., McDonald, K. C., Melton, J. R., Mühle, J., Müller, J., Murguia-Flores, F., Niwa, Y., Noce, S., Pan, S., Parker, R. J., Peng, C., Ramonet, M., Riley, W. J., Rocher-Ros, G., Rosentreter, J. A., Sasakawa, M., Segers, A., Smith, S. J., Stanley, E. H., Thanwerdas, J., Tian, H., Tsuruta, A., Tubiello, F. N., Weber, T. S., van der Werf, G. R., Worthy, D. E. J., Xi, Y., Yoshida, Y., Zhang, W., Zheng, B., Zhu, Q., Zhu, Q., and Zhuang, Q.: Global Methane Budget 2000–2020, Earth Syst. Sci. Data, 17, 1873–1958, https://doi.org/10.5194/essd-17-1873-2025, 2025.

---

## Author Comment (AC1)

**Responses to Reviewer #1**

**General comment**

The study examines atmospheric CH4 trends observed at the ZOtino Tall Tower Observatory (ZOTTO) during 2010-2021. The site has an excellent decadal observations of the atmospheric composition, and the authors study long-term trends on various aspects, including background level, seasonal and diurnal cycle amplitudes and summer night time mole fractions. The authors have also developed a method to calculate summer night time CH4 fluxes from the vertical gradient of atmospheric CH4 mole fraction observations, which I found unique. Not may observations over Siberia has such a long-term continuous observation of atmospheric CH4, and ZOTTO data play really important role in understanding the effect of climate change in this region. The study is based heavily on the observed data, and it is amazing to see how much information you can obtain from the wealthy of the dataset. This requires effort in maintaining the observation site, which is not easily reachable, and continuos data quality control. I appreciate your team effort.

The study is worth of publication, but I would like to rise a few questions and comments to be addressed.

Thank you for the constructive review and suggestions. We detail our responses below in blue.

1. What derives strong GR in 2014 and 2019-2021? In the discussion, you mentioned about 2012, 2016 and 2019, but how about 2014 and 2020-2021? Why are those years have strong GR above MBL? Did you find anything particular about those years in your seasonal/diurnal cycle analysis? If not, why do you think you did not see it?

As noted briefly in L441-449, the persistent increase in CH4 mole fractions at ZOTTO in 2014 and 2019-2021 aligns closely with the global trend reported by NOAA and other long-term monitoring networks (Lan et al., 2021). In the revised manuscript, the growth rate of 2016, and 2021 will also be highlighted in Table 1. We will modify L441-444 as:

"The observed persistent increase in CH4 growth rates at ZOTTO is consistent with the global trend reported by NOAA and other long-term monitoring networks (Lan et al., 2021). Peaks in CH4 growth at ZOTTO, particularly in 2014, 2016, and 2019-2021, mirror global trends of increasing CH4 levels, which began around 2014, steepened after 2018, and further intensified in 2020 (Worden et al., 2017; Nisbet et al., 2019). The global CH4 increases in 2014 and 2020 have been largely attributed to reductions in atmospheric OH radical concentrations (Zhang et al., 2021). In 2020, this OH-driven effect was likely amplified by decreases in anthropogenic NOx emissions and the associated reduction in free-tropospheric ozone during the COVID-19 lockdowns (Peng et al., 2022). High global growth rates from 2016 to 2020 were additionally driven by strong emissions from boreal wetlands in Eurasia (Yuan et al., 2024; Zhang et al., 2021). A comparison of CH4 mole fraction time series between the inland tall tower at ZOTTO and the *MBL* reference at 60° N shows consistently higher CH4 concentrations at ZOTTO. However, only in some of the peak years highlighted in red in Table 1 (i.e., 2014, 2016, and 2020) does ZOTTO exhibit annual growth rates exceeding those of the MBL, suggesting that, in addition to the global baseline increase, ZOTTO may have been influenced by additional regional sources in those vears."

We will add additional seasonal amplitude analysis for these years after L481 as below:

"Among the years with high CH4 growth rates (highlighted in red in Table 1), the seasonal amplitude at ZOTTO remained relatively small in 2014, 2016, and 2020. During these years, elevated CH4 mole fractions were distributed relatively uniformly across seasons, resulting in consistent increases throughout the year. This pattern enhanced the annual mean CH4 while leaving the winter-spring amplitude unchanged. Notably, these are the same years in which ZOTTO growth rates exceeded those of the MBL. In contrast, 2019 and 2021 exhibited both strong growth rates and enhanced seasonal amplitudes, particularly driven by elevated winter CH4 mole fractions. The underlying causes of these patterns remain uncertain. To address this, future studies should employ atmospheric inversion techniques, which integrate CH4 observations with atmospheric transport models to estimate fluxes. Analysing these inverted fluxes would help identify the regional sources responsible for the enhanced CH4 levels that increased the ZOTTO growth rate in 2014, 2016 and 2019-2021, as well as clarify the drivers behind the elevated winter CH4 observed in 2019 and 2021."

2. I am slightly surprised to see that June is included as "summer" when atmospheric CH4 mole fractions are rather low (e.g. Fig. 5). To examine "the potential factors that might contribute to this observed increase in the late summer peak unique to ZOTTO (L370-371)", I feel it would be more suitable to use e.g. August-October rather than including low concentration months. For some years, the annual minima occur in July even. Have you checked trends in the diurnal cycles (and night time atmospheric CH4 mole fractions) for each month separately? Do they look similar to those using all the summer months? 2.1. Related question: from which months can you calculate the nighttime CH4 fluxes? If possible, I would like to see trends in fluxes also for other periods than JJA.

Individual plots of the trends of diurnal cycles for each month of the year are plotted in Fig. R1-R3. Only July to October (Fig. R3) show statistically significant increasing trends (p < 0.05) in their diurnal amplitude driven by increasing in nighttime mole fraction, while the other months do not. We further examined the CH4 nighttime fluxes for the months in which there is a statistically significant increasing trend in the diurnal amplitude (July to October) (Fig. R4) and will update our diurnal amplitude and nighttime fluxes analysis on July-October period instead of June-August in the revised manuscript.

Our analysis of the long-term trend in nighttime CH4 flux calculation can only be analysed during warmer months (June to October), since it requires vertical profiles temperature and sensible heat flux, which are not fully available in the other months due to technical limitations at ZOTTO. Strong icing of the wind and temperature sensors prevents reliable flux estimates during the other colder months.

Figure R1. Timeseries of averaged CH4 diurnal cycle amplitude (left column); its daytime (10:00-16:00 LT averaged) CH4 mole fraction (middle column), and its nighttime (00:00-04:00 LT averaged) CH4 mole fraction (right column) at ZOTTO using detrended 52 m a.g.l. data in (a) November; (b) December; (c) January and (d) February. The Theil-Sen regression trend is depicted by the solid line, with the 95 % confidence interval of the trend shown as dashed lines. The p-value indicates whether the slope of the regression is significantly different (at 0.05 level) from zero.

Figure R2. Same as Fig. R1 but for March-June.

Figure R3. Same as Fig. R1 but for July to October.

Figure R4. Box and whisker plot of yearly nighttime (00:00-04:00 LT) net CH4 flux for each month from July to October. The box denotes the interquartile range (IQR), showing the median with a thick black line. The whiskers range from  $Q_1 = -1.5 \times IQR$  to  $Q_3 = +1.5 \times IQR$ , where  $Q_1$  and  $Q_3$  are the 25th and 75th percentiles, respectively. The blue line is the monthly mean. The Theil-Sen trend slope for the mean and its p-value are denoted on the top left corner of each plot. Numbers above each box indicate the number of available days (based on the number of high-pressure days and the availability of vertical profile (4, 52 and 92 m a.g.l) of CH4, potential temperature and sensible heat flux at 52 m) for analysis in that month.

3. I would like see an addition of trend analysis on atmospheric CH4 for all seasons/months months and separately for day and night. You may have done this, but without knowing the results, it is slightly hard to believe the increase in the seasonal cycle amplitude is associated with increase in summer atmospheric CH4 only. What is the trend for low-concentration months (spring)? Do you see e.g. downward trend, in which case could also contribute to increase in the seasonal cycle amplitude. In addition, the long-term trend could be assessed using winter months, too, from which you could possibly speculate about global background or anthropogenic emission trends (although you mention that it is unlikely that ZOTTO measures anthropogenic signals).

In our study, the seasonal cycle amplitude is defined as the difference between the winter maximum and the spring minimum (or the annual minimum) of CH4 mole fractions. The late-summer CH4 peak amplitude is calculated as the difference between the late-summer maximum (typically fall between late July-mid October) and the spring minimum.

Additional trend analyses of atmospheric CH4 for all months, separately for daytime and nighttime, are presented in Figures R1-R3. These results show significant increasing trends (at 0.05 level) in diurnal amplitude during July to October, driven mainly by a significant increase in nighttime CH4 mole fractions, while daytime concentrations remain relatively stable. During the spring period (mid-May to late June), which typically corresponds to the timing of the seasonal minimum, only May shows a significant decreasing trend in both daytime and nighttime CH4. Although nighttime CH4 decreases slightly faster than daytime CH4, this difference is insufficient to produce a significant decreasing trend in the diurnal amplitude for May.

On the seasonal scale, the results (Fig. R5) indicate an increasing trend in the winter CH4 maximum at ZOTTO, although this trend is not statistically significant (Fig. R5). The CH4 annual minima also exhibits a statistically insignificant decreasing trend, but this trend is smaller compared to the increase in the winter maximum. The increase in the ZOTTO seasonal amplitude (Fig. 6 in the manuscript), while also not significant, appears to be mainly driven by this rise in winter CH4 mole

fraction maxima accompanied by the slight decrease in the CH4 spring minima. This may reflect either a larger influence of regional wetland and/or anthropogenic emissions during winter or an increased occurrence of persistent, stable high-pressure systems, which can enhance the accumulation of CH4 near the surface under limited atmospheric mixing conditions. To further investigate the contributors to the increasing winter CH4 mole fractions at ZOTTO, further studies using atmospheric inversion techniques and isotopic analysis are required, which is outside the scope of this study.

There is a significant increasing trend in the late-summer CH4 peak (1.35 ppb year-1, p = 0.02) (Fig. R5). Therefore, it is reasonable to conclude that the observed increase in the late-summer CH4 peak amplitude (as shown in Fig. 7 in the manuscript) is primarily driven by the significant rise in the late-summer CH4 peak rather than by a decrease in springtime minima.

In the revised manuscript, Fig. R5 will be added in the appendix, and additional analyses will be added after L358:

"The increase in the ZOTTO seasonal amplitude (Fig. 6), while not statistically significant, appears to be mainly driven by an increasing winter  $CH_4$  mole fraction maximum (1.42 ppb year  $^1$ , p = 0.17) accompanied by a slight decrease in the  $CH_4$  spring minimum (-0.69 ppb year  $^1$ , p = 0.32) (Appendix – Fig...)."

**And after L370:**

"The enhanced late-summer peak amplitude (Fig. 7) is primarily attributed to the strong significant increase (1.35 ppb year-1, p = 0.02) in the late-summer CH4 maximum, rather than to a slight decrease in the spring minimum (-0.69 ppb year-1, p = 0.32) (Appendix – Fig.)"

Figure R5. Time series of the CH4 spring minimum, winter maximum and late summer maximum for detrended background-filtered daytime CH4 at ZOTTO (ZOTbg). The Theil-Sen regression trend is depicted by the solid line, with the 95 % confidence interval of the trend shown as dashed lines. The p-value indicates whether the slope of the regression is significantly different (at 0.05 level) from zero.

4. Regarding the seasonal cycle analysis, not only the amplitude, but I would also like to see discussion on timing of seasonal minima and maxima. What could be the reasons for differences in timing of seasonal minima and late summer peaks between different years? In comparison to MBL, it is also noticeable that the ZOTTO data often have seasonal minima earlier than MBL, and seasonal maxima earlier than MBL. Can you speculate why?

We have added the timeseries analysis of the timing of the spring minimum, winter maximum, and late-summer peak in Figure R6. The results show that the seasonal minima occur between mid-

May and early July, the winter maxima between mid-December and early March, and the late-summer peaks between late July and mid-October. All three variables show decreasing trends in timing, indicating that these seasonal events are occurring earlier in the year. However, only the trend in the timing of the late-summer maximum is statistically significant (Fig. R6). The trends observed in the winter maximum and spring minimum may reflect limitations of the curve-fitting method used in this study rather than genuine changes in the observations. The Thoning et al. (1989) approach, like other harmonic-based fitting methods, assumes that the overall shape of the seasonal cycle remains relatively constant over time. This makes it less capable of accurately representing non-stationary processes, where the seasonal phase changes due to varying environmental conditions (e.g. heatwaves, drought). As a result, when the method detects a timing shift in one part of the cycle (in this case, an earlier onset of the late-summer peak), it tends to apply that same shift uniformly across the entire seasonal pattern. Consequently, the changes in the winter maximum or spring minimum may not reflect a true physical trend, but rather an artifact of the model's assumption that the whole seasonal cycle has shifted equally in time.

To explore the potential drivers of the timing changes in the late-summer peaks, we examined the relationship between the day of year (DOY) of the late-summer maximum and several environmental variables, including, precipitation, air temperature and vapor pressure deficit (both at 52 m a.g.l.), as well as soil moisture and soil temperature at 32 cm b.g.l, and the preceding Feb-May averaged snow depth (from ERA5 reanalysis at the grid point closest to ZOTTO,  $60^{\circ}75'$  N,  $89^{\circ}25'$  E). Among these variables, only soil temperature showed a significant (p < 0.05) negative correlation (Fig. R7), suggesting that higher soil temperature not only enhances the late-summer CH4 flux (as shown in Fig. 11 of the manuscript) but also potentially could advance the timing of the late-summer CH4 peak.

However, because the curve-fitting approach has methodological limitations that make it difficult to determine whether the observed trend in Fig. R6 reflects a genuine signal or an artifact of the fitting process, we decided not to include the timing results in the main manuscript. We believe that this topic needs another dedicated study in which variations in timing can be evaluated using different curve-fitting methods (e.g. HPspline), parameter settings of each curve fitting method (as suggested in Pickers et al., 2015), and alternative definitions of seasonal phases and timing based on flux proxies, as demonstrated in Tran et al. (2024) and Kariyathan et al. (2023) for CO2.

A noticeable time shift exists between the annual minima observed at ZOT and MBL, with the minimum at ZOT occurring later in the year. The following explanation will be added to the revised manuscript in L348:

"This lag likely results from the atmospheric transport of  $CH_4$ -depleted air masses originating over the continents and moving toward the ocean. A similar pattern has been reported in the ZOTTO  $CO_2$  record (Tran et al., 2024) when compared with the MBL reference data, supporting the influence of large-scale air mass transport on the observed seasonal timing."

Figure R6. Trend analysis of the annual minimum, winter maximum, and late-summer maximum timings. Timings are calculated as the number of days since January 1 of the given year, with negative values occurring in winter maximum generally representing dates in December of the preceding year.

R7. Relationship between the timing (Day of Year (DOY)) of the late summer maximum and soil moisture measured at 32 cm b.g.l (%). Shaded areas represents 95 % confidence intervals.

5. You have almost discarded the effect of OH in atmospheric CH4 trends. Some studies report decline in atmospheric sinks over the decades and some specific years (e.g. 2020 due to Covid-19 pandemic). I understand that you could not include the effect in your results, but strongly recommend to add discussion on this point.

The impact of decreasing OH concentrations on the elevated atmospheric CH4 growth rates in certain years (i.e., 2014 (Zhang et al., 2021) and 2020 (Peng et al., 2022)) will be discussed in

more detail in Section 3.1 of the revised manuscript. This addition is also addressed in our answer to Question 1 above.

6. It is worrying to see that summer daytime atmospheric CH4 show much weaker trends compared to the nighttime considering that many atmospheric inverse models discard nighttime data in they assimilations and the flux estimates are based only on the daytime data. Do you think this could be a reason why inversion results have not been reporting strong increase in CH4 fluxes over northern high latitudes, i.e. they maybe underestimating the flux trend? If you could add thoughts regarding this point, it would be appreciated.

Indeed, all CH4 inversion systems primarily assimilate observations collected during daytime well-mixed periods. These measurements are less affected by local transport uncertainties and therefore capture larger-scale regional signals, but they are also less sensitive to local flux variability. Nighttime observations are excluded from inversions due to increased atmospheric transport uncertainties under stable boundary layer conditions, when vertical mixing is weak and dominated by intermittent or the absence of turbulence that is difficult to parameterise (e.g., Kurzeja et al., 2022).

As shown in our analysis, for a high-latitude station, summer daytime CH4 mole fractions exhibit much weaker increasing trends compared to nighttime measurements during the 2010-2021 period. This discrepancy might indeed suggest that the exclusion of nighttime data in inversion systems could lead to a systematic underestimation of CH4 emission trends, particularly over the northern high latitudes.

To address this issue, improved representations of the nocturnal boundary layer (NBL) and higher-resolution transport models capable of accurately simulating stable boundary layer dynamics are essential. In this study, we also demonstrate how a novel observation-based approach using multi-level tall-tower measurements can be used to estimate NBL heights. These observation-derived NBL estimates could be used by high-resolution transport models to inform and correct the NBL height. Incorporating such improved nighttime transport representations into inversion frameworks would reduce the need to exclude nighttime mole fraction data, enabling use of the complete observational records.

Nevertheless, we note that the weaker CH4 flux trends reported by current inversion studies may not solely be due to the use of daytime-only data. Limited spatial and temporal coverage of observations across the northern high latitudes, particularly in regions such as eastern Siberia, also likely contributes to the uncertainty and potential underestimation of high latitude CH4 flux trends.

**Specific comments**

L136: "monthly bins and selected the lowest 10-30% of values within each bin." Could you provide justification for the choice of this filtering method? I think this filtering approach works for CH4 considering that there is no strong sink to the ground. Do you think the approach is still acceptable considering the environment around the tall tower? Have you considered using other approaches such as those used to defined marine boundary layer data, i.e. based on meteorological conditions? I suppose results do not alter significantly by using different methods, but would like to know how you come up with this filtering method, including the choices of the bin size (monthly) and criteria (10-30%).

The primary motivation for our filtering approach is to minimise the influence of sporadic, large positive CH4 anomalies (e.g., local biomass burning events such as the 2012 wildfire) on the long-term growth rate, while still retaining the underlying regional seasonal signal. Without filtering, single extreme events can map into unrealistically strong annual growth-rate anomalies of the region.

We tested both monthly and weekly bin-size and found that monthly bins better smooth out highintensity events such as wildfires that could last for more than a week and reduce the risk of individual events dominating the growth-rate calculation while still being able to capture seasonal influence. Monthly binning provides a balance between temporal resolution and robustness against outliers.

We selected the 10-30 % range to focus on relatively clean background conditions while still retaining enough data points for robust statistics. The lower bound of 10 % (rather than 0 %) helps avoid potential biases in cases of abnormally enhanced springtime CH4 uptake by soils (May-June) (Ranniku et al., 2025). The upper limit of 30 % is chosen to exclude most large, abnormal large emission events such as wildfire. We performed a sensitivity test using alternative percentile ranges (5-10 %, 15-50 %, 10-50 %, 10-30 %) and found that the resulting growth rates for 2012 (for example) have a standard deviation of  $\pm$  0.94 ppb year-1, suggesting that our main conclusions are robust to the choice of thresholds.

We decided not to use other approaches such as those used to define MBL background data. Stations used in the MBL reference are typically at remote marine sea level locations. Background filtering for these stations relies heavily on wind-direction criteria to identify clean air masses (e.g., with prevailing onshore winds). This approach is not directly transferable to a continental site like ZOTTO, where wind-direction filtering does not reliably separate background from local influence. We therefore believe that our quantile-based filtering method is better suited for such a remote, continental setting.

**L335-338 and Table 1:**

The GR for 2021 is also very high. Why not to mark it as red in the Table, and modify text indicating that the strong growth of CH4 continued to 2021? About comparison to MBL, I am a bit confused about the last sentence in P13. The GR in the MBL is higher than ZOTTO for 2019 and 2021. Here, I think you have two things to think: the actual values of the GR and the rate of change compared to previous year's GR. Please consider rephrasing/adding arguments about those two points clearly separated. In addition, you focus on the high GR, but not the low ones. Why?

In the revised manuscript, the growth rate of 2016, and 2021 will also be highlighted. Line 337 will be rewritten as:

"The MBL growth rates show similar temporal patterns to ZOTTO, with notable increases in growth rate values in in 2014, 2016, and 2019-2021."

Regarding the focus on high GR periods rather than the low ones: years with abnormally low GR values, although potentially interesting, are rare at ZOTTO during the 2010-2021 period. With the scientific communities' interest in determining the drivers and reducing the uncertainties of the global CH4 increasing trend, especially in high-latitude regions, we prioritise examining the periods of unusually strong CH4 growth at ZOTTO. These high-GR periods are most relevant for identifying and understanding the potential drivers of enhanced CH4 emissions, such as wetland activity, temperature anomalies, OH reduction or atmospheric circulation patterns.

L367: "which is absent at MBL" I would not say it is absent. It is not as clear as the ZOTTO data, but MBL data peaks could be found in September or October.

The sentence "Notably, ZOTTO displays a secondary peak in late summer during the late summer (August) period, which is absent at MBL" will be revised as:

"Notably, ZOTTO displays a secondary peak in late summer (July-October), whereas at MBL stations this feature is less pronounced, with peaks occurring later in the season, around November-January."

L369: "the late summer (August) maximum and the seasonal minimum (during May-July period)". For some years, the late summer maximum seems to happen in other months than August. Why did you fix the month to calculate maximum while the month for minimum varies? Would your results in Figure 7 change if you use yearly varying months for the late summer maximum?

The analysis presented in Figure 7 was based on identifying the late-summer peak amplitude within the period July to October, rather than being fixed specifically to August. Therefore, the original phrasing in L369 "the late summer (August) maximum" was indeed misleading. We will revise the text to read "the late summer maximum (between July-October)" to more accurately reflect the analysis method. This clarification does not affect the results shown in Figure 7, as it already accounted for variability in the timing of the late-summer maximum (falls between late-July to October) across different years.

L410-411: "with the highest values occurring in August (Fig. 10), coinciding with the observed late-summer CH4 peak (Fig. 5)" Did high pressure days happen mostly in August or rather equally spread during summer?

The distribution of high-pressure days, as shown in Table E1, is relatively even across late summer months, with some interannual variability. We will update our Table E1 to Figure E1 as below for better visualisation. In this study, we have also done additional analysis for NBL height under high-pressure days in Appendix D and found no abnormally low values in August that could lead to increase in accumulation of CH4 emission. We confidently conclude that the highest nighttime flux values occurring in August is due to surface process (i.e. increase in CH4 emission) other than changes in nighttime boundary layer dynamics under high-pressure condition.

L426-432: But why are those affecting nighttime fluxes only? Do they also affect daytime fluxes? I know you were not able to calculate the fluxes from your data, but do you assume that CH4 fluxes have also strong diurnal cycle? If not, why are the daytime mole fractions showing weaker increase?

The diurnal variability of CH4 flux does not follow a consistent pattern. Depending on the season, ecosystem types, and environmental conditions, CH4 fluxes may be higher during the day, lower at night, show the opposite behaviour, or remain relatively constant throughout the day (Kohl et al., 2023). If assuming the case when CH4 fluxes do not exhibit a strong diurnal cycle as you suggested, the daytime net CH4 fluxes (although not calculated in our analysis) could indeed also be correlated with soil temperature and soil moisture, as observed for the nighttime fluxes. However, the weaker increase in daytime CH4 mole fractions compared to nighttime values is likely explained by differences in atmospheric boundary layer dynamics. During nighttime, the stable boundary layer is low (around 100-200 m at ZOTTO as shown in Appendix D) limiting vertical mixing and allowing locally surface CH4 emissions to accumulate and thus produce a stronger mole fraction signal. In contrast, during the daytime, the boundary layer becomes higher and more turbulent, enhancing vertical mixing and diluting surface CH4 emissions within a larger air volume. As a result, even with the same increase surface flux in both daytime and nighttime the daytime CH4 mole fraction changes could appear weaker than those observed at night due to more dilution and mixing.

However, because our current dataset and analysis framework do not provide direct constraints on daytime fluxes, we chose not to include this speculative interpretation in the discussion and suggested future studies to attempt other methods to derive daytime fluxes in L545-556.

L470-481 says anthropogenic influence is limited, while L488-489 says ZOTTO data is subject to contribution of fossil fuels also. Those contradicting arguments are slightly confusing.

We agree that this sentence can potentially lead to confusion. To clarify, in the revised manuscript, lines 488-489 will be updated to:

"In contrast, ZOTTO (and WLG) is located inland and subject to regional influences, including variable contributions from wetlands, fires and to some extend also fossil fuel sources."

L541-544:Did you also see high mole fraction values in June 2016 at ZOTTO? Were temperatures high or soil moisture low then?

Indeed, high CH4 mole fraction values were observed at ZOTTO in June 2016, coinciding with elevated air and soil temperatures (Fig. R8). This supports the statement that "The elevated emissions observed in June 2016 may be linked to increased wetland emissions, driven by the unusually high temperatures of that period." However, we will not include this in our revised manuscript since we will shift our analysis from June-August to July-October.

R8. Box and whisker plot of yearly air temperature at 52 m a.g.l, soil temperature at 32 cm b.g.l, and detrended background-filtered daytime CH4 at ZOTTO (ZOTbg) in June.

Equations: Please consider removing braces brackets and associated symbols/letters within the equation as they do not have mathematical meanings. I think it is clear from the text which part of the equation you refer to. For equations 4 and 7, you could add extra equations to define FEddy, Fstor and the correction factor within each equation.

In the revised manuscript we will remove the braces, brackets, and associated symbols/letters that do not carry mathematical meaning.

**Figure 1:**

- Why the net surface flux of CH4 and and entrainment flux happen twice a day? It's a bit unclear why they are there. Instead, how about illustrating them three times a day, before sunrise, after sunrise and before sunset (mid-day) and after sunset?
- Could you change the colour of h, so that it is red when h=NBL and blue when h=CBL? Maybe the top of residual layer can be illustrated in different line type so that it is not mixed with ABL?

- It would be informative to add "measurement height" so that it is clear which layer the mole fractions in the top panel is representative of. Addition information in the caption would also be sufficient that from which layer you expect to see this kind of diurnal cycle.
- Although the terms are explained in Eq. (1), please also add explanations of w' and  $\Phi$ ' in the caption as well the caption should be self-understandable without reading the main text.
- Please consider revising the y-label of the top panel as "Mole Fractions (φ)".

We thank the reviewer for the detailed suggestions regarding Figure 1. In the revised manuscript, Figure 1 and its caption will be updated as below:

Figure 1. Schematic overview of diurnal cycle of the mole fractions of atmospheric CH4 from the top of a forest canopy (z=ct) to the top of the Atmospheric Boundary Layer (ABL) (z=h) illustrating Eq. (1), adapted from Faassen et al. (2024). The figure illustrates the ABL is characterised by a Convective Boundary Layer (CBL) during daytime and a Nocturnal Boundary Layer (NBL) formation during nighttime.  $\phi$  denotes the CH4 mole fraction, and the overbar ( $\overline{\phi}$ ) represents a 30-minute time average. The prime symbol (') indicates the deviation of the instantaneous CH4 mole fraction from its time average. Similarly, w' represents the deviation of the instantaneous vertical wind speed from its time-averaged mean ( $\overline{w}$ ). The "Net Surface flux of CH4" term ( $\overline{(w'\phi')}_{NetSurf}$ ) refers to the fluxes from the vegetation layer, up to the top of the canopy (z=ct). The fluxes up to this level depend on terrestrial processes, which contribute to the CH4 mole fractions observed above the top of the canopy. Entrainment flux at the top of the ABL ( $\overline{(w'\phi')}_e$ ) represents the mixing of CH4 air from above the ABL, to inside the ABL. The horizontal advection of CH4 (adv( $\phi$ )), and the chemical reaction term ( $S_{\phi}$ ) in Eq. (1) are not included in this figure since they are not currently accounted for in this study.

Figure 7: Could you add marks on months when you consider them to be seasonal minima and

We believed the reviewer meant Figure 5. We updated Fig. 5 and its caption as below:

— zoT(ba) — MBL

Figure 5. The yearly seasonal cycles of background-filtered daytime  $CH_4$  at ZOTTO (ZOTbg) and biweekly marine boundary layer  $CH_4$  at  $60^\circ N$  (MBL), shown after removing long-term trends (i.e., subtracting the trend components presented in Fig. 4b from the data in Fig. 4a). The line plots with circle markers represent the monthly medians. The darker shaded boxes indicate the interquartile range (IQR). The lighter shaded boxes extend from  $Q_1 = -1.5 \times IQR$  to  $Q_3 = +1.5 \times IQR$ , where  $Q_1$  and  $Q_3$  are the 25th and 75th percentiles, respectively. Coloured ticks on the x-axis highlight: dark red (ZOT) and dark blue (ZOT) and dark blue (ZOT) and light blue (ZOT

**Figure 10: Why Net surface flux are not calculated for some years?**

The net surface flux calculations rely on the availability of several key measurements: the vertical profile of CH4 mole fraction and potential temperature at 4, 52, and 91 m a.g.l., and the sensible heat flux at 52 m. For the years where net fluxes were not calculated, one or more of these essential measurements were missing due to instrument malfunctions or data gaps. Specifically, either potential temperature at one of the required heights or the sensible heat flux at 92 m was unavailable, preventing reliable flux estimation for those years. To better clarify this, we will update our captions for Figure 10 with: "Numbers above each box indicate the sample size or the number of available days (based on the number of high-pressure days and the availability of the

vertical profile (4, 52 and 92 m a.g.l) of  $CH_4$ , potential temperature and sensible heat flux at 52 m) for analysis in that month."

Technical comments

L23-25: Please add references to the numbers given.

Citation (Lan et al., 2025) will be added in the revised manuscript.

Section 2.2.2

What is the data frequency of soil measurements and precipitation?

The frequency of soil measurement and precipitation will be added to the revised version: "Vertical profiles of soil temperature (°C) (measured at depths of 2, 4, 6, 8, 16, 32, 64, and 128 cm), soil moisture (%) (at 8, 16, 32, 64, and 128 cm), and precipitation are also recorded every 30 min."

L169-171: "the (φ) refer to CH4 mole fraction, the overbars ... wind speed w."

Could you consider modifying it as "the  $\phi$  refer to CH4 mole fraction, w to vertical wind speed, h to xxxx, ct to xxx, t to xx, S $\phi$  to xxx. The overlines ( $^-$ ) refer to 30 min. time averaged values, and the prime ( $^\prime$ ) representing the deviations from the mean."

L169-171 will be modified using these suggestions.

L171: "All symbols and their corresponding units in this study are provided in Appendix F – Table F1."

- Yes, but even if you have it in Appendix, I think it is very helpful to have definitions also in the main text for the first time. It's strange that some symbols are explain, but not all.
- Although it seems obvious, please add explanation for t, which appears in the derivative of left hand side  $\partial t$ ).

The corresponding units will be added when the definitions are first mentioned for clarification.

P7 bullet points I-V: There are references to Figure 1, but instead, Equation 1 should be a primary reference. Please add reference to Eq. 1 for IV and V.

"Fig.1 and Eq. 1" will both be added for I-V for clarification

End of Section 2.4: In Fig 1 caption, it is said that the atmospheric sink is also not included in this study. Please also add the statement in the main text.

The sentence: "V. The combination of production and loss of CH4 from chemical reactions with OH (S'), which assumed to be negligible within the diurnal scale due to the slow reaction rate of CH4 with OH compared to the atmospheric residence time of OH (Patra et al., 2009)." will be revised as:

"V. The combination of production and loss of  $CH_4$  from chemical reactions with OH (S'), which assumed to be negligible within the diurnal scale due to the slow reaction rate of  $CH_4$  with OH compared to the atmospheric residence time of OH (Patra et al., 2009) and therefore will be omitted from Eq. (1)."

Figure H3: Please add units for the heights in the caption.

The figure legend will be changed to include "m a.g.l" and "at six different heights 4, 52, 92, 157, 227 and 301 m a.g.l" will be added to the figure caption.

**Responses to Reviewer #2**

**General Comments**

The manuscript presents a valuable analysis of long-term methane variability in Central Siberia using ZOTTO tower data, focusing on diurnal and seasonal amplitudes. The topic is timely and relevant, given the role of boreal wetlands in global methane budgets and climate feedbacks. Overall, complex issues related to atmospheric dynamics are well described and the thinking is easy to follow. The paper is generally well written, and the figures are informative. However, even to have a greater impact, the study would benefit from clearer articulation of its objectives, improved transparency regarding methodological novelty, and a more comprehensive discussion of the implications of the findings. Additionally, some interpretations require clarification, and several technical issues need attention.

Thank you for the constructive review and suggestions. We detail our responses below in blue.

**Specific Comments**

The introduction gives a lot of background, but the main goal of the study is not clearly stated. Especially, I would like to see more discussion about how the diurnal cycle of atmospheric methane has been used in previous studies and why it is studied here.

In the revised manuscript. L46-61will be rewritten as below with additional motivation of investigating the CH4 at ZOTTO at diurnal scale. Track changes will also be provided.

"The ZOTTO facility, situated in central Siberia, was established in 2006, and from April 2009 to February 2022, it had been continuously measuring CH4 at multiple heights up to 301 meters along with other atmospheric gases and their isotopic compositions as well as meteorological data (Winderlich et al., 2010, Tran et al., 2024). This long-term monitoring effort makes ZOTTO a valuable atmospheric research station, providing high-time-resolution (half-minute frequency) CH4 measurements in a key high-latitude (above 50° N) region. An early analysis of ZOTTO data by Winderlich (2012), covering the period 2009-2011, identified pronounced CH4 mole fraction enhancement during mid- to late-summer between August and July. These were suggested to be due to microbial activity in nearby wetlands and episodic emissions from Siberian forest fires during the summer. In light of the globally observed post-2006 increase in atmospheric CH4 now widely linked to enhanced microbial activity, there is a compelling motivation to revisit and extend this analysis with the new extended observations (2009-2021) for ZOTTO. To better understand the drivers behind the mid- to late-summer CH4 enhancements observed at ZOTTO in Winderlich (2012), also in this study we will zoom in from broad seasonal patterns to a more resolved diurnal and local-scale analysis.

The diurnal variations in atmospheric CH4 mole fraction are determined by interactions between surface CH4 emissions originating from wetlands, agriculture, and fossil fuel combustion (Metya et al., 2021) and atmospheric boundary layer (ABL) dynamics including daytime mixing through entrainment and nighttime stratification that traps near-surface emissions. Previous research showed that small-scale ABL processes, such as entrainment from the free troposphere and the daily evolution of boundary layer depth, can strongly influence observed tracer concentrations (Denning et al., 1996; Larson & Volkmer, 2008; Pino et al., 2012; Williams et al., 2011; Schuh & Jacobson, 2023; Faassen et al., 2025). Correctly representing these processes is essential when interpreting CH4 observations at diurnal scales and linking them to surface emissions or when

applying them in high-resolution models (Yi et al., 2004; Kretschmer et al., 2014; Bonan et al., 2024).

Despite its importance, long-term trends and drivers of CH4 diurnal cycle have received little attention compared to seasonal or annual CH4 changes. Previous studies of diurnal CH4 mole fractions from observational towers have typically focused on characterising the patterns of the diurnal cycle (e.g., Metya et al., 2021; Mahata et al., 2017) rather than examining long-term changes in these patterns. Moreover, they have not systematically quantified the relative contributions of potential drivers, such as the extent to which observed variations are controlled by surface processes versus atmospheric dynamics. Other research has mainly investigated short-term diurnal CH4 fluxes at the leaf or ecosystem scale under laboratory or field conditions (e.g. Takahashi et al., 2022; Kohl et al., 2023), or has analysed long-term CH4 fluxes derived from process-based models (Duan et al., 2025).

The goal of this study is to investigate long-term CH4 variability at ZOTTO from 2010 to 2021 across interannual, seasonal, and diurnal timescales. We first analyse interannual and seasonal CH4 patterns, focusing on the late-summer peak period observed in Winderlich et al., (2012). We then zoom into the local scale and examine trends and interannual variability in the CH4 mole fraction diurnal cycle during June-October, when these peaks are most pronounced. Specifically, we quantify changes in CH4 diurnal amplitude, defined as the difference between daily maximum and minimum mole fractions, to assess how the diurnal cycle has evolved over time. Finally, we utilise six-level CH4 mole fraction and meteorological profiles at ZOTTO to calculate local surface CH4 fluxes, to determine whether the observed trends and variability in the CH4 diurnal cycle are primarily driven by changes in surface fluxes (related to biological activity) or by atmospheric boundary-layer processes."

It is not fully clear which parts of the method are new and which follow earlier work (e.g., Winderlich). Please make this clear in Section 2.

The long-term, seasonal analysis as well as separating diurnal cycle into atmospheric and surface processes are our new study at ZOTTO. In the diurnal analyses, the surface flux calculation (Sect 2.5.2) is inherited from Winderlich et al. (2012).

In the revised manuscript, this will be clarified at L265 as:

"using the vertical CH4 profile at ZOTTO following the approach of Winderlich et al. (2012)."

The paper shows that nighttime fluxes have increased. Does this also mean daytime fluxes have changed? Please discuss this.

The main limitation of our study lies in the ability to reliably estimate daytime fluxes, which we have noted in Lines 310-315 and 545-547. The diurnal variability of CH4 flux does not follow a consistent pattern. Depending on the season, ecosystem types, and environmental conditions, CH4 fluxes may be higher during the day, lower at night, show the opposite behaviour, or remain relatively constant throughout the day (Kohl et al., 2023). If assuming the case when CH4 fluxes do not exhibit a strong diurnal cycle, the daytime net CH4 fluxes (although not calculated in our analysis) could indeed also be correlated with soil temperature and soil moisture and increase over

the 2010-2021 period, as observed for the nighttime fluxes. However, because our current dataset and analysis framework do not provide estimations on daytime fluxes, we chose not to include this speculative interpretation in the discussion and suggested future studies to attempt other method to derive daytime fluxes in L545-556.

L220: "Fits with an R2 value greater than 0.7 were retained.": Did this threshold value exclude a lot of data?

In the revised version, this will be clarified as

"Fits with an  $R^2$  value greater than 0.7 were retained. This process eliminated 14.5 % of the vertical potential temperature data at ZOTTO."

L232-233: "By examining interannual variations in the summer NBL height, we can assess whether the nighttime stability leading to accumulation of near-surface CH4 mole fractions have strengthened or weakened over time." Could you clarify what it would indicate if nighttime stability strengthened or weakened over time?

In the revised manuscript, the sentence has been clarified as follows:

"By examining interannual variations in the summer NBL height, where a lower NBL indicates stronger nighttime stability and a higher NBL indicates weaker stability, we can assess whether the nighttime conditions that promote accumulation of near-surface  $CH_4$  mole fractions have strengthened or weakened over time"

L335-338: Could you explain why the year 2021 is not mentioned? It has the highest annual growth rate in the MBL data and also high for the ZOTTO data.

In the revised manuscript, the growth rate of 2016 and 2021 will also be highlighted in Table 1 and add a discussion of these years and other highlighted years in the context of long-term trends, seasonal patterns. We will modify L441-444 as:

"The observed persistent increase in CH4 growth rates at ZOTTO is consistent with the global trend reported by NOAA and other long-term monitoring networks (Lan et al., 2021). Peaks in CH4 growth at ZOTTO, particularly in 2014, 2016, and 2019-2021, mirror global trends of increasing CH4 levels, which began around 2014, steepened after 2018, and further intensified in 2020 (Worden et al., 2017; Nisbet et al., 2019). The global CH4 increases in 2014 and 2020 have been largely attributed to reductions in atmospheric OH radical concentrations (Zhang et al., 2021). In 2020, this OH-driven effect was likely amplified by decreases in anthropogenic NOx emissions and the associated reduction in free-tropospheric ozone during the COVID-19 lockdowns (Peng et al., 2022). High global growth rates from 2016 to 2020 were additionally driven by strong emissions from boreal wetlands in Eurasia (Yuan et al., 2024; Zhang et al., 2021). A comparison of CH4 mole fraction time series between the inland tall tower at ZOTTO and the MBL reference at 60° N shows consistently higher CH4 concentrations at ZOTTO. However, only in some of the peak years highlighted in red in Table 1 (i.e., 2014, 2016, and 2020) does ZOTTO exhibit annual growth rates exceeding those of the MBL, suggesting that, in addition to the global

baseline increase, ZOTTO may have been influenced by additional regional sources in those years."

Although CH4 mole fraction data are available for 2021 at ZOTTO and show a pronounced growth rate, no potential temperature data were recorded that year due to an instrumentation malfunction. This limitation prevents a reliable estimation of nighttime surface fluxes for 2021, and therefore, this year was excluded from subsequent summer flux analyses aimed at explaining the observed growth rate.

L390-393: The detrending was done based on daytime data, right? So, in some sense, you removed the daytime trend, and it is no wonder that the trend in daytime values is close to zero after that. Or maybe the point is to show that the trend in the nighttime values is largerthan in daytime values, but that is already stated. Please, consider phrasing this paragraph differently.

In our study, we applied two kinds of detrending:

- For the seasonal and annual analysis Sect 3.1 and 3.2, the long-term trend that is removed is derived from daytime ZOT bg data.
- For the diurnal analysis (at 52 m), the long-term trend that is removed is derived from the full hourly time series (both day and night).

Thus, the detrending applied in the diurnal analysis is based on the complete dataset, not restricted to daytime data. We will clarify this in L389-390: "When the long-term anthropogenic trend derived from the full 52 m time series (including both daytime and nighttime data) is removed, the influence of nighttime  $CH_4$  becomes even more evident (Appendix H - Fig. H1)."

L426-433: It would be nice if you explain earlier or somewhere how these environmental factors are expected to affect methane fluxes.

In the revised manuscript, L27 in the introduction will be added:

"This rise in wetland emissions is suggested to result from enhanced microbial methane production driven by warmer soil moisture, soil temperature, and extended periods of inundation in tropical and high-latitude regions, all of which promote anaerobic conditions favourable for methanogenesis (Basu et al., 2022; Bridgham et al., 2013)"

L451-456: There are also remote sensing products that have been studied near the ZOTTO station that could be a suitable reference or comparison. (e.g. Kivimäki et al. 2025). L460-468: How about snow melting? How would that affect methane fluxes near ZOTTO? When does snow melt in ZOTTO? Air temperature seems to rise above zero in April-May, so I would assume that the snow melts after that.

In the revised manuscript, we will focus our diurnal and flux analysis on July to October instead of June to August as suggested by reviewer 1. We will include Feb-May snow depth data from ERA5 at 60°75′ N, 89°25′ E (the closest available grid point to ZOTTO) in our analysis of

environmental drivers of late-summer (July-September) CH4 fluxes at the site (see new Fig. R9). Figure 11 will be updated accordingly, and Lines 446-433 will be revised as follows:

"Given a clear increasing trend observed in the nighttime net  $CH_4$  surface flux, we further investigate the potential environmental drivers of the increase in this surface flux. The relationship between July-October averaged nighttime net  $CH_4$  surface flux and various environmental variables (Fig. 11) indicates strong positive correlations with July-October averaged soil temperature ( $R^2 = 0.65$ , p < 0.01), soil moisture at 32 cm below ground ( $R^2 = 0.36$ , p = 0.032), and with preceding Feb-May averaged snow depth ( $R^2 = 0.54$ , p = 0.029). This result suggests that warmer late-summer (July-October) soil conditions, higher soil moisture and thicker spring snow cover are associated with increased late-summer  $CH_4$  emissions."

In addition, the following paragraph will be added to the discussion section (after Line 532):

"Our results also reveal a strong positive relationship between spring snow depth and latesummer CH4 fluxes, indicating that deeper snow in the preceding winter-spring period could enhance CH4 emissions during late growing season. This finding is consistent with Kivimäki et al. (2025), using satellite observations, identified snow depth as a key driver of the variations in CH4 seasonality. We hypothesise that thicker winter snowpacks act as an insulating layer, maintaining warmer subsurface temperatures that promote CH4 production during winter. During spring, snowmelt of larger snowpacks lead to stronger increases soil moisture, creating and maintaining anaerobic conditions that persist longer throughout the late growing season. The flat topography, impermeable subsurface layers, and poor drainage characteristic of western Siberia further enhance water retention, while the warmer soil temperatures in July-October promote the persistence of methanogenesis that sustain elevated CH4 emissions in the late summer."

Figure R9. Relationship between July-October averaged nighttime net CH4 flux and July-October averaged precipitation, soil temperature at 32 cm depth, soil moisture at 32 cm depth, air temperature measured at 52 m a.g.l., vapour pressure deficit (VPD) during summer, and Feb-May averaged snow depth from ERA5 data at 60°75′ N, 89°25′ E. Shaded areas represent 95 % confidence intervals. The colour gradient in the data points indicates temporal trends, with more recent years (darker blue) tending toward higher temperatures, lower soil moisture, and increased VPD.

L489-490: "These regional emissions may have contributed to enhanced seasonal amplitude observed at ZOTTO." Do you think that the enhanced seasonal amplitude is due to winter maxima being higher or summer minimum being lower?

The results (Fig. R5) indicate an increasing trend in the winter CH4 maximum at ZOTTO, although this trend is not statistically significant (Fig. R5). The CH4 annual minima also exhibits a statistically insignificant decreasing trend, but this trend is smaller compared to the increase in the winter maximum. The increase in the ZOTTO seasonal amplitude (Fig. 6 in the manuscript), while also not significant, appears to be mainly driven by this rise in winter CH4 mole fraction maxima

accompanied by the slight decrease in the CH4 spring minima. This may reflect either a larger influence of regional wetland and/or anthropogenic emissions during winter or an increased occurrence of persistent, stable high-pressure systems, which can enhance the accumulation of CH4 near the surface under limited atmospheric mixing conditions. To further investigate the contributors to the increasing winter CH4 mole fractions at ZOTTO, further studies using atmospheric inversion techniques and isotopic analysis are required, which is outside the scope of this study.

There is a significant increasing trend in the late-summer CH4 peak (1.35 ppb year-1, p = 0.02) (Fig. R5). Therefore, it is reasonable to conclude that the observed increase in the late-summer CH4 peak amplitude (as shown in Fig. 7 in the manuscript) is primarily driven by the significant rise in the late-summer CH4 peak rather than by a decrease in springtime minima.

In the revised manuscript, Fig. R5 will be added in the appendix, and additional analyses will be added after L358:

"The increase in the ZOTTO seasonal amplitude (Fig. 6), while not statistically significant, appears to be mainly driven by an increasing winter  $CH_4$  mole fraction maximum (1.42 ppb year  $^1$ , p = 0.17) accompanied by a slight decrease in the  $CH_4$  spring minima (-0.69 ppb year  $^1$ , p = 0.32) (Appendix – Fig...)."

**And after L370:**

"The enhanced late-summer peak amplitude (Fig. 7) is primarily attributed to the strong significant increase in the late-summer  $CH_4$  maximum (1.35 ppb year  $^1$ , p=0.02), rather than to a slight decrease in the springtime minimum (-0.69 ppb year  $^1$ , p=0.32) (Appendix... – Fig...)."

L520-522: "These findings align with studies from other boreal and wetland-dominated regions, where CH4 emissions peak in late summer due to sustained high soil temperature and moisture leading to high microbial activity." Could you expand on this point? For example, Rößger et al. (2022) also reported the highest methane emissions in August, but the long-term increase differs: Rößger et al. found the trend in July, while your results suggest the increase occurs in August.

Additional comparison for the study of Rößger et al. (2022) will be added in the discussion after L522:

"Similar to our finding at ZOTTO, Rößger et al. (2022) also observed pronounced seasonal CH4 flux peaks in both July and August in the North Siberian Lena River Delta tundra site (72.37° N, 126.50° E) (2002-2019). However, they found that long-term increases in CH4 emissions were limited to the early summer months (June-July), with no significant upward trend in August fluxes, despite August being the period of maximum emissions. The main differences in the August CH4 flux trends between our study at ZOTTO and Rößger et al. (2022) likely stem from differences in the long-term trends of environmental drivers at each site, particularly soil temperature. Rößger et al. (2022) attributed the stability of August fluxes at the Lena River Delta to relatively insignificant small increases in August soil temperature over their study period. In contrast, the increase in late summer fluxes observed at ZOTTO is significantly positively correlated with rising soil temperature, soil moisture during the late summer, and snow depth in the preceding spring during the 2010-2021 period."

**Technical Corrections**

It would be nice if the figures and Appendices mentioned would be in order, i.e., that the first mentioned would have A1, next one A2 or B1 etc.

We group the appendices into separate categories: Appendix A, Appendix B, Appendix C, etc., based on their content. Within each category, the figures will be numbered sequentially according to the order in which they are first referenced in the main text.

L22: "global greenhouse gases radiative forcing" -> "global greenhouse gas radiative forcing".

L24: "has risen" -> "rose"

L185: "as mentioned in" -> "similarly to" or "following..."

L301: "Equation 6 as in Winderlich et al., 2014 can be expanded" -> "Equation 6, as in

Winderlich et al., (2014), can be expanded"

L303: "Where" -> "where"

L384: "Between 2010 and 2021, the summer diurnal amplitude increased significantly at p = 0.01 level at a rate of 5.55 ppb yr-1 (p = 0.002)": Maybe you could form it a bit differently:

"Between 2010 and 2021, the summer diurnal amplitude increased significantly (p < 0.01) at a rate of 5.55 ppb yr-1 (p = 0.002)."

L453: "signal has" -> "signal had"

These technical comments above will be modified accordingly in the revised manuscript.

L405-415: You could first present the results and then conclude that based on that increasing nighttime CH4 surface fluxes during summer are the primary driver of the observed rise in nighttime CH4 mole fractions.

We appreciate the idea of first presenting the results and then concluding that increasing nighttime CH4 surface fluxes during summer are the main driver of the observed rise in nighttime CH4 mole fractions. However, we chose to emphasise our key finding first, followed by the supporting results, to maintain a clear narrative focus and highlight the main conclusion upfront.

L25: Reference for the atmospheric methane levels missing.

Citation (Lan et al., 2025) will be added in the revised manuscript

L31: The global methane budget has been updated (Saunois et al. (2025)), you could also update this reference.

The updated global methane budget (Saunois et al. (2025)) will be updated in the revised version.

L34-35: Do the area of undisturbed and disturbed boreal and temperate peatlands overlap with the area of permafrost region? I.e., can you just sum up the two numbers mentioned in the text to get peatland emission from northern peatlands?

The two estimates cannot be directly summed because they are derived from studies with different spatial domains and methodologies. The values from Folking et al. (2011) and Olsson et al. (2019) represent global estimates for undisturbed and disturbed boreal and temperate peatlands, providing an overview number based on of existing research up to the time of publication. In contrast, Hugelius et al. (2024) report average methane emissions from the entire northern permafrost region (2000-2020), which includes multiple land-cover types, peatlands among them. Therefore, simply

adding these numbers would result in double counting of peatland areas within the permafrost zone and lead to an overestimation of total northern peatland emissions.

L90: "Data were recorded every 30 seconds from each sampling line.": What does the "30 seconds" refer to exactly? Does it mean that there is a measurement taken every 30 seconds from the currently active sampling lines?

Yes, this is correct. There is one gas analyser connected to six sampling lines corresponding to six tower heights. Because only one analyser is available, it sequentially switches between the sampling lines, measuring each height for 3 minutes before moving to the next (full cycle of 6 heights = 18 minutes). During the 3-minute sampling period for a given height, the analyser records data every 30 seconds, resulting in six measurements per line. However, the first 1.5 minutes of each sampling period are discarded to allow the air from the selected height to fully flush through the tubing and stabilise. Thus, only the last 1.5 minutes (approximately three data points) are retained as valid measurements for that height. We will modify L90 as:

"Data were recorded every 30 seconds from each active sampling line."

L208: Was the ERA5 reanalysis data you used "boundary layer height"?

The atmospheric boundary layer (ABL) was treated separately for daytime (referred to as convective daytime boundary layer (CBL)) and nighttime (referred to as nighttime boundary layer (NBL)). For the CBL, we indeed used ERA5 reanalysis data. For the NBL height, we detailed the derivation as Section 2.5.1.1. We believe we have sufficiently clarified this in L207-215 in the manuscript.

L435, Figure 11: Why is the x-axis of VPD flipped?

The axis of VPD is flipped because it is essentially a visual choice to maintain consistency in the direction of positive correlations across the figure. We will add this in Fig. 11 figure caption in the revised manuscript.

L474: "natural gas emissions sum up to 1 to 10 % of the overall wetland emissions only". Which area and time period is being referred to here?

We will clarify this in the revised version as: "Natural gas emissions sum up to 1 to 10 % of the overall wetland emissions only during the 1999 to 2003 period."

L590, Figure A1: Consider using a "neutral" (neither blue or red) colour at 0 °C in the colour bar to make it easier to see where the air temperature crosses the freezing point.

Figure A1.a will be updated as below with colour at 0 °C in the colour bar 'neutral'.

**References:**

Basu, S., Lan, X., Dlugokencky, E., Michel, S., Schwietzke, S., Miller, J. B., Bruhwiler, L., Oh, Y., Tans, P. P., Apadula, F., Gatti, L. V., Jordan, A., Necki, J., Sasakawa, M., Morimoto, S., Di Iorio, T., Lee, H., Arduini, J., and Manca, G.: Estimating emissions of methane consistent with atmospheric measurements of methane and  $\delta 13C$  of methane, Atmos. Chem. Phys., 22, 15351–15377, https://doi.org/10.5194/acp-22-15351-2022, 2022.

Bridgham, S. D., Cadillo-Quiroz, H., Keller, J. K., and Zhuang, Q.: Methane emissions from wetlands: biogeochemical, microbial, and modeling perspectives from local to global scales, Glob. Change Biol., 19, 1325–1346, https://doi.org/10.1111/gcb.12131, 2013.

Denning, A. S., Randall, D. A., Collatz, G. J., and Sellers, P. J.: Simulations of terrestrial carbon metabolism and atmospheric CO2 in a general circulation model: Part 2. Simulated CO2 concentrations, Tellus B, 48, 543–567, https://doi.org/10.1034/j.1600-0889.1996.t01-1-00010.x, 1996.

Duan, W., Wu, M., Peichl, M., He, H., Roulet, N., Noumonvi, K. D., Ratcliffe, J. L., Nilsson, M. B., and Jansson, P.-E.: Seasonal and diurnal patterns of methane emissions from a northern pristine peatland in the last decade, Glob. Biogeochem. Cycles, 39, e2025GB008518, https://doi.org/10.1029/2025GB008518, 2025

Faassen, K. A. P., González-Armas, R., Koren, G., Adnew, G. A., van Asperen, H., de Boer, H., Botía, S., de Feiter, V. S., Hartogensis, O., Heusinkveld, B. G., Hulsman, L. M., Hutjes, R. W. A., Jones, S. P., Kers, B. A. M., Komiya, S., Machado, L. A. T., Martins, G., Miller, J. B., Mol, W., Moonen, R., Dias-Junior, C. Q., Röckmann, T., Snellen, H., Luijkx, I. T., and Vilà-Guerau de Arellano, J.: Tracing diurnal variations of atmospheric CO2, O2, and δ13CO2 over a tropical and a temperate forest, Geophys. Res. Lett., 52, e2025GL118016, https://doi.org/10.1029/2025GL118016, 2025

Kariyathan, T., Bastos, A., Marshall, J., Peters, W., Tans, P., and Reichstein, M.: Reducing errors on estimates of the carbon uptake period based on time series of atmospheric CO2, Atmos. Meas. Tech., 16, 3299–3312, https://doi.org/10.5194/amt-16-3299-2023, 2023.

Kivimäki, E., Tenkanen, M. K., Aalto, T., Buchwitz, M., Luojus, K., Pulliainen, J., Rautiainen, K., Schneising, O., Sundström, A.-M., Tamminen, J., Tsuruta, A., and Lindqvist, H.: Environmental

- drivers constraining the seasonal variability in satellite-observed and modelled methane at northern high latitudes, Biogeosciences, 22, 5193–5230, https://doi.org/10.5194/bg-22-5193-2025, 2025
- Kohl, L., Tenhovirta, S. A. M., Koskinen, M., Putkinen, A., Haikarainen, I., Polvinen, T., Galeotti, L., Mammarella, I., Siljanen, H. M. P., Robson, T. M., Adamczyk, B., & Pihlatie, M.: Radiation and temperature drive diurnal variation of aerobic methane emissions from Scots pine canopy, Proc. Natl. Acad. Sci. U.S.A., 120, e2308516120, https://doi.org/10.1073/pnas.2308516120, 2023.
- Kretschmer, K., Biastoch, A., Rüpke, L. H., and Burwicz, E.: Modeling the fate of methane hydrates under global warming, Glob. Biogeochem. Cycles, 29, 610–625, https://doi.org/10.1002/2014GB005011, 2015.
- Kurzeja, R. J., Leclerc, M. Y., Duarte, H. F., Zhang, G., Parker, M. J., Werth, D. W., Chiswell, S. R., and Buckley, R. L.: Turbulence and diffusion on weakly stable and stable nights near a 300 m tower in a complex landscape, J. Atmos. Sci., 80, 211–233, https://doi.org/10.1175/JAS-D-21-0268.1, 2023
- Lan, X., Basu, S., Schwietzke, S., Bruhwiler, L., Tans, P. P., and Dlugokencky, E. J.: Improved constraints on global methane emissions and sinks using δ13C-CH4, Glob. Biogeochem. Cycles, 35, e2021GB007000, https://doi.org/10.1029/2021GB007000, 2021.
- Larson, V. E. and Volkmer, H.: An idealized model of the one-dimensional carbon dioxide rectifier effect, Tellus B, 60, 525–536, https://doi.org/10.1111/j.1600-0889.2008.00368.x, 2008.
- Metya, A., Datye, A., Chakraborty, S., Tiwari, Y. K., Sarma, D., Bora, A., and Gogoi, N.: Diurnal and seasonal variability of CO2 and CH4 concentration in a semi-urban environment of western India, Sci. Rep., 11, 2931, https://doi.org/10.1038/s41598-021-82321-1, 2021.
- Nisbet, E. G., Manning, M. R., Dlugokencky, E. J., Fisher, R. E., Lowry, D., Michel, S. E., Myhre, C. L., Platt, S. M., Allen, G., Bousquet, P., Brownlow, R., Cain, M., France, J. L., Hermansen, O., Hossaini, R., Jones, A. E., Levin, I., Manning, A. C., Myhre, G., Pyle, J. A., Vaughn, B. H., Warwick, N. J., and White, J. W. C.: Very strong atmospheric methane growth in the 4 years 2014–2017: implications for the Paris Agreement, Glob. Biogeochem. Cycles, 33, 318–342, https://doi.org/10.1029/2018GB006009, 2019.
- Peng, S., Lin, X., Thompson, R. L., Xi, Y., Liu, G., Hauglustaine, D., Lan, X., Poulter, B., Ramonet, M., Saunois, M., Yin, Y., Zhang, Z., Zheng, B., and Ciais, P.: Wetland emission and atmospheric sink changes explain methane growth in 2020, Nature, 612, 477–482, https://doi.org/10.1038/s41586-022-05447-w, 2022.
- Pickers, P. A., Manning, A. C., and Le Quéré, C.: Investigating bias in the application of curve-fitting programs to atmospheric time series, Atmos. Meas. Tech., 8, 1469–1482, https://doi.org/10.5194/amt-8-1469-2015, 2015.
- Pino, D., Vilà-Guerau de Arellano, J., Peters, W., Schröter, J., van Heerwaarden, C. C., and Krol, M.: A conceptual framework to quantify the influence of convective boundary layer development on carbon dioxide mixing ratios, Atmos. Chem. Phys., 12, 2969–2985, https://doi.org/10.5194/acp-12-2969-2012, 2012.
- Ranniku, R., Kazmi, F. A., Espenberg, M., Truupõld, J., Escuer-Gatius, J., Mander, Ü., Soosaar, K., and Nõges, P.: Springtime soil and tree stem greenhouse gas fluxes and the related soil microbiome pattern in a drained peatland forest, Biogeochemistry, 168, 48, https://doi.org/10.1007/s10533-025-01238-3, 2025.

- Rößger, N., Sachs, T., Kling, G., Eichelmann, E., Rudaya, N., Kutzbach, L., Nöchel, U., Friborg, T., Turetsky, M. R., and Tanski, G.: Seasonal increase of methane emissions linked to warming in Siberian tundra, Nat. Clim. Change, 12, 325–332, https://doi.org/10.1038/s41558-022-01512-4, 2022.
- Saunois, M., Stavert, A. R., Poulter, B., Bousquet, P., Canadell, J. G., Jackson, R. B., and the Global Methane Budget Team: The Global Methane Budget 2000–2020, Earth Syst. Sci. Data, 17, 1873–1958, https://doi.org/10.5194/essd-17-1873-2025, 2025.
- Schuh, A. E. and Jacobson, A. R.: Uncertainty in parameterized convection remains a key obstacle for estimating surface fluxes of carbon dioxide, Atmos. Chem. Phys., 23, 6285–6297, https://doi.org/10.5194/acp-23-6285-2023, 2023.
- Takahashi, Y., Hashimoto, T., and Morimoto, S.: Insights into the mechanism of diurnal variations in methane emission from the stem surfaces of Alnus japonica, New Phytol., 236, https://doi.org/10.1111/nph.18283, 2022.
- Thoning, K. W., Tans, P. P., and Komhyr, W. D.: Atmospheric carbon dioxide at Mauna Loa Observatory: 2. Analysis of the NOAA GMCC data, 1974–1985, J. Geophys. Res., 94, 8549–8565, https://doi.org/10.1029/JD094iD06p08549, 1989.
- Tran, D. A., Gerbig, C., Rödenbeck, C., and Zaehle, S.: Interannual variations in Siberian carbon uptake and carbon release period, Atmos. Chem. Phys., 24, 8413–8440, https://doi.org/10.5194/acp-24-8413-2024, 2024.
- Williams, I. N., Riley, W. J., Torn, M. S., Berry, J. A., and Biraud, S. C.: Using boundary layer equilibrium to reduce uncertainties in transport models and CO2 flux inversions, Atmos. Chem. Phys., 11, 9631–9641, https://doi.org/10.5194/acp-11-9631-2011, 2011.
- Worden, J. R., Bloom, A. A., Pandey, S., Jiang, Z., Worden, H. M., Walker, T. W., Houweling, S., and Röckmann, T.: Reduced biomass burning emissions reconcile conflicting estimates of the post-2006 atmospheric methane budget, Nat. Commun., 8, 2227, https://doi.org/10.1038/s41467-017-02246-0, 2017.
- Yi, C., Davis, K. J., Bakwin, P. S., Denning, A. S., Zhang, N., Desai, A., Lin, J. C., and Gerbig, C.: Observed covariance between ecosystem carbon exchange and atmospheric boundary layer dynamics at a site in northern Wisconsin, J. Geophys. Res., 109, D08302, https://doi.org/10.1029/2003JD004164, 2004.
- Yuan, K., Li, F., McNicol, G., Chen, M., Hoyt, A., Knox, S., Riley, W. J., Jackson, R. B., and Zhu, Q.: Boreal–Arctic wetland methane emissions modulated by warming and vegetation activity, Nat. Clim. Change, 14, 282–288, https://doi.org/10.1038/s41558-024-01933-3
- Zhang, Y., Jacob, D. J., Lu, X., Maasakkers, J. D., Scarpelli, T. R., Sheng, J.-X., Shen, L., Qu, Z., Sulprizio, M. P., Chang, J., Bloom, A. A., Worden, J., Parker, R. J., and Boesch, H.: Attribution of the accelerating increase in atmospheric methane during 2010–2018 by inverse analysis of GOSAT observations, Atmos. Chem. Phys., 21, 3643–3667, https://doi.org/10.5194/acp-21-3643-2021, 2021.